# On the Convergence of Stochastic Multi-Objective Gradient Manipulation and Beyond

**Shiji Zhou**[*][†]
Tsinghua University
zhoushiji00@gmail.com

**Wenpeng Zhang**[*][†]
Ant Group
zhangwenpeng0@gmail.com

**Jiyan Jiang**
Tsinghua University
scjjy95@outlook.com

**Wenliang Zhong**
Ant Group
yice.zwl@antgroup.com

**Jinjie Gu**
Ant Group
jinjie.gujj@antgroup.com

**Wenwu Zhu**[†]
Tsinghua University
wwzhu@tsinghua.edu.cn

## Abstract

The conflicting gradients problem is one of the major bottlenecks for the effective training of machine learning models that deal with multiple objectives. To resolve this problem, various gradient manipulation techniques, such as PCGrad, MGDA, and CAGrad, have been developed, which directly alter the conflicting gradients to refined ones with alleviated or even no conflicts. However, the existing design and analysis of these techniques are mainly conducted under the full-batch gradient setting, ignoring the fact that they are primarily applied with stochastic mini-batch gradients. In this paper, we illustrate that the stochastic gradient manipulation algorithms may fail to converge to Pareto optimal solutions. Firstly, we show that these different algorithms can be summarized into a unified algorithmic framework, where the descent direction is given by the composition of the gradients of the multiple objectives. Then we provide an explicit two-objective convex optimization instance to explicate the non-convergence issue under the unified framework, which suggests that the non-convergence results from the determination of the composite weights solely by the instantaneous stochastic gradients. To fix the non-convergence issue, we propose a novel composite weights determination scheme that exponentially averages the past calculated weights. Finally, we show the resulting new variant of stochastic gradient manipulation converges to Pareto optimal or critical solutions and yields comparable or improved empirical performance.

## 1 Introduction

In many real-world application scenarios, the deployed machine learning models are designed to deal with multiple objectives. For example, in online advertising, the models need to maximize both the Click-Through Rate and the Post-Click Conversion Rate [33, 34, 27, 43]. In autonomous driving, the automatic pilots learn to simultaneously solve multiple tasks such as object identification and self-localization [14, 32]. Despite being widely used, multi-objective machine learning models is known to be very challenging to train as different objectives may conflict with each other, creating the conflicting gradients problem [19, 46, 28]. As is well acknowledged, the conflicting gradients are

---

[*]Equal contributions      [†]Corresponding authors

36th Conference on Neural Information Processing Systems (NeurIPS 2022).

usually detrimental to the model training as they pull the model parameters into conflicting directions. Severe gradient conflicts will lead to significantly degraded model performance.

To resolve the conflicting gradients problem, various multi-objective gradient manipulation (MOGM) techniques have been proposed, including MGDA [38], PCGrad [46], and CAGrad [28] etc. These techniques alleviate the conflicts between gradients by directly manipulating them. Typically, they alter the multiple gradients to seek a kind of common descent direction that can (approximately) simultaneously descend all the losses w.r.t. the multiple objectives. While they have been widely applied and achieved significant empirical success, their current design and analysis are merely derived with exact (i.e., full-batch) gradients [38, 46, 28, 42, 29, 16]. In contrast, in practice, these techniques are usually applied with stochastic (i.e., mini-batch) gradients, especially for training deep neural networks (DNNs) [22, 36], as using stochastic gradients is much more practical and efficient.

In the single-objective setting, since the stochastic gradients are unbiased estimators of the true gradients, directly using the stochastic gradients still maintains the right expected direction to optimize the objective. However, this is not the case for the multi-objective setting when MOGM is used. In the multi-objective case, since altering the stochastic gradients could amplify the negative impact of the randomness of the gradients, the common descent direction for the stochastic objectives may not be the right direction to descend the true objectives in expectation. Such gap between full and stochastic gradients could cause non-convergence issues for directly using MOGM algorithms.

In this paper, we rigorously show that the above non-convergence issues indeed exist. Surprisingly, we prove that *altering the stochastic gradients yields a direction that even inversely optimizes all the objectives in expectation* for some cases. Such an inverse direction that moves the algorithms in the wrong direction will resist the solution from convergence. This implies that current MOGM algorithms are suboptimal under stochastic gradient settings, which potentially degrades the performance of MOGM algorithms applied for training DNNs. In particular, we make the following key contributions.

To uniformly analyze different MOGM algorithms, we summarize them into a general framework where the altered direction is a weighted composition of the multiple gradients, and the composite weights are functions of the multiple gradients determined by the alteration rules of different MOGM algorithms. We provide strictly equivalent transformation to show that typical methods such as MGDA, PCGrad, and CAGrad are all included in the summarized form. This framework is general and incorporates a wide range of algorithms, such as recently proposed GradVac [42], IMTL-G [29].

Under the above general framework, we elucidate how the MOGM algorithms can cause non-convergence by providing an example of a simple stochastic convex optimization problem, where MGDA, PCGrad, and CAGrad provably do not converge to an optimal solution. The key reason is that the composite weights are calculated only from the instantaneous gradients in one step, and thus inherit the same randomness as the stochastic gradients. This strong correlation between the composite weights and stochastic gradients could lead to significant estimation bias of their composition, which forces the composite gradient in the wrong direction that even degrades all the objectives. Since MGDA, PCGrad, and CAGrad possess typical properties among MOGM algorithms, our analysis easily extends to other MOGM algorithms contained in our general framework as well. Furthermore, we point out that the previous analysis of the convergence for the MGDA algorithm [31] is established on an unjustified assumption, where the composite weights are Lipschitz functions w.r.t. each gradient. We rigorously prove that this assumption is not true for MOGM algorithms.

The above analysis suggests that in order to achieve guaranteed convergence, the optimization algorithms must reduce the correlation between the composite weights and stochastic gradients. To resolve this issue while preserving the practical advantages of MOGM, we fix MOGM algorithms with a principled and practical momentum mechanism that exponentially averages the composite weights instead of just using the values computed from the instantaneous gradients. Theoretically, momentum reduces the variance of the composite weight [4], then decreasing the correlation as well. Thus, the inverse direction problem can be alleviated, and we prove that gradient manipulation algorithms can converge to optimal with the same rate as single objective stochastic optimization by adopting this mechanism. Our approach is simple to implement and agnostic to model and optimizer, and hence can be easily plugged into various gradient manipulation algorithms.

Finally, we empirically verify our proposal in simulation and deep multi-task learning tasks, where we observe comparable or better performance. Our advantage becomes more significant when the batch size is smaller, implying that the convergence issue is more severe with larger stochastic noise.

## 2 Preliminaries

In this section, we briefly review the basic background knowledge of multiple-objective optimization, and introduce typical algorithms as well as their convergence analysis.

### 2.1 Basic Concepts of Multiple-Objective Optimization

Multiple-objective optimization (MOO) is concerned with solving the problems of optimizing multiple objective functions simultaneously [48, 38], termed as

$$\min_{\boldsymbol{x} \in \mathcal{K}} \boldsymbol{F}(\boldsymbol{x}) = (f^1(\boldsymbol{x}), \dots, f^m(\boldsymbol{x}))^\top, \tag{1}$$

where $m \geq 2$ denotes the number of objectives, and $\mathcal{K} \subset \mathbb{R}^d$ denotes the feasible set in a $d$ dimension space. Each objective $f^i : \mathcal{K} \to \mathbb{R}, i \in \{1, \dots, m\}$ is the $i$-th loss function. Different from single-objective optimization where solutions $\boldsymbol{x}, \boldsymbol{x}'$ can be order by $f(\boldsymbol{x}) \leq f(\boldsymbol{x}')$ or $f(\boldsymbol{x}) \geq f(\boldsymbol{x}')$. In MOO, it could have two parameter vectors where one performs better for task $i$ and the other performs better for task $j \neq i$. Therefore, Pareto optimality is defined to deal with such an incomparable case.

**Definition 1** (**Pareto optimality**). *For any two solutions $\boldsymbol{x}, \boldsymbol{x}' \in \mathcal{K}$, we say that $\boldsymbol{x}$ dominates $\boldsymbol{x}'$, denoted as $\boldsymbol{x} \prec \boldsymbol{x}'$ or $\boldsymbol{x}' \succ \boldsymbol{x}$, if $f^i(\boldsymbol{x}) \leq f^i(\boldsymbol{x}')$ for all $i$, and there exists one $i$ such that $f^i(\boldsymbol{x}) < f^i(\boldsymbol{x}')$; otherwise, we say that $\boldsymbol{x}$ does not dominate $\boldsymbol{x}'$, denoted as $\boldsymbol{x} \not\prec \boldsymbol{x}'$ or $\boldsymbol{x}' \not\succ \boldsymbol{x}$. A solution $\boldsymbol{x}^* \in \mathcal{K}$ is called Pareto optimal if it is not dominated by any other solution in $\mathcal{K}$.*

Note that there is a set of Pareto optimal solutions, termed as **Pareto set**. The goal of MOO is to find a Pareto optimal solution, which is necessary to be Pareto critical [3].

**Definition 2** (**Pareto criticality**). *A solution $\boldsymbol{x}^* \in \mathcal{K}$ is called Pareto critical if there is no common descent direction $\boldsymbol{d} \in \mathbb{R}^d$ such that $\nabla f^i(\boldsymbol{x}^*)^\top \boldsymbol{d} < 0, i = 1, \dots m$ for all objectives.*

This definition indicates that if $\boldsymbol{x}$ is not Pareto critical, such direction $\boldsymbol{d}$ will be a local descent direction for $\boldsymbol{F}$ at point $\boldsymbol{x}$. Optimizing through $\boldsymbol{d}$ in the local neighborhood of $\boldsymbol{x}$ is able to get a better solution that dominates $\boldsymbol{x}$ [8]. Since Pareto criticality reflects the local property compared with Pareto optimality, it is often used as the local minimal condition for MOO with non-convex objectives [9]. We then present sufficient conditions for determining Pareto criticality/optimality, which appear as metrics to study the convergence for MOO algorithm [9, 41]. Denotes the probability simplex as $S_m = \{(\lambda^1, \dots, \lambda^m) | \sum_{i=1}^m \lambda^i = 1, \lambda \in [0, 1]\}$, we have the following proposition.

**Proposition 1.** *For MOO problem 1, (a) If there exists $\boldsymbol{\lambda} \in S_m$ such that $\| \sum_{i=1}^m \lambda^i \nabla f^i(\boldsymbol{x}^*)\| = 0$, then $\boldsymbol{x}^* \in \mathcal{K}$ is Pareto critical. (b) For convex objectives $f^i(\cdot), i = 1, \dots, m$. If there exists $\boldsymbol{\lambda} \in S_m$ such that $\boldsymbol{x}^* = \arg\min_{\boldsymbol{x} \in \mathcal{K}} \boldsymbol{\lambda}^T \boldsymbol{F}(\boldsymbol{x})$, then $\boldsymbol{x}^*$ is Pareto optimal[2].*

(a) is derived from the equivalent definition of Pareto critical points in [28], it reflects the stationary condition for MOO algorithms. (b) refers to Theorem 5.13 and Lemma 5.14 in [15], it implies that the minimizer of any linearization is Pareto optimal for the multiple objectives.

### 2.2 Multi-Objective Gradient Manipulation

Similar to single-objective optimization, MOO can be solved by running iteratively with gradient-based algorithms. Suppose $n$ denotes the number of iterations. At each $k = 1, \dots, n$ round, the key challenge is the gradient conflicting problem, i.e., $\nabla f^i(\boldsymbol{x}_k)^\top \nabla f^j(\boldsymbol{x}_k) < 0, i \neq j, i, j \in [m] = \{1, \dots, m\}$, which leads to performance drop for an objective when optimizing another one [46], i.e. optimizing $f^i$ along the negative gradient direction $-\nabla f^i(\boldsymbol{x}_k)$ will inversely optimize $f^j$. multi-objective gradient manipulation (MOGM) algorithms aim to search for a direction $\boldsymbol{d}_k$ that is not conflicting with each negative gradient, i.e., $-\nabla f^i(\boldsymbol{x}_k)\boldsymbol{d}_k > 0, i \in [m]$. Using such a non-conflicting direction $\boldsymbol{d}_k$ to execute the gradient descent step to update the decision i.e., $\boldsymbol{x}_{k+1} = \boldsymbol{x}_k + \eta \boldsymbol{d}_k$ where $\eta$ is the step size, is shown to get better performance in practice [38, 46, 28]. Mathematically, we can measure the decrease of each objective $i \in [m]$

$$f^i(\boldsymbol{x}_k) - f^i(\boldsymbol{x}_k + \eta \boldsymbol{d}_k) \approx -\eta \boldsymbol{d}_k^\top \nabla f^i(\boldsymbol{x}_k)$$

---

[2]Precisely, $\boldsymbol{x}^*$ is weak Pareto optimal. If $\boldsymbol{x}^*$ is the unique minimizer, $\boldsymbol{x}^*$ is Pareto optimal. For convenience of understanding, we do not distinguish them in the main text, and provide detailed comparison in Appendix A

by the first-order Taylor approximation assuming $\eta$ is small [28]. Therefore, if $-\boldsymbol{d}_k^\top \nabla f^i(\boldsymbol{x}_k) > 0$ for each objective $i$, then this direction is able to descend all objectives simultaneously. There are various ways to compute such kind of direction, and we'll introduce typical MOGM methods as follows.

**Multiple Gradient Descent Algorithm (MGDA)** [38] directly optimizes towards the Pareto criticality by leveraging Definition 2. Specifically, in each iteration $k$, MGDA aims to find a direction $\boldsymbol{d}_k$ to maximize the minimum decrease across the losses by solving the following subproblem

$$\max_{\boldsymbol{d}\in\mathbb{R}^d} \min_{i\in[m]} (f^i(\boldsymbol{x}_k) - f^i(\boldsymbol{x}_k + \eta\boldsymbol{d}_k)) \approx -\eta \min_{\boldsymbol{d}\in\mathbb{R}^d} \max_{i\in[m]} \nabla f^i(\boldsymbol{x}_k)^\top \boldsymbol{d}_k.$$

By regularizing the norm of $\boldsymbol{d}_k$ on the right side, it computes the direction by

$$\boldsymbol{d}_k = \arg\min_{\boldsymbol{d}\in\mathbb{R}^d}\{\max_{i\in[m]} \nabla f_i(\boldsymbol{x}_k)^\top \boldsymbol{d} + \frac{1}{2}\|\boldsymbol{d}\|_2^2\}.$$

This sub-problem can be rewritten equivalently as the following differentiable quadratic optimization

$$\boldsymbol{d}_k, \mu = \arg\min_{\{d\in\mathbb{R}^d, \mu\in\mathbb{R}\}} (\frac{1}{2}\|\boldsymbol{d}\|_2^2 + \mu), \text{ s.t. } \nabla f_i(\boldsymbol{x}_k)^\top \boldsymbol{d} \leq \mu.$$

If $\mu < 0$, then $\nabla f_i(\boldsymbol{x}_k)^\top \boldsymbol{d}_k < 0$, which means $\boldsymbol{x}_k$ is not Pareto critical from Definition 2, and $\boldsymbol{d}_k$ is the direction to descent all the objectives simultaneously [8, 9]. To simplify the optimization, such primal problem has a dual objective as a min-norm oracle

$$\boldsymbol{\lambda}_k = \arg\min_{\lambda_k\in S_m} \|\sum_{i=1}^m \lambda_k^i \nabla f^i(\boldsymbol{x}_k)\|.$$

The direction is then calculated by $\boldsymbol{d}_k = -\sum_{i=1}^m \lambda_k^i \nabla f^i(\boldsymbol{x}_k)$.

**Projecting Conflicting Gradients (PCGrad)** [46] alters the conflicting gradients by projecting each onto the normal plane of the other, preventing the interfering components of the gradient from being applied to the network. Specifically, it initializes the projected gradients as $\boldsymbol{v}_i^{\text{PC}} = \nabla f^i(\boldsymbol{x}_k)$. If projected gradient $\boldsymbol{v}_i$ and raw gradient $\nabla f^j(\boldsymbol{x}_k)$ are conflicting, i.e., $\boldsymbol{v}_i^\top \nabla f^j(\boldsymbol{x}_k) < 0$, then it iteratively updates the projected gradients by

$$\boldsymbol{v}_i^{\text{PC}} = \boldsymbol{v}_i^{\text{PC}}(\boldsymbol{x}_k) - \frac{\boldsymbol{v}_i^{\text{PC}}(\boldsymbol{x}_k)^\top \nabla f^j(\boldsymbol{x}_k)}{\|\nabla f^j(\boldsymbol{x}_k)\|_2^2} \nabla f^j(\boldsymbol{x}_k),$$

for $i, j = 1, \ldots, m$ until the projected gradients do not conflict with the received gradients. Since the conflicting components are overall eliminated, it makes the direction $\boldsymbol{d}_k = -\frac{1}{m}\sum_{i=1}^m \boldsymbol{v}_i^{\text{PC}}$ satisfies $\boldsymbol{d}_k^\top \nabla f^i(\boldsymbol{x}_k) < 0$ for each objective $i$, thus optimizing all objectives simultaneously.

**Conflict-Averse Gradient (CAGrad)** [28] generalizes averaging linearization and MGDA. Specifically, it constrains search region for the common direction as a circle around the average gradient $\boldsymbol{v}_0 = \sum_{i=1}^m \frac{1}{m}\nabla f^i(\boldsymbol{x}_k)$ with diagram $c\|\boldsymbol{v}_0\|$. Specifically, the direction $\boldsymbol{d}_k$ is yielded by

$$\boldsymbol{d}_k = \arg\min_{d\in\mathcal{R}^d} \max_{i\in[m]} \nabla f_i(\boldsymbol{x}_k)^\top \boldsymbol{d}, s.t. \|\boldsymbol{d} - \boldsymbol{v}_0\| \leq c\|\boldsymbol{v}_0\|.$$

It can minimize the average loss function, while leveraging the worst local improvement of individual tasks to regularize the algorithm trajectory by using the common descent property of MGDA. Note that the gradient conflicting issue is not absolutely eliminated due to the new constraint. Still, the above objective forces direction to approach the common descent direction as close as possible. To simplify the optimization, the dual objective is

$$\boldsymbol{\lambda} = \arg\min_{\boldsymbol{\lambda}\in S_m} \left(\sum_{i=1}^m \lambda_k^i \nabla f^i(\boldsymbol{x}_k)\right)^\top \boldsymbol{v}_0 + c\|\boldsymbol{v}_0\| \left\|\sum_{i=1}^m \lambda_k^i \nabla f^i(\boldsymbol{x}_k)\right\|.$$

Then the direction is calculated by $\boldsymbol{d}_k = -\left(\boldsymbol{v}_0 + c\|\boldsymbol{v}_0\| \left(\sum_{i=1}^m \lambda_k^i \nabla f^i(\boldsymbol{x}_k)\right)\right)$.

**Convergence analysis.** MGDA has been shown to converge to an arbitrary Pareto critical/optimal point with the same rate as single-objective optimization [9]. A similar result has been proved with PCGrad [46]. CAGrad has been shown to converge to the minimizer or stationary point of the averaging loss $\frac{1}{m}\sum_{i=1}^m f^i(\boldsymbol{x})$ when $c \in [0, 1)$, or an arbitrary Pareto critical/optimal point when $c \geq 1$ [28]. By Proposition 1, we can infer that the minimizer or stationary point of the averaging loss is also Pareto optimal or critical. Hence, we'll analyze them uniformly under the Pareto optimality in the following analysis, rather than distinguish them.

# 3 General Framework for Multi-Objective Gradient Manipulation

In this section, we summarize multi-objective gradient manipulation (MOGM) algorithms into a general framework with the goal of uniformly analyzing the above algorithms.

## 3.1 The Framework with Full-batch Gradients

Without loss of generality, we summarize the direction search of MOGM in the following framework.

$$\boldsymbol{d}_k = -\nabla \boldsymbol{F}(\boldsymbol{x}_k)\boldsymbol{\lambda}_k, \tag{2}$$

where the composite weights $\boldsymbol{\lambda}_k = (\lambda_k^1, \ldots, \lambda_k^m)^\top$ is a function of the multiple gradients $\nabla \boldsymbol{F}(\boldsymbol{x}_k) = (\nabla f^1(\boldsymbol{x}_k), \ldots, \nabla f^m(\boldsymbol{x}_k))^\top$. Specifically, MGDA and CAGrad obtain $\boldsymbol{\lambda}_k$ by their dual form of the subproblem that depends on the received gradients $\nabla \boldsymbol{F}(\boldsymbol{x}_k)$. For PCGrad, we take the two-objective case as an example. If the gradients are not conflicting, we have $\boldsymbol{\lambda}_k = (1/2, 1/2)$ as $\boldsymbol{d}_k = -\frac{1}{2}(\nabla f^1(\boldsymbol{x}_k) + \nabla f^2(\boldsymbol{x}_k))$. If they are conflicting, we can rewrite $\boldsymbol{d}_k = -\frac{1}{2}(\boldsymbol{v}_1^{\mathrm{PC}} + \boldsymbol{v}_2^{\mathrm{PC}})$ by extending $\boldsymbol{v}_1^{\mathrm{PC}}$ and $\boldsymbol{v}_2^{\mathrm{PC}}$ as the following equivalent form.

$$\boldsymbol{d}_k = -\frac{1}{2}\left(\left(1 - \frac{\nabla f^1(\boldsymbol{x}_k) \cdot \nabla f^2(\boldsymbol{x}_k)}{\|\nabla f^1(\boldsymbol{x}_k)\|_2^2}\right) \nabla f^1(\boldsymbol{x}_k) + \left(1 - \frac{\nabla f^1(\boldsymbol{x}_k) \cdot \nabla f^2(\boldsymbol{x}_k)}{\|\nabla f^2(\boldsymbol{x}_k)\|_2^2}\right) \nabla f^2(\boldsymbol{x}_k)\right).$$

Then $\boldsymbol{\lambda}_k = -(\frac{1}{2} - \frac{\nabla f^1(\boldsymbol{x}_k) \cdot \nabla f^2(\boldsymbol{x}_k)}{2\|\nabla f^1(\boldsymbol{x}_k)\|_2^2}, \frac{1}{2} - \frac{\nabla f^1(\boldsymbol{x}_k) \cdot \nabla f^2(\boldsymbol{x}_k)}{2\|\nabla f^2(\boldsymbol{x}_k)\|_2^2})$. Since PCGrad with more than three objectives should be presented as a protocol [46], we leave more details for the equivalent protocol with the general framework version in Appendix B.

Note that the composite weights $\boldsymbol{\lambda}_k$ of PCGrad and CAGrad are not constrained in the probability simplex $S_m$. Hence, we here consider a more general assumption on the boundness of $\boldsymbol{\lambda}_k$ for including various MOGM methods than bounding it in $S_m$ as MGDA.

**Assumption 1.** *For the general framework 2, there exists a finite constant $B \in \mathbb{R}$, such that $0 \leq \lambda_k^i \leq B, \sum_{i=1}^m \lambda_k^i \geq 1$ for all $k = 1, \ldots, n, i = 1, \ldots, m$.*

***Remark.*** The general framework also contains more recent approaches that are based on the fundamentals of the introduced methods, such as GradVac [42], IMTL-G [29]. Note that we focus on the optimization side for multi-objective problems, thus we provide the derivation of existing fundamental optimizers. A vertical line of works that focus on neural architecture, such as RotoGrad [16], are not included in this work.

## 3.2 The Framework with Stochastic Gradients

MOO can be solved with gradient manipulation algorithms with full gradients with theoretical guarantees [9, 46, 28]. However, training with mini-batch gradient is much more practical for deep learning applications [36], while MOGM with stochastic gradients is largely underexplored. Specifically, we only obtain noisy gradient feedback $\boldsymbol{G}_k = (\boldsymbol{g}^1(\boldsymbol{x}_k, \boldsymbol{\xi}_k^1), \ldots, \boldsymbol{g}^m(\boldsymbol{x}_k, \boldsymbol{\xi}_k^m))$ that is corrupted by stochastic noise $\boldsymbol{\xi}_k^i, i = 1, \ldots, m$. Following 2, we have its stochastic version

$$\boldsymbol{d}_k = -\boldsymbol{G}_k \boldsymbol{\lambda}_k. \tag{3}$$

Note that here $\boldsymbol{\lambda}_k$ depends on stochastic gradients, not the exact ones. We then make the following standard assumption similar to single objective setting [18, 12], which says that we have access to unbiased stochastic estimates of the gradients for all objectives.

**Assumption 2.** *Let $\{\boldsymbol{x}_k\}_{k=1}^n$ be the iterate sequence for any algorithm. For all $k = 1, \ldots, n$, we have access to $\boldsymbol{g}^1(\boldsymbol{x}_k, \boldsymbol{\xi}_k^1), \ldots, \boldsymbol{g}^m(\boldsymbol{x}_k, \boldsymbol{\xi}_k^m)$ which are unbiased estimates of $\nabla f^1(\boldsymbol{x}_k), \ldots, \nabla f^m(\boldsymbol{x}_k)$, i.e. $\mathbb{E}_{\boldsymbol{\xi}_k}[\boldsymbol{g}^i(\boldsymbol{x}_k, \boldsymbol{\xi}_k^i)] = \nabla f^i(\boldsymbol{x}_k), i = 1, \ldots, m$. Further, each gradient variance is bounded by $\mathbb{E}_{\boldsymbol{\xi}_k}[\|\boldsymbol{g}^i(\boldsymbol{x}_k, \boldsymbol{\xi}_k^i) - \nabla f^i(\boldsymbol{x}_k)\|_2^2] \leq \sigma^2, i = 1, \ldots, m$.*

To analyze MOGM in the stochastic setting, we state the following definition to determine a sequence generated by an iterative algorithm to asymptotic converge to Pareto critical and optimal.

**Definition 3** (**Convergence condition**). *(a) A sequence $\{\boldsymbol{x}_k\}_{k=1}^\infty$ asymptotically converges to Pareto critical points if $\lim_{k\to\infty} \mathbb{E}[\min_{\boldsymbol{\lambda}_k^* \in S_m} \|\nabla \boldsymbol{F}(\boldsymbol{x}_k)\boldsymbol{\lambda}_k^*\|] \to 0$.*
*(b) For convex multi-objective function $\boldsymbol{F}(\cdot)$. A sequence $\{\boldsymbol{x}_k\}_{k=1}^\infty$ asymptotically converges to Pareto optimal if $\lim_{k\to\infty} \mathbb{E}[\max_{\boldsymbol{x}_k^* \in \mathcal{K}} \min_{\boldsymbol{\lambda}_k^* \in S_m} (\boldsymbol{\lambda}_k^{*\top} \boldsymbol{F}(\boldsymbol{x}_k) - \boldsymbol{\lambda}_k^{*\top} \boldsymbol{F}(\boldsymbol{x}_k^*))] \to 0$.*

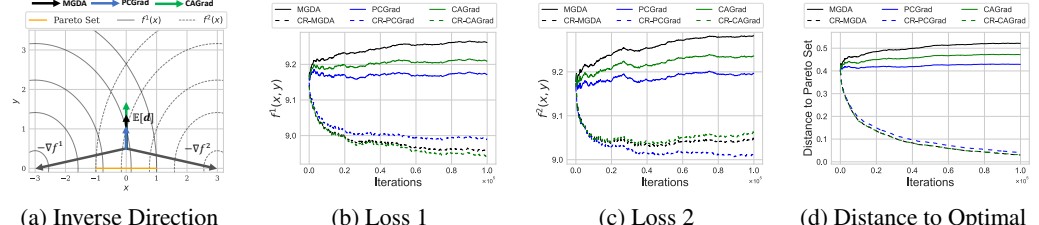

(a) Inverse Direction      (b) Loss 1      (c) Loss 2      (d) Distance to Optimal

Figure 1: (a) We plot the contours of two objectives. Gradient manipulation algorithms designed based on full-batch gradient could yield an expected composite gradient that conflicts with the true gradients and is inverse to Pareto set. (b) (c) This leads to the unexpected consequence of degrading all the objectives, making the objectives insufficiently optimized. (d) Hence, MGDA, PCGrad, and CAGrad can not converge to Pareto optimal solution, and suffer a large distance to Pareto optimal set.

The definition (a) directly follows Proposition 1 (a) and the definition of Pareto critical. The definition (b) is because $\boldsymbol{x}_k^*$ is the minimum of $\boldsymbol{\lambda}_k^{*\top} \boldsymbol{F}(\boldsymbol{x})$, thus is Pareto optimal (Proposition 1 (b)). Asymptotically converging to $\boldsymbol{\lambda}_k^{*\top} \boldsymbol{F}(\boldsymbol{x}_k^*)$ implies that $\boldsymbol{x}_k$ approaches Pareto optimal. Note that the minimization operator of $\boldsymbol{\lambda}_k^*$ is from the existence condition in Proposition 1 (b). This definition can also be viewed as the stochastic version of metric function [40] and Pareto suboptimal gap [17].

## 4 The Non-convergence of Stochastic Multi-Objective Gradient Manipulation

With the problem setup of stochastic MOGM in the previous section, we now discuss fundamental flaws among the current gradient manipulation methods that are all designed and analyzed based on full-batch gradients. We show that such batch algorithms fail to converge to a Pareto optimal solution even in a simple two-objective two-dimensional convex stochastic setting. The main issue lies in the strong dependence of $\boldsymbol{\lambda}_k$ on the received stochastic gradients $\boldsymbol{G}_k$. Surprisingly, we find that combining such correlated $\boldsymbol{\lambda}_k$ with the stochastic gradient yields a biased composite gradient $\mathbb{E}[\boldsymbol{d}_k] = -\mathbb{E}[\boldsymbol{G}_k \boldsymbol{\lambda}_k]$ that even conflicts with all full-batch gradients in expectation. This inverse direction even causes the algorithms to inversely optimize all the objectives simultaneously.

**An illustrative example.** Next, we provide a simple problem instance in which MGDA, PCGrad, CAGrad can not converge to Pareto optimal. Consider the following two-dimensional stochastic optimization setting over the domain $\mathcal{K} = \{(x,y) \mid x \in [-1,1], y \geq 0\}$, where $p$ represents the probability of occurrence

$$\tilde{f}^1(x,y) = \begin{cases} (x+1)^2 + (y-2)^2 & p = 1/2 \\ (x+5)^2 + (y+2)^2 & p = 1/2 \end{cases}, \tilde{f}^2(x,y) = \begin{cases} (x-1)^2 + (y-2)^2 & p = 1/2 \\ (x-5)^2 + (y+2)^2 & p = 1/2 \end{cases}. \quad (4)$$

The expected function is $f^1(x,y) = (x+3)^2 + y^2 + 8, f^2(x,y) = (x-3)^2 + y^2 + 8$. Thus, by ignoring the constant term, the optimization goal is to minimize the distance towards $(-3,0)$ and $(3,0)$ simultaneously. We can easily know that the Pareto set is the line segment $\{(a,0) \mid a \in [-1,1]\}$.

In the first step, by calculating the expected gradient of the point $(0, 0.4)$, we find that all the gradient manipulation algorithms generate a vertical-up direction in expectation, as shown in Figure 1a, which moves the algorithm in the wrong direction away from Pareto optimal setting. More seriously, the expected direction $\mathbb{E}[\boldsymbol{d}_k] = (0, \epsilon), \epsilon > 0$ is conflicting with both true gradients, suggesting that it inversely optimizes both objectives.

Furthermore, we could prove that the inverse direction problem happens with Pareto optimal points, which resists the algorithms to approach Pareto optimal. Using the intuition above, we show that gradient manipulation algorithms can not converge to Pareto optimal for this setting.

**Theorem 1.** *There is a stochastic convex optimization problem for which MGDA, PCGrad, CAGrad do not converge to the Pareto optimal solution.*

We relegate the proof to Appendix D. In the proof, we provide stronger support that at least in the range $\{(x,y) \mid x \in [-1,1], y \in [0, 0.05]\}$, the expected directions for the three typical algorithms all move the solution away from Pareto set. This means there exists a constant gap for gradient

manipulation algorithms to approach the Pareto optimal set. Note that this gap is a sufficient condition to verify the non-convergence, and it can be much larger, as shown in Figure 1d.

Intuitively, the convergence gap is caused by the strong correlation between the composite weights $\boldsymbol{\lambda}_k$ and the stochastic gradients $\boldsymbol{G}_k$, i.e., $\boldsymbol{\lambda}_k$ is solely determined by $\boldsymbol{G}_k$, which could produce an inverse direction in the example above. When the composite weights is independent of the stochastic gradients, the expected composite gradient $\mathbb{E}[\boldsymbol{d}_k] = -\mathbb{E}[\boldsymbol{G}_k\boldsymbol{\lambda}_k] = -\nabla F(\boldsymbol{x}_k)\boldsymbol{\lambda}_k$ will be a positive direction to approach the Pareto set. Hence, methods like linear scalarization are able to attain a point in the Pareto optimal set [11], but are deprived of the practical benefits of gradient manipulation.

We would also like to emphasize that while the example of non-convergence is carefully constructed to demonstrate the problems in the above three typical algorithms more easily, such an issue also happens with more general Gaussian noise, as shown in Appendix G.2. Hence, it is not unrealistic to suspect scenarios where such an issue can impair practical performance. It is also reasonable to assume that this problem is especially aggravated in high dimensional settings, where the variance of the gradients is more significant.

**Comparison with existing convergence results.** As introduced in Section 2.2, the optimal solution for the multi-task objective $\frac{1}{m}\sum_{i=1}^{m} f^i(\boldsymbol{x})$ must be Pareto optimal. Therefore, the above non-convergence analysis generates well with MOGM algorithms that optimize the averaging loss, such as CAGrad. This result shows that the previous convergence analyses in the full-batch setting are all invalid in the stochastic setting.

To the best of our knowledge, there is only MGDA that has convergence analysis in the stochastic setting. Liu et al. [31] establishes the convergence of MGDA with stochastic gradients, but based on two unjustified assumptions. The first is that the gradient variance asymptotically converges to 0. The second is that the subproblem for $\boldsymbol{\lambda}_k$, denoted as function $\boldsymbol{\lambda}_k(\boldsymbol{g}^1, \ldots, \boldsymbol{g}^m)$, is Lipschitz continuous with right to each gradient. The first assumption needs to gradually increase the batch size to full batch; however, the reliance on giant batch sizes is impractical for deep learning training [4]. The second one can significantly simplify the convergence analysis, but is actually not true as shown in the following proposition.

**Proposition 2.** *Subproblem $\boldsymbol{\lambda}_k(\cdot)$ is not Lipschitz continuous with right to each gradient for MGDA, i.e., there exist no finite $\beta$ such that $\|\boldsymbol{\lambda}_k(\boldsymbol{g}^1, \ldots, \boldsymbol{g}^i, \ldots, \boldsymbol{g}^m) - \boldsymbol{\lambda}_k(\boldsymbol{g}^1, \ldots, \boldsymbol{g}'^i, \ldots, \boldsymbol{g}^m)\| \leq \beta\|\boldsymbol{g}^i - \boldsymbol{g}'^i\|$ for $i = 1, \ldots, m$ and gradient $\boldsymbol{g}^1, \ldots, \boldsymbol{g}^i, \boldsymbol{g}'^i, \ldots, \boldsymbol{g}^m$.*

Similar results can be proved for PCGrad and CAGrad, and we relegate the proof to the appendix D.

## 5 Correlation-Reduced Stochastic Multi-Objective Gradient Manipulation

In this section, we develop a new principled exponential moving average mechanism on the composite weights $\boldsymbol{\lambda}_k$, and provide its theoretical benefit for fixing the convergence issue for MOGM algorithms. Our aim is to devise a new strategy with guaranteed convergence, while preserving the practical benefits of MOGM without introducing additional computation and memory overhead.

**Algorithmic insight.** To understand the design of the mechanism, recall that the main issue of the non-convergence for MOGM lies in the strong correlation between the composite weights $\boldsymbol{\lambda}_k$ and the stochastic gradients $\boldsymbol{G}_k$, which could cause the composite gradient $\mathbb{E}[\boldsymbol{d}_k] = -\mathbb{E}[\boldsymbol{G}_k\boldsymbol{\lambda}_k]$ inversely optimizing all the objectives in expectation. It is natural to consider reducing such correlation, which can be bounded by the following lemma.

**Lemma 1.** *Under Assumption 1, 2. The correlation between $\boldsymbol{\lambda}_k$ and $\boldsymbol{G}_k$ can be bounded as $\|\mathbb{E}[\boldsymbol{G}_k\boldsymbol{\lambda}_k] - \mathbb{E}[\boldsymbol{G}_k]\mathbb{E}[\boldsymbol{\lambda}_k]\|_2^2 \leq mB^2\sigma^2\mathbb{V}_{\boldsymbol{\xi}_k}(\boldsymbol{\lambda}_k)$, where $\mathbb{V}_{\boldsymbol{\xi}_k}(\boldsymbol{\lambda}_k) = \mathbb{E}_{\boldsymbol{\xi}_k}[\|\boldsymbol{\lambda}_k - \mathbb{E}_{\boldsymbol{\xi}_k}[\boldsymbol{\lambda}_k]\|_2^2]$.*

From Lemma 1, we know that the correlation can be reduced by reducing the variances of $\boldsymbol{\lambda}_k$. From this intuition, we can modify MOGM algorithms to reduce such correlation by regularizing the variation of $\boldsymbol{\lambda}_k$. Correlation-reduced MOGM (CR-MOGM) adopts a momentum mechanism with coefficient $\alpha_k$ on the update of composite weights as Line 5 in Algorithm 1, which is shown to be able to reduce variance as

**Lemma 2.** *Under Assumption 1, $\boldsymbol{\lambda}_k$ variance for Algorithm 1 is bounded $\mathbb{V}_{\boldsymbol{\xi}_k}[\boldsymbol{\lambda}_k] \leq m^2B^2(1-\alpha_k)^2$.*

This result shows that the correlation can be controlled by choosing $\alpha_k$ closed enough to 1. With a reduced correlation, the wrong direction generated by MOGM could be corrected to a positive

---

**Algorithm 1** Correlation-Reduced Stochastic Multi-objective Gradient Manipulation (CR-MOGM)

1: **Input:** Number of iterations $K$, regularization parameter $\alpha_k$, learning rate $\eta_k$.
2: **Initialize:** $x_0, \boldsymbol{\lambda}_0 \in \Delta_m$.
3: **for** $k = 1, \ldots, K$ **do**
4:     Compute $\hat{\boldsymbol{\lambda}}_k$ by gradient manipulation algorithms using the stochastic gradients.
5:     Update the weights for the gradient composition $\boldsymbol{\lambda}_k = \alpha_k \boldsymbol{\lambda}_{k-1} + (1 - \alpha_k)\hat{\boldsymbol{\lambda}}_k$.
6:     Compute the composite gradient $\boldsymbol{d}_k = -\boldsymbol{G}_k(\boldsymbol{x}_k, \xi)\boldsymbol{\lambda}_k$.
7:     Update $\boldsymbol{x}_{k+1} = \boldsymbol{x}_k + \eta_k \boldsymbol{d}_k$.
8: **end for**

---

direction, and thus the algorithm would be able to converge. Note that the momentum on $\boldsymbol{\lambda}_k$ only requires an additional negligible computation cost.

On the other hand, the proposed exponential moving average mechanism also preserves the advantage of MOGM by carefully setting the momentum parameter $\alpha_k$ to be 0 and asymptotically convergent to 1. Intuitively, CR-MOGM performs similarly to vanilla MOGM at the initial stage, and approximates like fixed linear scalarization in the late stage. Initially, the stochastic noise occupies a relatively small component of the stochastic gradient. Thus, methods like MGDA, PCGrad, CAGrad can search for good common descent directions. When the number of iterations increases and the solution approaches the Pareto optimal, the impact of the stochastic noise stands out, causing the inverse direction problem for MOGA as in the previous section. At this time, $\alpha_k$ becomes larger, maintaining relatively stable $\boldsymbol{\lambda}_k$ eliminates the strong correlation and guarantees the convergence.

**Convergence analysis.** We next provide convergence analysis for CR-MOGM 1 concluding all algorithms based on the general framework 3. Before that, we introduce basic assumptions.

**Assumption 3.** *$f^1(\boldsymbol{x}), \ldots, f^m(\boldsymbol{x})$ are all differentialable, $H$-Lipschitz and $L$-smoothness, suggesting that for all $\boldsymbol{x}, \boldsymbol{y} \in \mathcal{K}, i \in [m]$, it holds $\|\nabla f^i(\boldsymbol{y})\| \leq H$ and $\|\nabla f^i(\boldsymbol{x}) - \nabla f^i(\boldsymbol{y})\| \leq L\|\boldsymbol{x} - \boldsymbol{y}\|$.*

Lipschitz and smoothness are commonly used in complexity analysis for stochastic gradient algorithms [18, 12, 10, 4]. Based on Definition 3, we prove the following key results for CR-MOGM.

**Theorem 2** (**Convex**). *Under Assumption 1, 2, 3. For the sequence $\boldsymbol{x}_0, \boldsymbol{x}_1, \ldots, \boldsymbol{x}_n$ generated by Algorithm 1, we assume objective functions $f^1(\boldsymbol{x}), \ldots, f^m(\boldsymbol{x})$ are all convex with bounded optimal values as $f^1(\boldsymbol{x}^*), \ldots, f^m(\boldsymbol{x}^*) \leq F$ for $\boldsymbol{x}^*$ in Pareto set, and the distance from sequence to Pareto set is bounded, i.e., $\|\boldsymbol{x}_k - \boldsymbol{x}^*\| \leq D$. Set $0 \leq \eta_n \leq \ldots \leq \eta_1 \leq 1/mLB, \alpha_k \in (0, 1]$ for $k = 1, \ldots, n$ in Algorithm 1, it achieves*

$$\frac{1}{n}\sum_{k=1}^n \mathbb{E}\left[\max_{\boldsymbol{x}_k^* \in \mathcal{K}} \min_{\boldsymbol{\lambda}_k^* \in S_m} (\boldsymbol{\lambda}_k^{*\top}\boldsymbol{F}(\boldsymbol{x}_k) - \boldsymbol{\lambda}_k^{*\top}\boldsymbol{F}(\boldsymbol{x}_k^*))\right] \leq \frac{D^2}{\eta_n n} + \frac{m^2 B^2(\sigma^2 + \sigma H + H^2)}{n}\sum_{k=1}^n \eta_k$$

$$+ \frac{DB\sigma m^{3/2}}{n}\sum_{k=1}^n (1 - \alpha_k) + \frac{6m^{5/2}B^2 H\sigma}{n}\sum_{k=1}^n (1 - \alpha_k)\eta_k + \frac{2F}{n}\sum_{l=1}^n l(1 - \alpha_{l+1})\sum_{i=1}^m |\lambda_l^i - \hat{\lambda}_{l+1}^i|.$$

**Theorem 3** (**Non-convex**). *Under Assumption 1, 2, 3. Under Assumption 1, 2, 3. For the sequence $\boldsymbol{x}_0, \boldsymbol{x}_1, \ldots, \boldsymbol{x}_n$ generated by Algorithm 1, we assume objective functions $f^1(\boldsymbol{x}_k), \ldots, f^m(\boldsymbol{x}_k) \leq F$ are all bounded for $k = 1, \ldots, n$. Set $0 \leq \eta_n \leq \ldots \leq \eta_1 \leq 1/mLB$ and $\alpha_k \in (0, 1]$ in Algorithm 1, it achieves*

$$\frac{1}{n}\sum_{k=1}^n \mathbb{E}\left[\min_{\boldsymbol{\lambda}_k^* \in S_m} \|\nabla \boldsymbol{F}(\boldsymbol{x}_k)\boldsymbol{\lambda}_k^*\|^2\right] \leq \frac{F}{n}\sum_{k=2}^n \frac{2}{\eta_k}(1 - \alpha_k)\sum_{i=1}^m |\hat{\lambda}_k^i - \lambda_{k-1}^i| + \frac{2mBF}{n\eta_n}$$

$$+ \frac{4m^{5/2}B^2 H\sigma}{n}\sum_{k=1}^n (1 - \alpha_k) + \frac{Lm^3 B^3\sigma^2}{2n}\sum_{k=1}^n \eta_k.$$

***Remark.*** Note that our theoretical results imply that CR-MOGM is convergent in averaging by choosing suitable $\eta$ and $\alpha$. Such an averaging analysis scheme is adopted by CAGrad [28], and is often used in stochastic optimization [39, 13, 4], e.g. Adagrad [26], Adam [22, 36]. It can be transformed into the traditional version, as shown in Appendix E.7

The above results falls as an immediate corollary of the above Theorem.

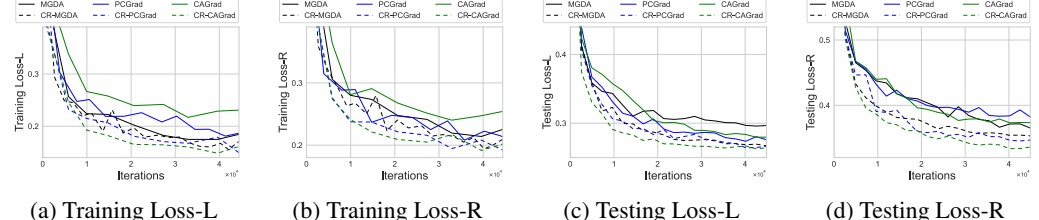

| (a) Training Loss-L | (b) Training Loss-R | (c) Testing Loss-L | (d) Testing Loss-R |

Figure 2: Performance comparison of gradient manipulation algorithms and their correlation-reduced counter part for MultiMNIST. We report the mean over 3 runs.

Table 1: Accuracy on MultiMNIST. We report the means over 3 runs.

| Method | task-L Acc | task-R Acc |
|---|---|---|
| MGDA | 96.46% | 95.29% |
| CR-MGDA | **96.92**% ↑ | **95.44**% ↑ |
| PCGrad | 96.63% | 95.37% |
| CR-PCGrad | **96.90**% ↑ | **95.75**% ↑ |
| CAGrad | 96.47% | 95.20% |
| CR-CAGrad | **96.85**% ↑ | **95.70**% ↑ |

Table 2: Losses of the final output of MGDA and CR-MGDA on MultiMNIST. We report the mean over 3 runs.

| | MGDA | | CR-MGDA | |
|---|---|---|---|---|
| Batch Size | loss-L | loss-R | loss-L | loss-R |
| 8 | 0.44 | 0.54 | **0.24** | **0.35** |
| 4 | 0.57 | 0.65 | **0.26** | **0.36** |
| 2 | 0.71 | 0.81 | **0.28** | **0.40** |

**Corollary 1.** *Set* $1 \geq \eta_k = O(1/\sqrt{k}), \alpha_k = \max\{1 - \eta_k/\eta_1, 1 - \eta_k/\eta_1(k \sum_{i=1}^m |\lambda_{k-1}^i - \hat{\lambda}_k^i|)\}$ *in Theorem 2, 3. Algorithm 1 converges with* $O(n^{1/2})$, $O(n^{1/4})$ *rates in the convex and non-convex case.*

This result indicates that Algorithm 1 has a convergence rate that matches the one in the single-objective stochastic setting [5, 2]. It demonstrates the efficacy of the proposed momentum mechanism.

## 6 Experiments

In this section, we present empirical results on both synthetic with convex objectives and real-world datasets for multi-task learning with deep neural networks, representing convex and nonconvex settings, respectively. We apply CR-MOGM to MGDA, PCGrad, CAGrad, and use CR- to distinguish whether it adopts our approach. We conduct experiments to answer the following questions:

**Q1:** Does our approach fix the non-convergence and improve the optimality for MOGM algorithms?

**Q2:** Does our approach consistently improve the practical performance of MOGM in deep learning?

**Q3:** Does the non-convergence issue demonstrated with the special case still exists in practice?

**Q4:** Is our approach flexible to be implemented and agnostic to models as well as optimizers?

**Synthetic experiments.** To demonstrate whether CR-MOGM fixes the convergence issue for gradient manipulation algorithm, we first consider the same simple convex setting as the example 4 in Section 4. We initialize $\boldsymbol{x}_0 = (0, 0.4)$ and use SGD as the optimizer. To enable fair comparison, the adaptive stepsize is set with $\eta_k = 0.006/\sqrt{k}$ for all the algorithms, and the momentum parameter for smoothing the composite weights is set with the suggested parameter setting in Corollary 1.

Figure 1 illustrates the multi-objective losses and distance to the Pareto set for this problem. We first note that both the two objective losses of MGDA, PCGrad, CAGrad are significantly larger than their counterparts applied with the proposed mechanism. Furthermore, the losses are even increasing at the initial stage as a result of the inverse direction issue demonstrated in Figure 1a. It moves $\boldsymbol{x}_k$ away from the Pareto set, which unfortunately causes an increasing suboptimal gap. In contrast, applying CR-MOGM enhances the algorithms to converge to Pareto optimal. This positively answers **Q1**.

**Multi-task classification.** For multi-task classification scenarios, we follow the experiment setup from Sener et al. [38], and consider the MultiMNIST dataset [37] with 60K examples. MultiMNIST contains two tasks: classifying the digit on the top-left (task-L) and classifying the digit on the bottom-

| Method | Segmentation (Higher Better) | | | | Depth (Lower Better) | | | |
|---|---|---|---|---|---|---|---|---|
| | (Training) | | (Testing) | | (Training) | | (Testing) | |
| | mIoU | Pix Acc | mIoU | Pix Acc | Abs Err | Rel Err | Abs Err | Rel Err |
| MGDA [38] | 76.05 | 94.08 | 66.32 | 88.85 | **0.0065** | 18.90 | 0.0248 | **33.50** |
| CR-MGDA | **79.58** | **95.12** | **73.21** | **93.01** | 0.0067 | **18.51** | **0.0151** | 33.59 |
| PCGrad [46] | **80.19** | 95.31 | 75.25 | **93.50** | 0.0087 | 27.18 | **0.0148** | 45.37 |
| CR-PCGrad | 80.12 | **95.32** | **75.37** | **93.50** | **0.0085** | **26.44** | 0.0153 | **43.05** |
| CAGrad [28] | 80.24 | 95.29 | 75.09 | 93.52 | 0.0081 | **25.06** | 0.0145 | 40.97 |
| CR-CAGrad | **80.33** | **95.33** | **75.58** | **93.53** | **0.0080** | 25.30 | **0.0142** | **38.51** |

Table 3: Multi-task learning results on CityScapes Challenge. We report the mean over 3 runs.

right (task-R). We use the LeNet architecture [23], and use SGD as the optimizer. The stepsize and parameters are tuned from vanilla MGDA, PCGrad, CAGrad, and the correlation-reduced counterparts follow the same parameter setting to enable a fair comparison. The momentum parameter is set with suggested value in Corollary 1.

We report the train and test losses in Figure 2. We can see that algorithms with $\lambda$ momentum perform better than original ones with respect to both train and test losses. We also observed that $\lambda$ momentum consistently improves the accuracy for all the objectives in Table 1. We hence positively answer **Q2**.

In particular in Table 2, we observe that the loss of vanilla MGDA becomes significantly larger when the batch size is smaller, while CR-MGDA appears not sensitive to the batch size. The possible reason is that the convergence issue of MOGM becomes more severe with larger stochastic noise, which severely degrades the performance of MOGM. Since CR-MOGM is proven to converge to the Pareto optimal/critical, it is much more robust to stochastic noises. Thus, in this regard, our method is more practical for real-world applications. We hence positively answer **Q3**.

**Semantic segmentation.** For semantic segmentation scenarios, we follow the experiment setup from Yu et al. [46] and Liu et al. [28], and consider the CityScapes. CityScapes similarly contains two tasks: 7-class semantic segmentation and depth estimation. We combine algorithms with MTAN [30] based on SegNet architecture [1]. We use adam [22] as the optimizer. All parameters are tuned from vanilla MOGM algorithms, and the correlation-reduced counterparts follow the same parameter setting to enable a fair comparison. The momentum parameter is set with the suggested value.

We report mean Intersection-Over-Union (mIoU), pixel accuracy (Pix Acc), and absolute/relative (abs/rel) errors in Table 3. We observe comparable or better performance when adopting our approach with the three gradient manipulation methods. In particular, the improvements for MGDA and CAGrad are more significant than PCGrad, possibly due to the special gradient manipulation rule that PCGrad does not alter the gradient when they are not conflicting, then the non-convergence issue of PCGrad could be less severe. Combining the above results, we demonstrate that our methods work for different models and optimizers. We hence give a positive answer to **Q4**.

# 7 Conclusions

In this work, we study multi-objective gradient manipulation in the stochastic mini-batch setting and identify a fundamental flaw in algorithms designed under the exact full-batch setting. Through an illustrative example, we rigorously prove the non-convergence issue of typical methods, including MGDA, PCGrad, and CAGrad. In particular, we fix this problem by slightly changing these algorithms by an exponential smoothing mechanism without introducing additional computational or memory overhead. We prove the convergence of the resulting algorithms and show that they yield consistent empirical performance improvement. Our analysis is general and can be extended to analyze combining MOGM algorithms with other stochastic optimizers such as Adam and Adagrad.

**Limitations.** This paper discusses the typical MOGM algorithms with strict convergence guarantees, i.e., MGDA, PCGrad, and CAGrad. Other empirical-driven methods without rigorous theoretical guarantees, such as RotoGrad [16], are too expensive to be included in our analysis. We believe that the convergence issue in that line of literature will be an excellent future work to explore in depth.

## Acknowledgement

This work is supported by the National Key Research and Development Program of China No. 2020AAA0106300 and National Natural Science Foundation of China No. 62250008. This work was supported by Ant Group through Ant Research Intern Program. We would like to thank Tianyu Liu, Xin Wang, and Ziwei Zhang for their advices on paper writing and Guannan Zhang for generous support on this project.

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
