# Supplementary Materials for "On the Convergence of Stochastic Multi-Objective Gradient Manipulation and Beyond"

We organize the supplemental materials as follows. Section A provides a detailed comparison between weak Pareto optimal and Pareto optimal. Section B provides the derivation of PCGrad with more than two objectives to our general framework. Section C discusses more details of the convergence analysis framework in Definition 3. Section D supplements the theoretical results on the non-convergence for MOGM algorithms. Section E supplements the convergence results for both convex (Theorem 2) and non-convex (Theorem 3) cases. In particular, we discuss the averaging scheme of the convergence theorem in more detail, and provide a way to transform it to traditional convergence using a typical uniformly sampling technique in stochastic convergence analysis [4, 24, 5, 20]. Section F provides the convergence analysis for the cases with strongly convex functions. Section G introduces additional details of the experiments, and provides an additional simulation result with Gaussian noises.

## A Detailed Comparison between Weak Pareto Optimal and Pareto Optimal

In this section, we give a rigorous comparison between the definitions of Pareto optimal and weak Pareto optimal, where the latter is defined as

**Definition 4** (Weak Pareto optimal). *A solution $\boldsymbol{x}^* \in \mathcal{K}$ is called weak Pareto optimal if there is not $x \in \mathcal{K}$ such that $F(\boldsymbol{x}) < F(\boldsymbol{x}^*)$, i.e. $f^i(\boldsymbol{x}) \leq f^i(\boldsymbol{x}^*)$ for all $i = 1, \ldots, m$.*

By the Definition 1 and Definition 4, it can be easy to know that Pareto optimal points are always weak Pareto optimal, and the converse is not always true. Take a two-objectives example, the minimal points for $f^1$ are all Pareto weak optimal, but only ones that are also optimal for $f^2$ are Pareto optimal. Furthermore, if we assume $f^1$ has a unique minimizer, which is then Pareto optimal.

In rigorous words, we state below that relation among Pareto optimal and weak Pareto optimal, as a rigorous version for Proposition 1 (b).

**Proposition 3** (Theorem 5.13 and Lemma 5,14 in [15]). *For MOO problem 1, the following statements hold: (a) For convex objectives $\boldsymbol{F}(\cdot)$. If there exists $\boldsymbol{\lambda} \in S_m$ such that $\boldsymbol{x}^* = \arg\min_{\boldsymbol{x} \in \mathcal{K}} \boldsymbol{\lambda}^T \boldsymbol{F}(\boldsymbol{x})$, then $\boldsymbol{x}^*$ is weak Pareto optimal. (b) Further if $\boldsymbol{x}^*$ is the unique minimizer, then $\boldsymbol{x}^*$ Pareto optimal.*

For ease to understand, previous studies [9, 31, 28] do not distinguish the detailed difference between the above definitions. This paper also follows this presentation trick.

## B More Details of the General Framework for MOGM

**PCGrad with more than 2 objectives.** Specifically, the PCGrad is initialized by $\boldsymbol{v}_i^{\mathrm{PC}} = \nabla f^i(\boldsymbol{x}_k)$. When subtracting the conflicting component, i.e., $\boldsymbol{v}_i^{\mathrm{PC}} = \boldsymbol{v}_i^{\mathrm{PC}} - \frac{\boldsymbol{v}_i^{\mathrm{PC}\top} \nabla f^j(\boldsymbol{x}_k)}{\|\nabla f^j(\boldsymbol{x}_k)\|^2} \nabla f^j(\boldsymbol{x}_k)$, it is equivalent to subtracting a weight $\frac{\boldsymbol{v}_i^{\mathrm{PC}\top} \nabla f^j(\boldsymbol{x}_k)}{\|\nabla f^j(\boldsymbol{x}_k)\|^2} \nabla f^j(\boldsymbol{x}_k)$ for gradient $\boldsymbol{v}_j$ if using the framework 2. Provided with such equivalence, $\boldsymbol{d}_k = -\frac{1}{m} \sum_{i=1}^m \boldsymbol{v}_i^{\mathrm{PC}}$ is equivalent to $\boldsymbol{d}_k = -\nabla \boldsymbol{F}(\boldsymbol{x}_k) \boldsymbol{\lambda}_k$ using Algorithm 2

**Algorithm 2** Calculation for composite weight $\boldsymbol{\lambda}_k$ in PCGrad

1: **Input:** $\nabla f^i(\boldsymbol{x}_k), i = 1, \ldots, m$
2: **Initialize:** $v_i = \nabla f^i(\boldsymbol{x}_k), v_i^{\text{PC}} = v_i, \lambda^i = 1$ for $i = 1, \ldots, m$
3: **for** $i \in [m]$ **do**
4:     **for** $j \overset{\text{uniformly}}{\sim} [m] \setminus i$ in random order **do**
5:         **if** $v_i^{\text{PC}\top} \nabla f^j(\boldsymbol{x}_k) < 0$ **then**
6:             Set $v_i^{\text{PC}} = v_i^{\text{PC}} - \frac{v_i^{\text{PC}\top} \nabla f^j(\boldsymbol{x}_k)}{\|\nabla f^j(\boldsymbol{x}_k)\|^2} \nabla f^j(\boldsymbol{x}_k)$
7:             Update $\lambda^j = \lambda^j - \frac{v_i^{\text{PC}\top} \nabla f^j(\boldsymbol{x}_k)}{\|\nabla f^j(\boldsymbol{x}_k)\|^2}$
8:         **end if**
9:     **end for**
10: **end for**
11: **return** $\boldsymbol{\lambda}$

## C   More Details of the Convergence Analysis Framework

In this section, we provide more details about our stochastic convergence analysis framework. We first offer proof details of the Proportion 1, and then generalize previous analysis based on it. Finally, we give a discussion for Definition 3 that is the stochastic convergence framework based on the previous full-batch setting.

### C.1   Proof of Proportion 1

*Proof.* The Pareto criticality can be equivalently defined as: a $\boldsymbol{x}^* \in \mathcal{K}$ is Pareto critical, if $\min_{\boldsymbol{\lambda}^* \in S_m} \| \sum_{i=1}^m \lambda^{*i} \nabla f^i(\boldsymbol{x}^*) \| = 0$, where $S_m = \{(\lambda^1, \ldots, \lambda^m) | \sum_{i=1}^m \lambda^i = 1, \lambda \in [0, 1]\}$ denotes the probability simplex. Therefore, if there exist a $\boldsymbol{\lambda} \in S_m$, such that $0 \leq \| \sum_{i=1}^m \lambda^i \nabla f^i(\boldsymbol{x}^*) \| = 0$, then $\min_{\boldsymbol{\lambda}^* \in S_m} \| \sum_{i=1}^m \lambda^{*i} \nabla f^i(\boldsymbol{x}^*) \| \leq \| \sum_{i=1}^m \lambda^i \nabla f^i(\boldsymbol{x}^*) \| = 0$. We thus prove Proposition **(a)**, and Proposition **(b)** directly comes from Proposition 3. $\qquad\square$

### C.2   More Details of Previous Convergence Analysis in Full-batch Gradient Setting

Previous convergence analyses in full-batch gradient setting, including MGDA [9, 40], PCGrad [46], CAGrad [28] can be analyzed based on Proposition 1 that reflects Pareto optimality and criticality, which induces the following convergence analysis framework.

**Definition 5.** *(a) A sequence $\{\boldsymbol{x}_k\}_{k=1}^\infty$ asymptotically converges to Pareto critical points if $\lim_{k \to \infty} \min_{\boldsymbol{\lambda}_k^* \in S_m} \|\nabla \boldsymbol{F}(\boldsymbol{x}_k) \boldsymbol{\lambda}_k^*\| \to 0$.*
*(b) For convex multi-objective function $\boldsymbol{F}(\cdot)$. A sequence $\{\boldsymbol{x}_k\}_{k=1}^\infty$ asymptotically converges to Pareto optimal if $\lim_{k \to \infty} \max_{\boldsymbol{x}_k^* \in \mathcal{K}} \min_{\boldsymbol{\lambda}_k^* \in S_m} (\boldsymbol{\lambda}_k^{*\top} \boldsymbol{F}(\boldsymbol{x}_k) - \boldsymbol{\lambda}_k^{*\top} \boldsymbol{F}(\boldsymbol{x}_k^*)) \to 0$.*

**(a)** can be reviewed as the limit for attaining the criticality condition (the equivalent definition in [28]). **(b)** is equivalent to that there exists a composite weight $\boldsymbol{\lambda}_k^* \in S_m$ such that $\boldsymbol{\lambda}_k^{*\top} \boldsymbol{F}(\boldsymbol{x}_k) - \min_{\boldsymbol{x}_k^* \in \mathcal{K}} \boldsymbol{\lambda}_k^{*\top} \boldsymbol{F}(\boldsymbol{x}_k^*) \to 0$, which is essentially the limit for attaining Pareto optimal condition in Proposition 1. It can be also reviewed the the convergence for multiobjective metric function [40].

We next briefly introduce the analysis of MGDA [9, 40] as an typical example that provides both nonconvex and convex convergence. It proves that the norm of min-norm direction $\min_{\boldsymbol{\lambda}_k^* \in S_m} \|\nabla \boldsymbol{F}(\boldsymbol{x}_k) \boldsymbol{\lambda}_k^*\|$ is asymptotically convergent to zero with an order of $O(1/\sqrt{n})$, which is reviewed as the convergence rate with non-convex functions.

**Theorem 4** (Theorem 3.1 in [9]). *For MGDA algorithm updating with stepsize $\eta \leq 1/L$*

$$\frac{1}{n} \sum_{k=1}^n \min_{\boldsymbol{\lambda}_k^* \in S_m} \|\nabla \boldsymbol{F}(\boldsymbol{x}_k) \boldsymbol{\lambda}_k^*\| \leq O(1/\sqrt{n})$$

This result also implies that the composite gradient $\boldsymbol{d}_k = \|\nabla \boldsymbol{F}(\boldsymbol{x}_k) \boldsymbol{\lambda}_k^*\|$ for MGDA asymptotically vanishes, which indicates that the sequence $\boldsymbol{x}_k$ iterated by gradient descent using $\boldsymbol{d}_k$ limits to a critical point $\boldsymbol{x}^*$.

If the multi-objective function is convex for each objective, such a critical point $x^*$ is also weak Pareto optimal [41]. It is also not difficult to verify that if $x^*$ is a Pareto stationary point, then it satisfies $\nabla F(x^*)\lambda^* = 0$ for one $\lambda^* \in S_m$. If the functions are convex, this meets the optimal condition of the composite function $\lambda^{*\top} F(x^*)$. And by Proposition 1, we then know that $x$ is weak Pareto optimal.

It can prove that $\sum_{k=1}^n \lambda_k^{*\top} F(x_k) - \sum_{k=1}^n \lambda_k^{*\top} F(x^*) \le O(1)$. Since in the batch-gradient setting, $d_k$ is the common descent direction such that $F(x_k)$ is decreasing with respect to $k$. Thus it has $\sum_{k=1}^n \lambda_k^{*\top} F(x_n) - \sum_{k=1}^n \lambda_k^{*\top} F(x^*) \le \sum_{k=1}^n \lambda_k^{*\top} F(x_k) - \sum_{k=1}^n \lambda_k^{*\top} F(x^*) \le O(1)$. Therefore, we have the following theorem

**Theorem 5** (Theorem 4.1 in [9]). *For MGDA algorithm updating with stepsize $\eta \le 1/L$*

$$\left(\frac{\sum_{k=1}^n \lambda_k^*}{n}\right)^\top F(x_n) - \left(\frac{\sum_{k=1}^n \lambda_k^*}{n}\right)^\top F(x^*) \le O(1/k).$$

Where the averaging weight $\frac{\lambda_k^*}{n}$ is also in $S_m$. Since in the proof, the comparing point $x^*$ can be generated to arbitrarily chosen $u \in \mathcal{K}$ [41]. We can further generate the result with $\left(\frac{\sum_{k=1}^n \lambda_k^*}{n}\right)^\top F(x_n) - \left(\frac{\sum_{k=1}^n \lambda_k^*}{n}\right)^\top F(u) \le O(1/k)$. By letting $u = \arg\min_{x \in \mathcal{K}} \left(\frac{\sum_{k=1}^n \lambda_k^*}{n}\right)^\top F(x)$, it finally has

$$\max_{x_k^* \in \mathcal{K}} \min_{\lambda_k^* \in S_m} (\lambda_k^{*\top} F(x_k) - \lambda_k^{*\top} F(x_k^*))$$

$$\le \left(\frac{\sum_{k=1}^n \lambda_k^*}{n}\right)^\top F(x_n) - \left(\frac{\sum_{k=1}^n \lambda_k^*}{n}\right)^\top F(x^*) \le O(1/k).$$

Similarly, PCGrad [46] is proved to converge at a Pareto critical point (Theorem 1 in [46]). CA-Grad [28] generalizes the analysis of MGDA, and shows that it converges to the Pareto critical point that tends to be the critical point of the average loss when $c \in [0, 1)$. These convergence results can all be analyzed under the Pareto convergence framework as Definition 5.

### C.3 More Discussion for Definition 3

**Definition 3.** *(a) A sequence $\{x_k\}_{k=1}^\infty$ asymptotic converges to Pareto critical points if $\lim_{k\to\infty} \mathbb{E}[\min_{\lambda_k^* \in S_m} \|\nabla F(x_k)\lambda_k^*\|] \to 0$.*
*(b) For convex multi-objective function $F(\cdot)$. A sequence $\{x_k\}_{k=1}^\infty$ asymptotic converges to Pareto optimal if $\lim_{k\to\infty} \mathbb{E}[\max_{x_k^* \in \mathcal{K}} \min_{\lambda_k^* \in S_m} (\lambda_k^{*\top} F(x_k) - \lambda_k^{*\top} F(x_k^*))] \to 0$.*

Definition 3 is a stochastic version for Definition 5. Similarly, **(a)** can be reviewed as the expected limit for attaining the criticality condition. **(b)** is equivalent to that there exists a composite weight $\lambda_k^* \in S_m$ such that $\mathbb{E}[\lambda_k^{*\top} F(x_k) - \min_{x_k^* \in \mathcal{K}} \lambda_k^{*\top} F(x_k^*)] \to 0$, which means $\lambda_k^{*\top} \mathbb{E}[F(x_k)] \to \min_{x_k^* \in \mathcal{K}} \lambda_k^{*\top} F(x_k^*)$, suggesting that the expected function value $\mathbb{E}[F(x_k)]$ are optimal. Thus, Definition 3 is indeed derived from Pareto optimality and criticality.

## D  Missing Proofs for the Non-convergence of MOGM

In this section, we give detailed proof for the non-convergence for MGDA, PCGrad, CAGrad with the example introduced in Section 4. We then prove the non-Lipschitz of the composite weight for the above MOGM algorithms.

### D.1  Proof of Theorem 1

*Proof.* Recall the two-dimensional stochastic optimization setting over the domain $\mathcal{K} = \{(x,y) \mid x \in [-1, 1], y \ge 0\}$

$$\tilde{f}^1(x,y) = \begin{cases} (x+1)^2 + (y-2)^2 & p = 1/2 \\ (x+5)^2 + (y+2)^2 & p = 1/2 \end{cases}, \tilde{f}^2(x,y) = \begin{cases} (x-1)^2 + (y-2)^2 & p = 1/2 \\ (x-5)^2 + (y+2)^2 & p = 1/2 \end{cases}.$$

The expected function is $f^1(x, y) = (x + 3)^2 + y^2 + 8$, $f^2(x, y) = (x - 3)^2 + y^2 + 8$. Therefore, the optimization goal is to minimize the distance towards $(-3, 0)$ and $(3, 0)$ simultaneously. We can easily know that the Pareto set is the line segment $\{(a, 0) \mid a \in [-1, 1]\}$.

We can calculate the stochastic gradient in the above setting as

$$\nabla \tilde{f}^1(x, y) = \begin{cases} 2(x + 1, y - 2) & p = 1/2 \\ 2(x + 5, y + 2) & p = 1/2 \end{cases}, \nabla \tilde{f}^2(x, y) = \begin{cases} 2(x - 1, y - 2) & p = 1/2 \\ 2(x - 5, y + 2) & p = 1/2 \end{cases}.$$

Next, we prove that for all point in domain $\{(a, b) | a \in [-1, 1], b \in [0, 0.05]\}$, the expected composite directions $\mathbb{E}[\boldsymbol{d}_k] = (d_k^1, d_k^2)$ of MGDA, PCGrad, CAGrad have positive value in y-axis, i.e. $d_k^2 > 0$, which forces the algorithms to move away from the Pareto set.

**Lemma 3** (Sener et al. [38]). *The min-norm solver has closed form solution for two-objective case. Specifically, for $\boldsymbol{g}^1, \boldsymbol{g}^2$, the composite weight $\lambda$ for $\boldsymbol{g}^1$ can be calculated as*

$$\underset{\lambda \in [0,1]}{\arg\min} \|\lambda \boldsymbol{g}^1 + (1 - \lambda)\boldsymbol{g}^2\| = \min \left\{ \left[ \frac{(\boldsymbol{g}^2 - \boldsymbol{g}^1)^\top \boldsymbol{g}^2}{\|\boldsymbol{g}^2 - \boldsymbol{g}^1\|_2^2} \right]_+, 1 \right\}, \tag{5}$$

*where the operator $[a]_+ = \max\{a, 0\}$.*

**Lemma 4.** *For any point in domain $\{(a, b) | a \in [-1, 1], b \in [0, 0.05]\}$, MGDA has a expected composite directions $\mathbb{E}[\boldsymbol{d}_k] = (d_k^1, d_k^2)$ such that $d_k^2 > 0$.*

*Proof.* We adopt simple enumeration method to calculate the expected direction.

**Case 1:** when the received gradients are

$$\nabla \tilde{f}^1(a, b) = 2(a + 1, b - 2), \quad \nabla \tilde{f}^2(a, b) = 2(a - 1, b - 2).$$

From 5, we can get $\lambda = (1 - a)/2$, then the direction for $(a, b)$ is

$$\boldsymbol{d}_k = -2(0, b - 2).$$

**Case 2:** when the received gradients are

$$\nabla \tilde{f}^1(a, b) = 2(a + 5, b + 2), \quad \nabla \tilde{f}^2(a, b) = 2(a - 5, b + 2).$$

From 5, we can get $\lambda = (5 - a)/10$, then the direction for $(a, b)$ is

$$\boldsymbol{d}_k = -2(0, b + 2).$$

**Case 3:** when the received gradients are

$$\nabla \tilde{f}^1(a, b) = 2(a + 1, b - 2), \quad \nabla \tilde{f}^2(a, b) = 2(a - 5, b + 2).$$

From 5, we can get $\lambda = (-3a + 2b + 19)/26$, then the direction for $(a, b)$ is

$$\boldsymbol{d}_k = -2(\frac{4a + 6b - 8}{13}, \frac{6a + 9b - 12}{13}).$$

**Case 4:** when the received gradients are

$$\nabla \tilde{f}^1(a, b) = -2(a + 5, b + 2), \quad \nabla \tilde{f}^2(a, b) = 2(a - 1, b - 2).$$

From 5, we can get $\lambda = (-3a - 2b + 7)/26$, then the direction for $(a, b)$ is

$$\boldsymbol{d}_k = -2(\frac{4a - 6b + 8}{13}, \frac{-6a + 9b - 12}{13}).$$

By summing up the above cases and multiply the probability, we have

$$\mathbb{E}[\boldsymbol{d}_k] = (-\frac{8}{13a}, \frac{24 - 44b}{13}).$$

Since we know that $b \leq 0.05$, we then know $d_k^2 \geq \frac{12 - 22*0.05}{13} = \frac{10.9}{13} > 0$. We thus prove the Lemma. $\square$

**Lemma 5.** *For the point in domain* $\{(a, b)|a \in [-1, 1], b \in [0, 0.05]\}$, *PCGrad has a expected composite directions* $\mathbb{E}[\boldsymbol{d}_k] = (d_k^1, d_k^2)$ *such that* $d_k^2 > 0$.

*Proof.* Similar with the proof of MGDA, we also adopt simple enumeration method to calculate the expected direction.

**Case 1:** when the received gradients are

$$\nabla \tilde{f}^1(a, b) = 2(a + 1, b - 2), \quad \nabla \tilde{f}^2(a, b) = 2(a - 1, b - 2).$$

It is easy to verify that for $\{(a, b)|a \in [-1, 1], b \in [0, 0.05]\}$, the two gradients has not conflicting, then we know the y-axis value of the composite gradient for $(a, b)$ is $4 - 2b$.

**Case 2:** when the received gradients are

$$\nabla \tilde{f}^1(a, b) = 2(a + 5, b + 2), \quad \nabla \tilde{f}^2(a, b) = 2(a - 5, b + 2).$$

It is easy to verify that for $\{(a, b)|a \in [-1, 1], b \in [0, 0.05]\}$, the two gradients consistently conflict with each other, then we know the y-axis value of the composite gradient for $(a, b)$ is

$$-4(b + 2) + 8(a^2 + (b + 2)^2 - 25)(\frac{b + 2}{(a + 5)^2 + (b + 2)^2} - \frac{b + 2}{(a - 5)^2 + (b + 2)^2}).$$

**Case 3:** when the received gradients are

$$\nabla \tilde{f}^1(a, b) = 2(a + 1, b - 2), \quad \nabla \tilde{f}^2(a, b) = 2(a - 5, b + 2).$$

It is easy to verify that for $\{(a, b)|a \in [-1, 1], b \in [0, 0.05]\}$, the two gradients consistently conflict with each other, then we know the y-axis value of the composite gradient for $(a, b)$ is

$$-4b + 8(a^2 - 4a + b^2 - 9)(\frac{b - 2}{(a + 1)^2 + (b - 2)^2} - \frac{b + 2}{(a - 5)^2 + (b + 2)^2}).$$

**Case 4:** when the received gradients are

$$\nabla \tilde{f}^1(a, b) = 2(a + 5, b + 2), \quad \nabla \tilde{f}^2(a, b) = 2(a - 1, b - 2).$$

It is easy to verify that for $\{(a, b)|a \in [-1, 1], b \in [0, 0.05]\}$, the two gradients consistently conflict with each other, then we know the y-axis value of the composite gradient for $(a, b)$ is

$$-4b + 8(a^2 + 4a + b^2 - 9)(\frac{b + 2}{(a + 5)^2 + (b + 2)^2} - \frac{b - 2}{(a - 1)^2 + (b - 2)^2}).$$

Sum up the above, it is easy to know $d_k^2$ is a increasing function for $b$, and is creasing for $a \in [-1, 0]$ and decreasing for $a \in [0, 1]$. Plug in $a = \pm 1, b = 0.05$, we can get $d_k^2 \geq 0.2 > 0$. □

**Lemma 6.** *For the point in domain* $\{(a, b)|a \in [-1, 1], b \in [0, 0.05]\}$, *CAGrad with* $c \in [0.8, 1]$ *has a expected composite directions* $\mathbb{E}[\boldsymbol{d}_k] = (d_k^1, d_k^2)$ *such that* $d_k^2 > 0$.

*Proof.* Similar with the proof of MGDA, we also adopt simple enumeration method to calculate the expected direction.

**Case 1:** when the received gradients are

$$\nabla \tilde{f}^1(a, b) = 2(a + 1, b - 2), \quad \nabla \tilde{f}^2(a, b) = 2(a - 1, b - 2).$$

It is easy to verify that for $\{(a, b)|a \in [-1, 1], b \in [0, 0.05]\}$, the two gradients has not conflicting, then we know the y-axis value of the composite gradient for $(a, b)$ is $-2b + 4 + 2c\sqrt{a^2 + (b - 2)^2} \geq -2 * 0.05 + 4 + 2c\sqrt{0^2 + (0.05 - 2)^2} = 3.8c + 3.9 \geq 6.94$.

**Case 2:** when the received gradients are

$$\nabla \tilde{f}^1(a, b) = 2(a + 5, b + 2), \quad \nabla \tilde{f}^2(a, b) = 2(a - 5, b + 2).$$

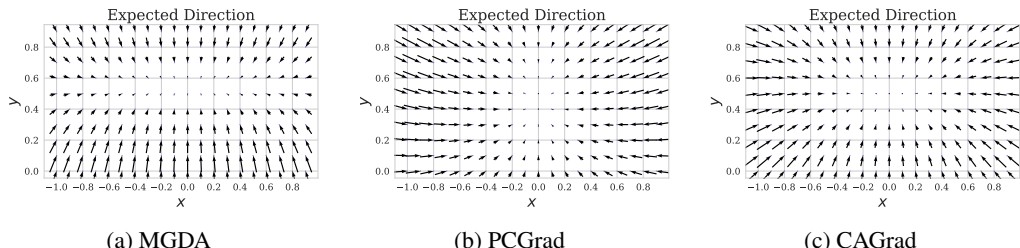

|             |             |             |
|-------------|-------------|-------------|
| (a) MGDA    | (b) PCGrad  | (c) CAGrad  |

Figure 3: Expected direction vector field for MGDA, PCGrad, CAGrad. We can observe that this biased direction can move algorithms away from the Pareto optimal set, which resists the algorithm to converge. Additionally, the stationary points for all three algorithms does not lies near the Pareto set.

It is easy to verify that for $\{(a,b)|a \in [-1,1], b \in [0,0.05]\}$, the two gradients consistently conflict with each other, then we know the y-axis value of the composite gradient for $(a,b)$ is $-2b - 4 - 2c\sqrt{a^2 + (b+2)^2} \geq -2 * 0.05 - 4 - 2c\sqrt{1^2 + (0+2)^2} = -4.1 - c2\sqrt{5} \geq -9$.

**Case 3:** when the received gradients are

$$\nabla \tilde{f}^1(a,b) = 2(a+1, b-2), \quad \nabla \tilde{f}^2(a,b) = 2(a-5, b+2).$$

It is easy to verify that for $\{(a,b)|a \in [-1,1], b \in [0,0.05]\}$, the two gradients consistently conflict with each other, then we know the y-axis value of the composite gradient for $(a,b)$ is $-2b + \frac{c\sqrt{2}}{2}\sqrt{(a-4)^2 + b^2} \geq -2 * 0.05 + \frac{c\sqrt{2}}{2}\sqrt{(1-4)^2 + 0^2} = -0.1 + 3c \leq 2.3$.

**Case 4:** when the received gradients are

$$\nabla \tilde{f}^1(a,b) = 2(a+5, b+2), \quad \nabla \tilde{f}^2(a,b) = 2(a-1, b-2).$$

It is easy to verify that for $\{(a,b)|a \in [-1,1], b \in [0,0.05]\}$, the two gradients consistently conflict with each other, then we know the y-axis value of the composite gradient for $(a,b)$ is $-2b + \frac{c\sqrt{2}}{2}\sqrt{(a+4)^2 + b^2} \geq -2 * 0.05 + \frac{c\sqrt{2}}{2}\sqrt{(-1+4)^2 + 0^2} = -0.1 + 3c \leq 2.3$.

Sum up the above, we can get $d_k^2 \geq 2.5 > 0$. □

Figure 3 shows the vector field for the composite direction calculated by MGDA, PCGrad, CAGrad, which aligns with the above Lemmas that indicate the expected direction is inverse to the Pareto set. This problem implies that $\mathbb{E}[\boldsymbol{x}_{k+1}]$ is more away from Pareto optimal than $\mathbb{E}[\boldsymbol{x}_k]$ if $\boldsymbol{x}_k$ lies in domain $\{(a,b)|a \in [-1,1], b \in [0,0.05]\}$. Thus the expected suboptimality gap is always $> 0$, which implies that MGDA, PCGrad, CAGrad do not converge to the optimal solution. In addition, we could also infer from the figure that, the three MOGM algorithms are stationary around $(0, 0.4\ 0.6)$, which represents a large expected suboptimality gap in this case.

□

## D.2 Proof of Proposition 2

*Proof.* The proposition can be proved by the following simple 2 objectives case, where we are provided with $\boldsymbol{g}^1 = (1,0), \boldsymbol{g}^2 = (1-\varepsilon, 0), \boldsymbol{g'}^2 = (1+\varepsilon, 0)$ where $\varepsilon << 1$. For this function, the min-norm direction is $\boldsymbol{g}^2$ for the first case, while is $\boldsymbol{g}^1$ for the second case, which means $\boldsymbol{\lambda}(\boldsymbol{g}^1, \boldsymbol{g}^2) = (0, 1)$ and $\boldsymbol{\lambda}(\boldsymbol{g}^1, \boldsymbol{g'}^2) = (1, 0)$. However, we know that $\boldsymbol{g}^2, \boldsymbol{g'}^2$ only have $2\varepsilon$ gap. When $\varepsilon$ limits to 0, there exists no finite $\beta$ to bound $\|\boldsymbol{\lambda}(\boldsymbol{g}^1, \boldsymbol{g}^2) - \boldsymbol{\lambda}(\boldsymbol{g}^1, \boldsymbol{g'}^2)\| / \|\boldsymbol{g}^2 - \boldsymbol{g'}^2\|$. □

**Generalize to PCGrad.** This property can be generalized to PCGrad, we next provide another counterexample to prove it.

*Proof.* Suppose we are provided with $\boldsymbol{g}^1 = (1,0), \boldsymbol{g}^2 = (-\varepsilon, 0), \boldsymbol{g'}^2 = (\varepsilon, 0), \varepsilon << 1$. For this function, $\boldsymbol{g}^1$ conflicts with $\boldsymbol{g}^2$ while aligned with $\boldsymbol{g'}^2$. This makes $\boldsymbol{\lambda}(\boldsymbol{g}^1, \boldsymbol{g}^2) = (\frac{1}{2} + \frac{\varepsilon}{2}, \frac{1}{2} + \frac{1}{2\varepsilon})$ and

$\boldsymbol{\lambda}(\boldsymbol{g}^1, \boldsymbol{g'}^2) = (\frac{1}{2}, \frac{1}{2})$. However, we know that $\boldsymbol{g}^2, \boldsymbol{g'}^2$ only have $2\varepsilon$ gap. When $\varepsilon$ limits to 0, there exists no finite $\beta$ to bound $\|\boldsymbol{\lambda}(\boldsymbol{g}^1, \boldsymbol{g}^2) - \boldsymbol{\lambda}(\boldsymbol{g}^1, \boldsymbol{g'}^2)\| / \|\boldsymbol{g}^2 - \boldsymbol{g'}^2\| = \sqrt{1 + \frac{1}{\varepsilon^4}}/4$. $\qquad\square$

**Generalize to CAGrad.** This property can also be generalized to CAGrad with parameter $c \in (0, 1/4)$ with the similar example with PCGrad. When $c \in [1/4, +\infty)$, CAGrad performs more like MGDA, it is not difficult to generalize the example of MGDA.

*Proof.* The proposition can be proved by the following simple 2 objectives case, where we are provided with $\boldsymbol{g}^1 = (1, 0), \boldsymbol{g}^2 = (-\varepsilon, 0), \boldsymbol{g'}^2 = (\varepsilon, 0), \varepsilon << 1$. For this function, $\boldsymbol{g}^1$ conflicts with $\boldsymbol{g}^2$ while aligned with $\boldsymbol{g'}^2$. This makes $\boldsymbol{\lambda}(\boldsymbol{g}^1, \boldsymbol{g}^2) = (\frac{1-c(1-\varepsilon)}{2}, \frac{1}{2})$ and $\boldsymbol{\lambda}(\boldsymbol{g}^1, \boldsymbol{g'}^2) = (\frac{1+c(1+\varepsilon)}{2}, \frac{1}{2})$. However, we know that $\boldsymbol{g}^2, \boldsymbol{g'}^2$ only have $2\varepsilon$ gap. When $\varepsilon$ limits to 0, there exists no finite $\beta$ to bound $\|\boldsymbol{\lambda}(\boldsymbol{g}^1, \boldsymbol{g}^2) - \boldsymbol{\lambda}(\boldsymbol{g}^1, \boldsymbol{g'}^2)\| / \|\boldsymbol{g}^2 - \boldsymbol{g'}^2\| = c/\varepsilon$. $\qquad\square$

# E   Missing Proofs for the Convergence of CRMOGM

We here give a rigorous convergence analysis of CRMOGM in the stochastic (mini-batch) gradient setting. We first fulfill the proofs for Lemma 1 and Lemma 2. Before providing the final proof, we would like to introduce some useful and insightful lemmas that reflect the algorithmic property and can be used for the final proof. Specifically, Appendix E.4 and E.5 decompose the convex and non-convex convergence bounds into sub-terms bounded by lemmas from Appendix E.3.

## E.1   Proof of Lemma 1

**Lemma 1.** *Under Assumption 1, 2. The correlation between $\boldsymbol{\lambda}_k$ and $\boldsymbol{G}_k$ can be bounded as $\|\mathbb{E}[\boldsymbol{G}_k \boldsymbol{\lambda}_k] - \mathbb{E}[\boldsymbol{G}_k]\mathbb{E}[\boldsymbol{\lambda}_k]\|_2^2 \le m\sigma^2 \mathbb{V}_{\boldsymbol{\xi}_k}(\boldsymbol{\lambda}_k)$, where $\mathbb{V}_{\boldsymbol{\xi}_k}(\boldsymbol{\lambda}_k) = \mathbb{E}_{\boldsymbol{\xi}_k}[\|\boldsymbol{\lambda}_k - \mathbb{E}_{\boldsymbol{\xi}_k}[\boldsymbol{\lambda}_k]\|_2^2]$.*

*Proof.* Denote $\boldsymbol{g}^i(\boldsymbol{x}_k, \boldsymbol{\xi}_k^i) = \boldsymbol{g}^i(\boldsymbol{x}_k), i \in [m]$ for simplicity. By the unbiasness of each stochastic gradient $\mathbb{E}[\boldsymbol{g}^i(\boldsymbol{x}_k)] = \nabla f^i(\boldsymbol{x}_k)$ for each objective, we have

$$
\begin{aligned}
\|\mathbb{E}[G_k \boldsymbol{\lambda}_k] - \mathbb{E}[G_k]\mathbb{E}[\boldsymbol{\lambda}_k]\|_2^2 &= \left\|\mathbb{E}[\sum_{i=1}^m \boldsymbol{\lambda}_k^i \boldsymbol{g}^i(\boldsymbol{x}_k)] - \mathbb{E}[\sum_{i=1}^m \boldsymbol{\lambda}_k^i \nabla f^i(\boldsymbol{x}_k)]\right\|_2^2 \\
&= \left\|\mathbb{E}[\sum_{i=1}^m \boldsymbol{\lambda}_k^i (\nabla f^i(\boldsymbol{x}_k) - \boldsymbol{g}^i(\boldsymbol{x}_k))]\right\|_2^2 \\
&= \left\|\mathbb{E}[\sum_{i=1}^m (\lambda_k^i - \mathbb{E}[\lambda_k]^i)(\nabla f^i(\boldsymbol{x}_k) - \boldsymbol{g}^i(\boldsymbol{x}_k))]\right\|_2^2 .
\end{aligned}
$$

The last equation is by the fact that

$$
\mathbb{E}[\sum_{i=1}^m \mathbb{E}[\lambda_k]^i (\nabla f^i(\boldsymbol{x}_k) - \boldsymbol{g}^i(\boldsymbol{x}_k))] = \sum_{i=1}^m \mathbb{E}[\lambda_k]^i \mathbb{E}[\nabla f^i(\boldsymbol{x}_k) - \boldsymbol{g}^i(\boldsymbol{x}_k)] = 0,
$$

where the first equation is by the linearity of expectation, and the second one is by the unbiasness of each stochastic gradient. Further, by the fact that $\|\mathbb{E}[a]\| \le \mathbb{E}[\|a\|]$ and the triangle inequality for the $l_2$ norm, we know that

$$
\|\mathbb{E}[\sum_{i=1}^m (\lambda_k^i - \mathbb{E}[\lambda_k]^i)(\nabla f^i(\boldsymbol{x}_k) - \boldsymbol{g}^i(\boldsymbol{x}_k))]\|_2^2 \le \mathbb{E}[\|\sum_{i=1}^m (\lambda_k^i - \mathbb{E}[\lambda_k]^i)(\nabla f^i(\boldsymbol{x}_k) - \boldsymbol{g}^i(\boldsymbol{x}_k))\|]^2
$$

$$
\le (\mathbb{E}[\sum_{i=1}^m \|(\lambda_k^i - \mathbb{E}[\lambda_k]^i)(\nabla f^i(\boldsymbol{x}_k) - \boldsymbol{g}^i(\boldsymbol{x}_k))\|])^2.
$$

Where, since $2ab \le \beta a^2 + \frac{1}{\beta} b^2$ for $\beta > 0$, we can get

$$\|(\lambda_k^i - \mathbb{E}[\lambda_k^i])(\nabla f^i(\boldsymbol{x}_k) - \boldsymbol{g}^i(\boldsymbol{x}_k))\| = |\lambda_k^i - \mathbb{E}[\lambda_k^i]| \|(\nabla f^i(\boldsymbol{x}_k) - \boldsymbol{g}^i(\boldsymbol{x}_k))\|$$
$$\le \frac{\beta_k}{2}(\lambda_k^i - \mathbb{E}[\lambda_k^i])^2 + \frac{1}{2\beta_k}\|(\nabla f^i(\boldsymbol{x}_k) - \boldsymbol{g}^i(\boldsymbol{x}_k))\|^2$$

Plug in, we obtain

$$(\mathbb{E}[\sum_{i=1}^m \|(\lambda_k^i - \mathbb{E}[\lambda_k]^i)(\nabla f^i(\boldsymbol{x}_k) - \boldsymbol{g}^i(\boldsymbol{x}_k))\|])^2$$
$$\le \mathbb{E}\left[ \sum_{i=1}^m \frac{\beta_k}{2}(\lambda_k^i - \mathbb{E}[\lambda_k]^i)^2 + \sum_{i=1}^m \frac{\|\nabla f^i(\boldsymbol{x}_k) - \boldsymbol{g}^i(\boldsymbol{x}_k)\|^2}{2\beta_k} \right]^2$$
$$= (\frac{\beta_k}{2}\mathbb{V}_{\boldsymbol{\xi}_k}(\lambda_k) + \frac{\sum_{i=1}^m \mathbb{V}_{\boldsymbol{\xi}_k}(\boldsymbol{g}^i(\boldsymbol{x}_k))}{2\beta_k})^2.$$

The last equation is by the definition of $\mathbb{V}_{\boldsymbol{\xi}_k}$. Here we let $\beta_k = \frac{\sqrt{\sum_{i=1}^m \mathbb{V}_{\boldsymbol{\xi}_k}(\boldsymbol{g}^i(\boldsymbol{x}_k))}}{\sqrt{\mathbb{V}_{\boldsymbol{\xi}_k}(\lambda_k)}}$, and apply all the above result, we finally get

$$\|\mathbb{E}[G_k\lambda_k] - \mathbb{E}[G_k]\mathbb{E}[\lambda_k]\|_2^2 \le \mathbb{V}_{\boldsymbol{\xi}_k}(\lambda_k)(\sum_{i=1}^m \mathbb{V}_{\boldsymbol{\xi}_k}(\boldsymbol{g}^i(\boldsymbol{x}_k))) \le m\sigma^2 \mathbb{V}_{\boldsymbol{\xi}_k}(\lambda_k).$$

We thus prove the Lemma. $\qquad\square$

***Remark.*** Lemma 1 suggests that the correlation between the composite weights and stochastic gradients is bounded by both the variance of composite weights $\boldsymbol{\lambda}_k$ and multiple stochastic gradients $\boldsymbol{g}^i(\boldsymbol{x}_k)$. Therefore, there are two effective ways to decrease this correlation: reduce the variance of $\lambda_k$, or the variance of $\boldsymbol{g}^i(\boldsymbol{x}_k)$. For the latter way, one can adopt variance reduction framework [18, 4] for stochastic optimization. However, it [18] requires global gradients that may be computational expensive, or slows down the convergence rate [4]. For the former one, the momentum mechanism on $\boldsymbol{\lambda}_k$ is a simple and effective way, which enhances both theoretical and empirical results without additional computational cost.

### E.2 Proof of Lemma 2

**Lemma 2.** *Under Assumption 1, $\boldsymbol{\lambda}_k$ variance for Algorithm 1 is bounded $\mathbb{V}_{\boldsymbol{\xi}_k}[\boldsymbol{\lambda}_k] \le m^2 B^2 (1 - \alpha_k)^2$.*

*Proof.* By the definition of Algorithm 1, we know that

$$\boldsymbol{\lambda}_k = \alpha_k \boldsymbol{\lambda}_{k-1} + (1 - \alpha_k)\hat{\boldsymbol{\lambda}}_k.$$

From the fact that $\mathbb{E}[\|X - \mathbb{E}[X]\|_2^2] \le \mathbb{E}[\|X - a\|_2^2]$ for any $a$, we can get

$$\mathbb{V}_{\boldsymbol{\xi}_k}[\boldsymbol{\lambda}_k] = \mathbb{E}_{\boldsymbol{\xi}_k}[\|\boldsymbol{\lambda}_k - \mathbb{E}_{\boldsymbol{\xi}_k}[\boldsymbol{\lambda}_k]\|_2^2] \le \mathbb{E}_{\boldsymbol{\xi}_k}[\|\boldsymbol{\lambda}_k - \boldsymbol{\lambda}_{k-1}\|_2^2]$$
$$= \mathbb{E}_{\boldsymbol{\xi}_k}[\|(1 - \alpha_k)(\hat{\boldsymbol{\lambda}} - \boldsymbol{\lambda}_{k-1})\|_2^2] = (1 - \alpha_k)^2 \mathbb{E}_{\boldsymbol{\xi}_k}[\|(\hat{\boldsymbol{\lambda}} - \boldsymbol{\lambda}_{k-1})\|_2^2] \le m^2 B^2 (1 - \alpha_k)^2.$$

The last inequality is by the fact $0 \le \lambda_k^i, \hat{\lambda}_k^i \le B, i = 1, \dots, m$. $\qquad\square$

### E.3 Lemmas needed for Proving Thm2/Thm3

This subsection presents several useful Lemmas for proving Theorem 2 and Theorem 3. Lemma 1 measures the expected gap between $-\sum_{i=1}^m \lambda_k^i \nabla f^i(\boldsymbol{x}_k)$ and $d_k$. The following Lemma bounds the dot production between them, which measures the local decrease of the composite functional value.

**Lemma 7.** *Under Assumption 1, 2, 3, for any gradient descent algorithm updated with composite gradient $G_k\boldsymbol{\lambda}_k$, we have the following inequality*

$$\mathbb{E}_{\boldsymbol{\xi}_k}\left[ \left( \sum_{i=1}^m \lambda_k^i \nabla f^i(\boldsymbol{x}_k) \right)^\top \boldsymbol{d}_k \right] \le 2mBH\sqrt{\mathbb{V}_{\boldsymbol{\xi}_k}[\boldsymbol{\lambda}_k]\sum_{i=1}^m \mathbb{V}_{\boldsymbol{\xi}_k}[\boldsymbol{g}^i(\boldsymbol{x}_k)]} - \mathbb{E}_{\boldsymbol{\xi}_k}\left[ \left\| \sum_{i=1}^m \lambda_k^i \nabla f^i(\boldsymbol{x}_k) \right\|_2^2 \right].$$

*Proof.* We first decompose the term into

$$\mathbb{E}_{\boldsymbol{\xi}_k}\left[\left(\sum_{i=1}^{m}\lambda_k^i\nabla f^i(\boldsymbol{x}_k)\right)^{\top}\boldsymbol{d}_k\right] = \mathbb{E}_{\boldsymbol{\xi}_k}\left[\left(\sum_{i=1}^{m}\lambda_k^i\nabla f^i(\boldsymbol{x}_k)\right)^{\top}\left(\boldsymbol{d}_k+\sum_{i=1}^{m}\lambda_k^i\nabla f^i(\boldsymbol{x}_k)\right)\right]$$
$$-\mathbb{E}_{\boldsymbol{\xi}_k}\left[\left\|\sum_{i=1}^{m}\lambda_k^i\nabla f^i(\boldsymbol{x}_k)\right\|_2^2\right].$$

The first term can potentially corrupt the effectiveness of optimization, and the second term measures the descent value. We next bound the first term. By definitions and decomposition, we get

$$\mathbb{E}_{\boldsymbol{\xi}_k}\left[\left(\sum_{i=1}^{m}\lambda_k^i\nabla f^i(\boldsymbol{x}_k)\right)^{\top}\left(\boldsymbol{d}_k+\sum_{i=1}^{m}\lambda_k^i\nabla f^i(\boldsymbol{x}_k)\right)\right]$$
$$=\mathbb{E}_{\boldsymbol{\xi}_k}\left[\left(\sum_{i=1}^{m}\lambda_k^i\nabla f^i(\boldsymbol{x}_k)\right)^{\top}\left(\sum_{i=1}^{m}\lambda_k^i(\nabla f^i(\boldsymbol{x}_k)-\boldsymbol{g}^i(\boldsymbol{x}_k))\right)\right]$$
$$=\mathbb{E}_{\boldsymbol{\xi}_k}\left[\left(\sum_{i=1}^{m}\lambda_k^i\nabla f^i(\boldsymbol{x}_k)\right)^{\top}\left(\sum_{i=1}^{m}(\lambda_k^i-\mathbb{E}[\lambda_k^i])(\nabla f^i(\boldsymbol{x}_k)-\boldsymbol{g}^i(\boldsymbol{x}_k))\right)\right] \quad \text{(term A)}$$
$$+\mathbb{E}_{\boldsymbol{\xi}_k}\left[\left(\sum_{i=1}^{m}(\lambda_k^i-\mathbb{E}[\lambda_k^i])\nabla f^i(\boldsymbol{x}_k)\right)^{\top}\left(\sum_{i=1}^{m}\mathbb{E}[\lambda_k^i](\nabla f^i(\boldsymbol{x}_k)-\boldsymbol{g}^i(\boldsymbol{x}_k))\right)\right] \quad \text{(term B)}$$
$$+\mathbb{E}_{\boldsymbol{\xi}_k}\left[\left(\sum_{i=1}^{m}\mathbb{E}[\lambda_k^i]\nabla f^i(\boldsymbol{x}_k)\right)^{\top}\left(\sum_{i=1}^{m}\mathbb{E}[\lambda_k^i](\nabla f^i(\boldsymbol{x}_k)-\boldsymbol{g}^i(\boldsymbol{x}_k))\right)\right] \quad \text{(term C)}.$$

We then bound each term individually. By Assumption 1 and Assumption 3, we know that $\lambda_k^i \in [0, B], \|\nabla f^i(\boldsymbol{x}_k)\| \leq H, \forall i = 1, \ldots, m$. By Cauchy–Schwartz inequality, we further know that for the term A

$$\text{term A} = \mathbb{E}_{\boldsymbol{\xi}_k}\left[\left(\sum_{i=1}^{m}\lambda_k^i\nabla f^i(\boldsymbol{x}_k)\right)^{\top}\left(\sum_{i=1}^{m}(\lambda_k^i-\mathbb{E}[\lambda_k^i])(\nabla f^i(\boldsymbol{x}_k)-\boldsymbol{g}^i(\boldsymbol{x}_k))\right)\right]$$
$$\leq\mathbb{E}_{\boldsymbol{\xi}_k}\left[\left\|\sum_{i=1}^{m}\lambda_k^i\nabla f^i(\boldsymbol{x}_k)\right\|_2\left\|\sum_{i=1}^{m}(\lambda_k^i-\mathbb{E}[\lambda_k^i])(\nabla f^i(\boldsymbol{x}_k)-\boldsymbol{g}^i(\boldsymbol{x}_k))\right\|_2\right]$$
$$\leq\mathbb{E}_{\boldsymbol{\xi}_k}\left[\left(\sum_{i=1}^{m}\lambda_k^i\left\|\nabla f^i(\boldsymbol{x}_k)\right\|_2\right)\left\|\sum_{i=1}^{m}(\lambda_k^i-\mathbb{E}[\lambda_k^i])(\nabla f^i(\boldsymbol{x}_k)-\boldsymbol{g}^i(\boldsymbol{x}_k))\right\|_2\right]$$
$$\leq\mathbb{E}_{\boldsymbol{\xi}_k}\left[\left(\sum_{i=1}^{m}BH\right)\left\|\sum_{i=1}^{m}(\lambda_k^i-\mathbb{E}[\lambda_k^i])(\nabla f^i(\boldsymbol{x}_k)-\boldsymbol{g}^i(\boldsymbol{x}_k))\right\|_2\right]$$
$$\leq mBH\mathbb{E}_{\boldsymbol{\xi}_k}\left[\sum_{i=1}^{m}|\lambda_k^i-\mathbb{E}[\lambda_k^i]|\|\nabla f^i(\boldsymbol{x}_k)-\boldsymbol{g}^i(\boldsymbol{x}_k)\|_2\right].$$

Similar with the proof in Lemma 1, by the fact that $ab \leq \frac{1}{2\beta_k}a^2 + \frac{\beta_k}{2}b^2$ for any $\beta_k > 0$, and by the linearity of expectation, we can get

$$\text{term A} \leq \frac{mBH}{2\beta_k}\sum_{i=1}^{m}\mathbb{E}_{\boldsymbol{\xi}_k}\left[|\lambda_k^i-\mathbb{E}[\lambda_k^i]|^2\right] + \frac{mBH\beta_k}{2}\sum_{i=1}^{m}\mathbb{E}_{\boldsymbol{\xi}_k}\left[\|\nabla f^i(\boldsymbol{x}_k)-\boldsymbol{g}^i(\boldsymbol{x}_k)\|_2^2\right]$$
$$\leq \frac{mBH}{2\beta_k}\mathbb{V}[\boldsymbol{\lambda}_k] + \frac{mBH\beta_k}{2}\sum_{i=1}^{m}\mathbb{V}[\boldsymbol{g}^i(\boldsymbol{x}_k)].$$

By setting $\beta_k = \sqrt{\mathbb{V}_{\boldsymbol{\xi}_k}[\lambda_k]}/\sqrt{\sum_{i=1}^m \mathbb{V}_{\boldsymbol{\xi}_k}[\boldsymbol{g}^i(\boldsymbol{x}_k)]}$, we have

$$\text{term A} \leq mBH\sqrt{\mathbb{V}[\boldsymbol{\lambda}_k]\sum_{i=1}^m \mathbb{V}[\boldsymbol{g}^i(\boldsymbol{x}_k)]}.$$

With similar tricks, we have for the term B

$$\text{term B} = \mathbb{E}_{\boldsymbol{\xi}_k}\left[\left(\sum_{i=1}^m (\lambda_k^i - \mathbb{E}[\lambda_k^i])\nabla f^i(\boldsymbol{x}_k)\right)^\top \left(\sum_{i=1}^m \mathbb{E}[\lambda_k^i](\nabla f^i(\boldsymbol{x}_k) - \boldsymbol{g}^i(\boldsymbol{x}_k))\right)\right]$$

$$\leq \mathbb{E}_{\boldsymbol{\xi}_k}\left[\left\|\sum_{i=1}^m (\lambda_k^i - \mathbb{E}[\lambda_k^i])\nabla f^i(\boldsymbol{x}_k)\right\|_2 \left\|\sum_{i=1}^m \mathbb{E}[\lambda_k^i](\nabla f^i(\boldsymbol{x}_k) - \boldsymbol{g}^i(\boldsymbol{x}_k))\right\|_2\right]$$

$$\leq \mathbb{E}_{\boldsymbol{\xi}_k}\left[\left(\sum_{i=1}^m H|\lambda_k^i - \mathbb{E}[\lambda_k^i]|\right)\left\|\sum_{i=1}^m \mathbb{E}[\lambda_k^i](\nabla f^i(\boldsymbol{x}_k) - \boldsymbol{g}^i(\boldsymbol{x}_k))\right\|_2\right]$$

$$= \mathbb{E}_{\boldsymbol{\xi}_k}\left[\sum_{i=1}^m H|\lambda_k^i - \mathbb{E}[\lambda_k^i]|\left\|\sum_{j=1}^m \mathbb{E}[\lambda_k^j](\nabla f^j(\boldsymbol{x}_k) - \boldsymbol{g}^j(\boldsymbol{x}_k))\right\|_2\right]$$

$$\leq H\mathbb{E}_{\boldsymbol{\xi}_k}\left[\left(\sum_{i=1}^m \frac{1}{2\beta_k}|\lambda_k^i - \mathbb{E}[\lambda_k^i]|^2 + \frac{\beta_k}{2}\left\|\sum_{i=1}^m \mathbb{E}[\lambda_k^i](\nabla f^i(\boldsymbol{x}_k) - \boldsymbol{g}^i(\boldsymbol{x}_k))\right\|_2^2\right)\right]$$

$$\leq H\mathbb{E}_{\boldsymbol{\xi}_k}\left[\left(\sum_{i=1}^m \frac{1}{2\beta_k}|\lambda_k^i - \mathbb{E}[\lambda_k^i]|^2 + \frac{\beta_k}{2}(\sum_{i=1}^m \mathbb{E}[\lambda_k^i]^2)(\sum_{i=1}^m \left\|\nabla f^i(\boldsymbol{x}_k) - \boldsymbol{g}^i(\boldsymbol{x}_k)\right\|_2^2)\right)\right]$$

$$\leq \frac{H}{2\beta_k}\sum_{i=1}^m \mathbb{E}_{\boldsymbol{\xi}_k}\left[|\lambda_k^i - \mathbb{E}[\lambda_k^i]|^2\right] + \frac{mB^2H\beta_k}{2}\sum_{i=1}^m \mathbb{E}_{\boldsymbol{\xi}_k}\left[\|\nabla f^i(\boldsymbol{x}_k) - \boldsymbol{g}^i(\boldsymbol{x}_k)\|_2^2\right]$$

$$= \frac{H}{2\beta_k}\mathbb{V}[\boldsymbol{\lambda}_k] + \frac{mB^2H\beta_k}{2}\sum_{i=1}^m \mathbb{V}[\boldsymbol{g}^i(\boldsymbol{x}_k)].$$

The first inequality is by Cauchy–Schwarz inequality. The second one is by the triangle inequality of $l_2$ norm, we have $\|\sum_{i=1}^m (\lambda_k^i - \mathbb{E}[\lambda_k^i])\nabla f^i(\boldsymbol{x}_k)\|_2 \leq \sum_{i=1}^m \|(\lambda_k^i - \mathbb{E}[\lambda_k^i])\nabla f^i(\boldsymbol{x}_k)\|_2 = \sum_{i=1}^m |\lambda_k^i - \mathbb{E}[\lambda_k^i]|\|\nabla f^i(\boldsymbol{x}_k)\|_2$. With Assumption 3, we know $\sum_{i=1}^m |\lambda_k^i - \mathbb{E}[\lambda_k^i]|\|\nabla f^i(\boldsymbol{x}_k)\|_2 \leq H\sum_{i=1}^m |\lambda_k^i - \mathbb{E}[\lambda_k^i]|$. The third one is from the fact that $ab \leq \frac{1}{2\beta_k}a^2 + \frac{\beta_k}{2}b^2$. The forth one is also from Cauchy-Schwartz inequality. The last one is from Assumption 1 that states $\mathbb{E}[\lambda_k^i] \leq B$. Further by setting $\beta_k = \sqrt{\mathbb{V}[\boldsymbol{\lambda}_k]}/B\sqrt{\sum_{i=1}^m \mathbb{V}[\boldsymbol{g}^i(\boldsymbol{x}_k)]}$, we have

$$\text{term B} \leq \frac{(m+1)BH}{2}\sqrt{\mathbb{V}[\boldsymbol{\lambda}_k]\sum_{i=1}^m \mathbb{V}[\boldsymbol{g}^i(\boldsymbol{x}_k)]} \leq mBH\sqrt{\mathbb{V}[\boldsymbol{\lambda}_k]\sum_{i=1}^m \mathbb{V}[\boldsymbol{g}^i(\boldsymbol{x}_k)]}.$$

For term C, by the fact that only $\boldsymbol{g}^i(\boldsymbol{x}_k)$ has randomness, we get

$$\text{term C} = \mathbb{E}_{\boldsymbol{\xi}_k}\left[\left(\sum_{i=1}^m \mathbb{E}[\lambda_k^i]\nabla f^i(\boldsymbol{x}_k)\right)^\top \left(\sum_{i=1}^m \mathbb{E}[\lambda_k^i](\nabla f^i(\boldsymbol{x}_k) - \boldsymbol{g}^i(\boldsymbol{x}_k))\right)\right]$$

$$= \left(\sum_{i=1}^m \mathbb{E}[\lambda_k^i]\nabla f^i(\boldsymbol{x}_k)\right)^\top \left(\sum_{i=1}^m \mathbb{E}[\lambda_k^i]\left(\nabla f^i(\boldsymbol{x}_k) - \mathbb{E}_{\boldsymbol{\xi}_k}[\boldsymbol{g}^i(\boldsymbol{x}_k)]\right)\right) = 0.$$

By summing up the above results, we obtain

$$\mathbb{E}_{\boldsymbol{\xi}_k}\left[\left(\sum_{i=1}^m \lambda_k^i \nabla f^i(\boldsymbol{x}_k)\right)^\top \left(\boldsymbol{d}_k + \sum_{i=1}^m \lambda_k^i \nabla f^i(\boldsymbol{x}_k)\right)\right] \leq 2mBH\sqrt{\mathbb{V}[\boldsymbol{\lambda}_k]\sum_{i=1}^m \mathbb{V}[\boldsymbol{g}^i(\boldsymbol{x}_k)]}.$$

Plug in the first decomposition in the beginning, we finally get

$$\mathbb{E}_{\boldsymbol{\xi}_k}\left[\left(\sum_{i=1}^{m}\lambda_k^i\nabla f^i(\boldsymbol{x}_k)\right)^{\top}\boldsymbol{d}_k\right]\leq 2mBH\sqrt{\mathbb{V}_{\boldsymbol{\xi}_k}[\lambda_k]\sum_{i=1}^{m}\mathbb{V}_{\boldsymbol{\xi}_k}[\boldsymbol{g}^i(\boldsymbol{x}_k)]}-\mathbb{E}_{\boldsymbol{\xi}_k}\left[\left\|\sum_{i=1}^{m}\lambda_k^i\nabla f^i(\boldsymbol{x}_k)\right\|_2^2\right].$$

$\square$

***Remark.*** The result from Lemma 7 indicates that the local decrease for expected composite loss $\mathbb{E}[\boldsymbol{\lambda}_k]^T\boldsymbol{F}(\boldsymbol{x})$ is bounded by $2mBH\sqrt{\mathbb{V}_{\boldsymbol{\xi}_k}[\boldsymbol{\lambda}_k]\sum_{i=1}^{m}\mathbb{V}_{\boldsymbol{\xi}_k}[\boldsymbol{g}^i(\boldsymbol{x}_k)]}-\mathbb{E}_{\boldsymbol{\xi}_k}\left[\left\|\sum_{i=1}^{m}\lambda_k^i\nabla f^i(\boldsymbol{x}_k)\right\|_2^2\right]$. The first term consists of the variances for composite weights and multiple gradients, and the second term is the negative norm of the effective direction. When the solution approximates the optimal, the second term would be smaller than the former term when the variances are large, then the value of the composite loss will increase, leading to the non-convergence. This result is aligned with the non-convergence for MOGM algorithms.

We next give a second-order bias analysis of the composite gradient $\boldsymbol{d}_k$ by the following lemma.

**Lemma 8.** *Under Assumption [1, 2, 3], for any gradient descent algorithm updated with composite gradient $\boldsymbol{G}_k\boldsymbol{\lambda}_k$, we have the following inequality*

$$\mathbb{E}_{\boldsymbol{\xi}_k}\left[\|\boldsymbol{d}_k\|_2^2-\|\sum_{i=1}^{m}\lambda_k^i\nabla f^i(\boldsymbol{x}_k)\|_2^2\right]\leq mB^2\sum_{i=1}^{m}\mathbb{V}_{\boldsymbol{\xi}_k}(\boldsymbol{g}_i)+4mBH\sqrt{\mathbb{V}_{\boldsymbol{\xi}_k}[\boldsymbol{\lambda}_k]\sum_{i=1}^{m}\mathbb{V}_{\boldsymbol{\xi}_k}[\boldsymbol{g}^i(\boldsymbol{x}_k)]}.$$

*Proof.* We first decompose the expectation into

$$\mathbb{E}_{\boldsymbol{\xi}_k}\left[\|\boldsymbol{d}_k\|_2^2-\|\sum_{i=1}^{m}\lambda_k^i\nabla f^i(\boldsymbol{x}_k)\|_2^2\right]$$

$$=\mathbb{E}_{\boldsymbol{\xi}_k}\left[\|\boldsymbol{d}_k+\sum_{i=1}^{m}\lambda_k^i\nabla f^i(\boldsymbol{x}_k)-\sum_{i=1}^{m}\lambda_k^i\nabla f^i(\boldsymbol{x}_k)\|_2^2-\|\sum_{i=1}^{m}\lambda_k^i\nabla f^i(\boldsymbol{x}_k)\|_2^2\right]$$

$$=\underbrace{\mathbb{E}_{\boldsymbol{\xi}_k}\left[\|\boldsymbol{d}_k+\sum_{i=1}^{m}\lambda_k^i\nabla f^i(\boldsymbol{x}_k)\|_2^2\right]}_{\text{term A}}-\underbrace{2\mathbb{E}_{\boldsymbol{\xi}_k}\left[\left(\sum_{i=1}^{m}\lambda_k^i\nabla f^i(\boldsymbol{x}_k)\right)^{\top}\left(\boldsymbol{d}_k+\sum_{i=1}^{m}\lambda_k^i\nabla f^i(\boldsymbol{x}_k)\right)\right]}_{\text{term B}}.$$

We next analyze the term A. By the triangle inequality of the $l_2$ norm, we have

$$\text{term A}=\mathbb{E}_{\boldsymbol{\xi}_k}\left[\left\|\sum_{i=1}^{m}\lambda_k^i\nabla f^i(\boldsymbol{x}_k)+\boldsymbol{d}_k\right\|_2^2\right]=\mathbb{E}_{\boldsymbol{\xi}_k}\left[\left\|\sum_{i=1}^{m}\lambda_k^i(\nabla f^i(\boldsymbol{x}_k)-\boldsymbol{g}^i(\boldsymbol{x}_k))\right\|_2^2\right]$$

$$\leq\mathbb{E}_{\boldsymbol{\xi}_k}\left[\left(\sum_{i=1}^{m}\lambda_k^i\left\|(\nabla f^i(\boldsymbol{x}_k)-\boldsymbol{g}^i(\boldsymbol{x}_k))\right\|_2\right)^2\right],$$

Further, by the fact that $\lambda_k^i\in[0,B], i=1,\ldots,m$, we know that

$$\text{term A}\leq\mathbb{E}_{\boldsymbol{\xi}_k}\left[\left(\sum_{i=1}^{m}\lambda_k^{i^2}\right)\left(\sum_{i=1}^{m}\left\|\nabla f^i(\boldsymbol{x}_k)-\boldsymbol{g}^i(\boldsymbol{x}_k)\right\|_2^2\right)\right]$$

$$\leq\mathbb{E}_{\boldsymbol{\xi}_k}\left[mB^2\sum_{i=1}^{m}\left\|(\nabla f^i(\boldsymbol{x}_k)-\boldsymbol{g}^i(\boldsymbol{x}_k))\right\|^2\right]=mB^2\sum_{i=1}^{m}\mathbb{V}_{\boldsymbol{\xi}_k}(\boldsymbol{g}_i).$$

We then analyze the term B. Similar with the proof in Lemma 7, and by the fact that the minus sign does not affect the inequality, we know that

$$\text{term B} = -2\mathbb{E}_{\boldsymbol{\xi}_k}\left[\left(\sum_{i=1}^m \lambda_k^i \nabla f^i(\boldsymbol{x}_k)\right)^\top \left(\boldsymbol{d}_k + \sum_{i=1}^m \lambda_k^i \nabla f^i(\boldsymbol{x}_k)\right)\right]$$

$$\leq 4mBH\sqrt{\mathbb{V}_{\boldsymbol{\xi}_k}[\lambda_k]\sum_{i=1}^m \mathbb{V}_{\boldsymbol{\xi}_k}[\boldsymbol{g}^i(\boldsymbol{x}_k)]}.$$

Combining all the results, we obtain

$$\mathbb{E}_{\boldsymbol{\xi}_k}\left[\|\boldsymbol{d}_k\|_2^2 - \|\sum_{i=1}^m \lambda_k^i \nabla f^i(\boldsymbol{x}_k)\|_2^2\right] \leq mB^2\sum_{i=1}^m \mathbb{V}_{\boldsymbol{\xi}_k}(\boldsymbol{g}_i) + 4mBH\sqrt{\mathbb{V}_{\boldsymbol{\xi}_k}[\lambda_k]\sum_{i=1}^m \mathbb{V}_{\boldsymbol{\xi}_k}[\boldsymbol{g}^i(\boldsymbol{x}_k)]}.$$

$\square$

***Remark.*** In the single-objective stochastic optimization, the second term is zero, leaving only the gradient variance in the bound. While in the multi-objective stochastic optimization, the second term is non-zero and related to the number of objectives, the variance of composite weight, and the gradient variance, which makes it significantly different.

### E.4 Proof for Theorem 2 (Convex Convergence)

**Theorem 2.** *Under Assumption 1, 2, 3. For the sequence $\boldsymbol{x}_0, \boldsymbol{x}_1, \ldots, \boldsymbol{x}_n$ generated by Algorithm 1, we assume objective functions $f^1(\boldsymbol{x}), \ldots, f^m(\boldsymbol{x})$ are all convex with bounded optimal values as $f^1(\boldsymbol{x}^*), \ldots, f^m(\boldsymbol{x}^*) \leq F$ for $\boldsymbol{x}^*$ in Pareto set, and the distance from sequence to Pareto set is bounded, i.e., $\|\boldsymbol{x}_k - \boldsymbol{x}^*\| \leq D$. Set $0 \leq \eta_n \leq \ldots \leq \eta_1 \leq 1/mLB, \alpha_k \in (0,1]$ for $k = 1, \ldots, n$ in Algorithm 1, it achieves*

$$\frac{1}{n}\sum_{k=1}^n \mathbb{E}\left[\max_{\boldsymbol{x}_k^* \in \mathcal{K}} \min_{\boldsymbol{\lambda}_k^* \in S_m} (\boldsymbol{\lambda}_k^{*\top}\boldsymbol{F}(\boldsymbol{x}_k) - \boldsymbol{\lambda}_k^{*\top}\boldsymbol{F}(\boldsymbol{x}_k^*))\right] \leq \frac{D^2}{\eta_n n} + \frac{m^2B^2(\sigma^2 + \sigma H + H^2)}{n}\sum_{k=1}^n \eta_k$$

$$+ \frac{DB\sigma m^{3/2}}{n}\sum_{k=1}^n (1-\alpha_k) + \frac{6m^{5/2}B^2H\sigma}{n}\sum_{k=1}^n (1-\alpha_k)\eta_k + \frac{2F}{n}\sum_{l=1}^n l(1-\alpha_{l+1})\sum_{i=1}^m |\lambda_l^i - \hat{\lambda}_{l+1}^i|.$$

To prove Theorem 2, it is sufficient to prove the bound for

$$\frac{1}{n}\sum_{k=1}^n \mathbb{E}[\sum_{i=1}^m \lambda_k^i f^i(\boldsymbol{x}_k) - \min_{\boldsymbol{x}_k^* \in \mathcal{K}}\sum_{i=1}^m \lambda_k^i f^i(\boldsymbol{x}_k^*)]. \tag{6}$$

We next provide the reason. Specifically, from Assumption 1, we have $\sum_{i=1}^m \lambda_k^i \geq 1$ for $k = 1, \ldots, n$. Therefore, we know that

$$\frac{1}{n}\sum_{k=1}^n \mathbb{E}[\frac{\sum_{i=1}^m \lambda_k^i f^i(\boldsymbol{x}_{k+1})}{\sum_{j=1}^m \lambda_k^j} - \frac{\min_{\boldsymbol{x}_k^* \in \mathcal{K}}\sum_{i=1}^m \lambda_k^i f^i(\boldsymbol{x}^*)}{\sum_{i=1}^m \lambda_k^i}] \leq \frac{1}{n}\sum_{k=1}^n \mathbb{E}[\sum_{i=1}^m \lambda_k^i f^i(\boldsymbol{x}_k) - \min_{\boldsymbol{x}_k^* \in \mathcal{K}}\sum_{i=1}^m \lambda_k^i f^i(\boldsymbol{x}_k^*)].$$

Furthermore, denote $\bar{\boldsymbol{\lambda}}_k = \boldsymbol{\lambda}_k / \sum_{j=1}^m \lambda_k^j$. By the property for the max-min operator, the left side of the above inequality can upper bound what we need to bound in Theorem 2 as

$$\frac{1}{n}\sum_{k=1}^n \mathbb{E}\left[\max_{\boldsymbol{x}_k^* \in \mathcal{K}} \min_{\boldsymbol{\lambda}_k^* \in S_m}(\boldsymbol{\lambda}_k^{*\top}\boldsymbol{F}(\boldsymbol{x}_k) - \boldsymbol{\lambda}_k^{*\top}\boldsymbol{F}(\boldsymbol{x}_k^*))\right] \leq \frac{1}{n}\sum_{k=1}^n \mathbb{E}[\sum_{i=1}^m \bar{\boldsymbol{\lambda}}_k^\top \boldsymbol{F}(\boldsymbol{x}_k) - \min_{\boldsymbol{x}_k^* \in \mathcal{K}} \bar{\boldsymbol{\lambda}}_k^\top \boldsymbol{F}(\boldsymbol{x}_k^*)].$$

Therefore, the rest of the proof is to bound 6. Before the final proof, we first introduce several Lemmas. The following Lemma give an induction as

**Lemma 9.** *Under the same assumption as Theorem 2, for any gradient descent algorithm updated with composite gradient $\boldsymbol{G}_k\boldsymbol{\lambda}_k$, and any $\boldsymbol{x}^*$ in Pareto set, we have the following inequality*

$$\mathbb{E}_{\boldsymbol{\xi}_k}[\sum_{i=1}^{m}\lambda_k^i f^i(\boldsymbol{x}_k) - \sum_{i=1}^{m}\lambda_k^i f^i(\boldsymbol{x}^*)] \le \frac{1}{2\eta_k}\mathbb{E}_{\boldsymbol{\xi}_k}\left[\|\boldsymbol{x}_k - \boldsymbol{x}^*\|_2^2 - \|\boldsymbol{x}_{k+1} - \boldsymbol{x}^*\|_2^2\right]$$

$$+ D\sqrt{\mathbb{V}_{\boldsymbol{\xi}_k}(\lambda_k)\left(\sum_{i=1}^{m}\mathbb{V}_{\boldsymbol{\xi}_k}(\boldsymbol{g}^i(\boldsymbol{x}_k))\right)} + 6\eta_k mBH\sqrt{\mathbb{V}_{\boldsymbol{\xi}_k}[\lambda_k]\sum_{i=1}^{m}\mathbb{V}_{\boldsymbol{\xi}_k}[\boldsymbol{g}^i(\boldsymbol{x}_k)]}$$

$$+ \eta_k mB^2\sum_{i=1}^{m}\mathbb{V}_{\boldsymbol{\xi}_k}(\boldsymbol{g}_i) + \eta_k mB^2 H\left(mH + \sqrt{m\sum_{i=1}^{m}\mathbb{V}[\boldsymbol{g}^i(\boldsymbol{x}_k)]}\right).$$

*Proof.* From the $L$-smoothness of each objective function, we have

$$\lambda_k^i f^i(\boldsymbol{x}_{k+1}) \le \lambda_k^i\left(f^i(\boldsymbol{x}_k) + \eta_k\nabla f^i(\boldsymbol{x}_k)^\top(\boldsymbol{x}_{k+1} - \boldsymbol{x}_k) + \frac{L}{2}\|\boldsymbol{x}_{k+1} - \boldsymbol{x}_k\|_2^2\right).$$

By the iterative rule, we know that $\boldsymbol{x}_{k+1} = \boldsymbol{x}_k + \eta_k\boldsymbol{d}_k$. Further, by the convexity of $f^i$, and denote $\boldsymbol{x}^*$ as any point in the feasible set, we have $f^i(\boldsymbol{x}_k) \le f^i(\boldsymbol{x}^*) + \nabla f^i(\boldsymbol{x}_k)^\top(\boldsymbol{x}_k - \boldsymbol{x}^*)$. Then we can get

$$\lambda_k^i f^i(\boldsymbol{x}_{k+1}) \le \lambda_k^i f^i(\boldsymbol{x}_k) + \eta_k\lambda_k^i\nabla f^i(\boldsymbol{x}_k)^\top\boldsymbol{d}_k + \frac{L\eta_k^2}{2}\lambda_k^i\|\boldsymbol{d}_k\|_2^2$$

$$\le \lambda_k^i f^i(\boldsymbol{x}^*) + \lambda_k^i\nabla f^i(\boldsymbol{x}_k)^\top(\boldsymbol{x}_k - \boldsymbol{x}^*) + \eta_k\lambda_k^i\nabla f^i(\boldsymbol{x}_k)^\top\boldsymbol{d}_k + \frac{L\eta_k^2}{2}\lambda_k^i\|\boldsymbol{d}_k\|_2^2.$$

Summing up above for $i = 1, \ldots, m$, we have

$$\sum_{i=1}^{m}\lambda_k^i f^i(\boldsymbol{x}_{k+1}) - \sum_{i=1}^{m}\lambda_k^i f^i(\boldsymbol{x}^*)$$

$$\le \left(\sum_{i=1}^{m}\lambda_k^i\nabla f^i(\boldsymbol{x}_k)\right)^\top(\boldsymbol{x}_k - \boldsymbol{x}^*) + \eta_k\left(\sum_{i=1}^{m}\lambda_k^i\nabla f^i(\boldsymbol{x}_k)\right)^\top\boldsymbol{d}_k + \frac{L\eta_k^2}{2}\left(\sum_{i=1}^{m}\lambda_k^i\right)\|\boldsymbol{d}_k\|_2^2.$$

By setting $\eta_k \le 1/mLB$, we can know that $L\eta_k\left(\sum_{i=1}^{m}\lambda_k^i\right) \le L\eta_k mB \le 1$. Then, the third term can be bounded as $\frac{L\eta_k^2}{2}\left(\sum_{i=1}^{m}\lambda_k^i\right)\|\boldsymbol{d}_k\|_2^2 \le \frac{\eta_k}{2}\|\boldsymbol{d}_k\|_2^2$. Plugging such a bound for the third term, and take the expectation on random variable $\boldsymbol{\xi}_k$, we then obtain

$$\mathbb{E}_{\boldsymbol{\xi}_k}\left[\sum_{i=1}^{m}\lambda_k^i f^i(\boldsymbol{x}_{k+1}) - \sum_{i=1}^{m}\lambda_k^i f^i(\boldsymbol{x}^*)\right]$$

$$\le \mathbb{E}_{\boldsymbol{\xi}_k}\left[\left(\sum_{i=1}^{m}\lambda_k^i\nabla f^i(\boldsymbol{x}_k)\right)^\top(\boldsymbol{x}_k - \boldsymbol{x}^*) + \eta_k\left(\sum_{i=1}^{m}\lambda_k^i\nabla f^i(\boldsymbol{x}_k)\right)^\top\boldsymbol{d}_k + \frac{\eta_k}{2}\|\boldsymbol{d}_k\|_2^2\right].$$

By the linearity of expectation, we rearrange the above term into

$$\mathbb{E}_{\boldsymbol{\xi}_k}[\sum_{i=1}^{m}\lambda_k^i f^i(\boldsymbol{x}_{k+1}) - \sum_{i=1}^{m}\lambda_k^i f^i(\boldsymbol{x}^*)]$$

$$\le \mathbb{E}_{\boldsymbol{\xi}_k}\left[\left(\sum_{i=1}^{m}\lambda_k^i\nabla f^i(\boldsymbol{x}_k)\right)^\top(\boldsymbol{x}_k - \boldsymbol{x}^*)\right] + \eta_k\mathbb{E}_{\boldsymbol{\xi}_k}\left[\left(\sum_{i=1}^{m}\lambda_k^i\nabla f^i(\boldsymbol{x}_k)\right)^\top\boldsymbol{d}_k\right] + \frac{\eta_k}{2}\mathbb{E}_{\boldsymbol{\xi}_k}\left[\|\boldsymbol{d}_k\|_2^2\right].$$

We next to bound the first term. By rearranging the equation, we get

$$\mathbb{E}_{\boldsymbol{\xi}_k}\left[\left(\sum_{i=1}^{m}\lambda_k^i\nabla f^i(\boldsymbol{x}_k)\right)^\top(\boldsymbol{x}_k - \boldsymbol{x}^*)\right] = \mathbb{E}_{\boldsymbol{\xi}_k}\left[-\boldsymbol{d}_k^\top(\boldsymbol{x}_k - \boldsymbol{x}^*) + \left(\sum_{i=1}^{m}\lambda_k^i\nabla f^i(\boldsymbol{x}_k) + \boldsymbol{d}_k\right)^\top(\boldsymbol{x}_k - \boldsymbol{x}^*)\right]$$

$$= \frac{1}{2\eta_k}\mathbb{E}_{\boldsymbol{\xi}_k}\left[\|\boldsymbol{x}_k - \boldsymbol{x}^*\|_2^2 - \|\boldsymbol{x}_{k+1} - \boldsymbol{x}^*\|_2^2\right] + \frac{\eta_k}{2}\mathbb{E}_{\boldsymbol{\xi}_k}\left[\|\boldsymbol{d}_k\|_2^2\right] + \mathbb{E}_{\boldsymbol{\xi}_k}\left[\left(\sum_{i=1}^{m}\lambda_k^i\nabla f^i(\boldsymbol{x}_k) + \boldsymbol{d}_k\right)^\top(\boldsymbol{x}_k - \boldsymbol{x}^*)\right],$$

where the last equation is by the fact that $\boldsymbol{x}_{k+1} = \boldsymbol{x}_k + \eta_k \boldsymbol{d}_k$. Since we know that

$$
\mathbb{E}_{\boldsymbol{\xi}_k} \left[ \sum_{i=1}^{m} \lambda_k^i \nabla f^i(\boldsymbol{x}_k) + \boldsymbol{d}_k \right]^{\top} (\boldsymbol{x}_k - \boldsymbol{x}^*) \leq \left\| \mathbb{E}_{\boldsymbol{\xi}_k} \left[ \sum_{i=1}^{m} \lambda_k^i \nabla f^i(\boldsymbol{x}_k) + \boldsymbol{d}_k \right] \right\|_2 \|\boldsymbol{x}_k - \boldsymbol{x}^*\|_2
$$

$$
\leq D \left\| \mathbb{E}_{\boldsymbol{\xi}_k} \left[ \sum_{i=1}^{m} \lambda_k^i \nabla f^i(\boldsymbol{x}_k) + \boldsymbol{d}_k \right] \right\|_2 = D \sqrt{ \left\| \mathbb{E}_{\boldsymbol{\xi}_k} \left[ \sum_{i=1}^{m} \lambda_k^i \nabla f^i(\boldsymbol{x}_k) + \boldsymbol{d}_k \right] \right\|_2^2 }
$$

$$
\leq D \sqrt{ \mathbb{V}_{\boldsymbol{\xi}_k}(\lambda_k) \left( \sum_{i=1}^{m} \mathbb{V}_{\boldsymbol{\xi}_k}(\boldsymbol{g}^i(\boldsymbol{x}_k)) \right) }.
$$

The second inequality is by the assumption that the set $\{\boldsymbol{x}_k, k = 1, \dots, K\}$ is a subset of a bounded set with diagram $D$, and the third one is from Lemma 1. By combining all the above results, we have

$$
\mathbb{E}_{\boldsymbol{\xi}_k} [\sum_{i=1}^{m} \lambda_k^i f^i(\boldsymbol{x}_{k+1}) - \sum_{i=1}^{m} \lambda_k^i f^i(\boldsymbol{x}^*)]
$$

$$
\leq \frac{1}{2\eta_k} \mathbb{E}_{\boldsymbol{\xi}_k} \left[ \|\boldsymbol{x}_k - \boldsymbol{x}^*\|_2^2 - \|\boldsymbol{x}_{k+1} - \boldsymbol{x}^*\|_2^2 \right] + D \sqrt{ \mathbb{V}_{\boldsymbol{\xi}_k}(\lambda_k) \left( \sum_{i=1}^{m} \mathbb{V}_{\boldsymbol{\xi}_k}(\boldsymbol{g}^i(\boldsymbol{x}_k)) \right) } + \eta_k \mathbb{E}_{\boldsymbol{\xi}_k} \left[ \|\boldsymbol{d}_k\|_2^2 \right]
$$

$$
+ \eta_k \mathbb{E}_{\boldsymbol{\xi}_k} \left[ \left( \sum_{i=1}^{m} \lambda_k^i \nabla f^i(\boldsymbol{x}_k) \right)^{\top} \boldsymbol{d}_k \right]
$$

$$
\leq \frac{1}{2\eta_k} \mathbb{E}_{\boldsymbol{\xi}_k} \left[ \|\boldsymbol{x}_k - \boldsymbol{x}^*\|_2^2 - \|\boldsymbol{x}_{k+1} - \boldsymbol{x}^*\|_2^2 \right] + D \sqrt{ \mathbb{V}_{\boldsymbol{\xi}_k}(\lambda_k) \sum_{i=1}^{m} \mathbb{V}_{\boldsymbol{\xi}_k}(\boldsymbol{g}^i(\boldsymbol{x}_k)) }
$$

$$
+ \eta_k \mathbb{E}_{\boldsymbol{\xi}_k} \left[ \|\boldsymbol{d}_k\|_2^2 - \| \sum_{i=1}^{m} \lambda_k^i \nabla f^i(\boldsymbol{x}_k) \|_2^2 \right] + 2\eta_k m B H \sqrt{ \mathbb{V}_{\boldsymbol{\xi}_k}[\lambda_k] \sum_{i=1}^{m} \mathbb{V}_{\boldsymbol{\xi}_k}[\boldsymbol{g}^i(\boldsymbol{x}_k)] } \quad \text{(from Lemma 7)}
$$

$$
\leq \frac{1}{2\eta_k} \mathbb{E}_{\boldsymbol{\xi}_k} \left[ \|\boldsymbol{x}_k - \boldsymbol{x}^*\|_2^2 - \|\boldsymbol{x}_{k+1} - \boldsymbol{x}^*\|_2^2 \right] + D \sqrt{ \mathbb{V}_{\boldsymbol{\xi}_k}(\lambda_k) \left( \sum_{i=1}^{m} \mathbb{V}_{\boldsymbol{\xi}_k}(\boldsymbol{g}^i(\boldsymbol{x}_k)) \right) }
$$

$$
+ 6\eta_k m B H \sqrt{ \mathbb{V}_{\boldsymbol{\xi}_k}[\lambda_k] \sum_{i=1}^{m} \mathbb{V}_{\boldsymbol{\xi}_k}[\boldsymbol{g}^i(\boldsymbol{x}_k)] } + \eta_k m B^2 \sum_{i=1}^{m} \mathbb{V}_{\boldsymbol{\xi}_k}(\boldsymbol{g}_i) \quad \text{(from Lemma 8)}.
$$

In addition, there is a slight different with the Lemma that we aim to prove: the left side is $\boldsymbol{x}_{k+1}$ not $\boldsymbol{x}_k$. The following we fix this issue. By the H-Lipschitz continuous for each objective, we know that

$$
\mathbb{E}_{\boldsymbol{\xi}_k} [\sum_{i=1}^{m} \lambda_k^i f^i(\boldsymbol{x}_k) - \sum_{i=1}^{m} \lambda_k^i f^i(\boldsymbol{x}_{k+1})] \leq \mathbb{E}_{\boldsymbol{\xi}_k} [\sum_{i=1}^{m} \lambda_k^i H \|\boldsymbol{x}_k - \boldsymbol{x}_{k+1}\|)] \leq m B H \eta_k \mathbb{E}_{\boldsymbol{\xi}_k} [\|\boldsymbol{d}_k\|].
$$

By the triangle inequality of norm and the definition for the composite gradient $\boldsymbol{d}_k$, we have

$$
\begin{aligned}
\mathbb{E}_{\boldsymbol{\xi}_k}[\|\boldsymbol{d}_k\|] &= \mathbb{E}_{\boldsymbol{\xi}_k}[\|\boldsymbol{d}_k + \sum_{i=1}^m \lambda_k^i \nabla f^i(\boldsymbol{x}_k) - \sum_{i=1}^m \lambda_k^i \nabla f^i(\boldsymbol{x}_k)\|] \\
&\leq \mathbb{E}_{\boldsymbol{\xi}_k}[\|\boldsymbol{d}_k + \sum_{i=1}^m \lambda_k^i \nabla f^i(\boldsymbol{x}_k)\| + \|\sum_{i=1}^m \lambda_k^i \nabla f^i(\boldsymbol{x}_k)\|] \\
&= \mathbb{E}_{\boldsymbol{\xi}_k}[\|\sum_{i=1}^m \lambda_k^i (\nabla f^i(\boldsymbol{x}_k) - \boldsymbol{g}^i(\boldsymbol{x}_k)\| + \|\sum_{i=1}^m \lambda_k^i \nabla f^i(\boldsymbol{x}_k)\|] \\
&\leq \mathbb{E}_{\boldsymbol{\xi}_k}[\sum_{i=1}^m \lambda_k^i \|\nabla f^i(\boldsymbol{x}_k) - \boldsymbol{g}^i(\boldsymbol{x}_k)\| + \sum_{i=1}^m \lambda_k^i \|\nabla f^i(\boldsymbol{x}_k)\|] \\
&\leq \mathbb{E}_{\boldsymbol{\xi}_k}[\sum_{i=1}^m \lambda_k^i \|\nabla f^i(\boldsymbol{x}_k) - \boldsymbol{g}^i(\boldsymbol{x}_k)\| + \sum_{i=1}^m \lambda_k^i H] \\
&\leq \mathbb{E}_{\boldsymbol{\xi}_k}[\sum_{i=1}^m \lambda_k^i \|\nabla f^i(\boldsymbol{x}_k) - \boldsymbol{g}^i(\boldsymbol{x}_k)\|] + mBH.
\end{aligned}
$$

The third inequality is from the H-Lipschitz, and the last one is from Assumption 1. By the fact that $ab \leq \frac{1}{2\beta_k}a^2 + \frac{\beta_k}{2}b^2$ for any $\beta_k > 0$, we can have

$$
\begin{aligned}
\mathbb{E}_{\boldsymbol{\xi}_k}[\|\boldsymbol{d}_k\|] &\leq \mathbb{E}_{\boldsymbol{\xi}_k}[\sum_{i=1}^m \frac{1}{2\beta_k}(\lambda_k^i)^2 + \sum_{i=1}^m \frac{\beta_k}{2}\|\nabla f^i(\boldsymbol{x}_k) - \boldsymbol{g}^i(\boldsymbol{x}_k\|^2] + mBH \\
&\leq \frac{mB^2}{2\beta_k} + \frac{\beta_k}{2}\sum_{i=1}^m \mathbb{V}[\boldsymbol{g}^i(\boldsymbol{x}_k] + mBH.
\end{aligned}
$$

Let $\beta_k = B\sqrt{m}/\sqrt{\sum_{i=1}^m \mathbb{V}[\boldsymbol{g}^i(\boldsymbol{x}_k)]}$, we have $\mathbb{E}_{\boldsymbol{\xi}_k}[\|\boldsymbol{d}_k\|] \leq B\sqrt{m\sum_{i=1}^m \mathbb{V}[\boldsymbol{g}^i(\boldsymbol{x}_k)]} + mBH$. Plug in the result, we can get

$$
\mathbb{E}_{\boldsymbol{\xi}_k}[\sum_{i=1}^m \lambda_k^i f^i(\boldsymbol{x}_k) - \sum_{i=1}^m \lambda_k^i f^i(\boldsymbol{x}_{k+1})] \leq \eta_k mB^2 H(mH + \sqrt{m\sum_{i=1}^m \mathbb{V}[\boldsymbol{g}^i(\boldsymbol{x}_k)]}.
$$

Combining all the above, we prove the lemma. ☐

By summing up the above lemma, we get the following one

**Lemma 10.** *Under the same assumption as Theorem 2, for any $\boldsymbol{x}^*$ in Pareto set, we have the following inequality for CRMOGM algorithm*

$$
\begin{aligned}
\sum_{k=1}^n \mathbb{E}[\sum_{i=1}^m \lambda_k^i f^i(\boldsymbol{x}_{k+1}) - \sum_{i=1}^m \lambda_k^i f^i(\boldsymbol{x}^*)] &\leq \left(\frac{1}{2\eta_n} + \frac{1}{2\eta_1}\right) D^2 + m^{3/2} DB\sigma \sum_{k=1}^n (1 - \alpha_k) \\
&+ 6m^{5/2} B^2 H\sigma \sum_{k=1}^n (1 - \alpha_k)\eta_k + m^2 B^2(\sigma^2 + \sigma H + H^2)\sum_{k=1}^n \eta_k.
\end{aligned}
$$

*Proof of Theorem 2.* From Lemma 2, we know that $\mathbb{V}_{\boldsymbol{\xi}_k}[\lambda_k] \leq m^2 B^2(1-\alpha_k)^2$. From Assumption 2, we know that $\mathbb{V}_{\boldsymbol{\xi}_k}(\boldsymbol{g}^i(\boldsymbol{x}_k)) \leq \sigma^2$. Then plug into Lemma 9, we get

$$
\begin{aligned}
\sum_{i=1}^m \mathbb{E}_{\boldsymbol{\xi}_k}[\lambda_k^i f^i(\boldsymbol{x}_{k+1}) - \lambda_k^i f^i(\boldsymbol{x}^*)] &\leq \frac{1}{2\eta_k}\mathbb{E}_{\boldsymbol{\xi}_k}\left[\|\boldsymbol{x}_k - \boldsymbol{x}^*\|_2^2 - \|\boldsymbol{x}_{k+1} - \boldsymbol{x}^*\|_2^2\right] \\
&+ mDB\sigma\sqrt{m}(1 - \alpha_k) + 6m^2 B^2 H\sigma\sqrt{m}(1 - \alpha_k)\eta_k + m^2 B^2(\sigma^2 + \sigma H + H^2)\eta_k.
\end{aligned}
$$

Sum up the above inequality from $k=1$ to $k=n$, we have

$$\sum_{k=1}^{n}\sum_{i=1}^{m}\mathbb{E}_{\boldsymbol{\xi}_k}[\lambda_k^i f^i(\boldsymbol{x}_{k+1}) - \lambda_k^i f^i(\boldsymbol{x}^*)] \leq \sum_{k=1}^{n}\frac{1}{2\eta_k}\mathbb{E}_{\boldsymbol{\xi}_k}\left[\|\boldsymbol{x}_k - \boldsymbol{x}^*\|_2^2 - \|\boldsymbol{x}_{k+1} - \boldsymbol{x}^*\|_2^2\right]$$

$$+ \sum_{k=1}^{n}(mDB\sigma\sqrt{m}(1-\alpha_k) + 6m^2 B^2 H\sigma\sqrt{m}(1-\alpha_k)\eta_k + m^2 B^2(\sigma^2 + \sigma H + H^2)\sum_{k=1}^{n}\eta_k).$$

Take expectation of $\xi_1, \xi_2, \ldots, \xi_n$ on the both sides, we can get

$$\sum_{k=1}^{n}\sum_{i=1}^{m}\mathbb{E}[\lambda_k^i f^i(\boldsymbol{x}_{k+1}) - \lambda_k^i f^i(\boldsymbol{x}^*)] \leq \sum_{k=1}^{n}\frac{1}{2\eta_k}\mathbb{E}\left[\|\boldsymbol{x}_k - \boldsymbol{x}^*\|_2^2 - \|\boldsymbol{x}_{k+1} - \boldsymbol{x}^*\|_2^2\right]$$

$$+ DB\sigma m\sqrt{m}\sum_{k=1}^{n}(1-\alpha_k) + 6m^2 B^2 H\sigma\sqrt{m}\sum_{k=1}^{n}(1-\alpha_k)\eta_k + m^2 B^2(\sigma^2 + \sigma H + H^2)\sum_{k=1}^{n}\eta_k.$$

By the fact that $\eta_k \geq \eta_{k+1}$, we then bound the first term as

$$\sum_{k=1}^{n}\frac{1}{2\eta_k}\mathbb{E}\left[\|\boldsymbol{x}_k - \boldsymbol{x}^*\|_2^2 - \|\boldsymbol{x}_{k+1} - \boldsymbol{x}^*\|_2^2\right]$$

$$= \sum_{k=2}^{n}(\frac{1}{2\eta_k} - \frac{1}{2\eta_{k-1}})\mathbb{E}\left[\|\boldsymbol{x}_k - \boldsymbol{x}^*\|_2^2\right] + \frac{1}{2\eta_1}\mathbb{E}\left[\|\boldsymbol{x}_1 - \boldsymbol{x}^*\|_2^2\right] - \frac{1}{2\eta_n}\mathbb{E}\left[\|\boldsymbol{x}_{n+1} - \boldsymbol{x}^*\|_2^2\right]$$

$$\leq \sum_{k=2}^{n}(\frac{1}{2\eta_k} - \frac{1}{2\eta_{k-1}})D^2 + \frac{1}{2\eta_1}D^2 \leq \left(\frac{1}{2\eta_n} + \frac{1}{2\eta_1}\right)D^2.$$

Plug in the above result, we have

$$\sum_{k=1}^{n}\mathbb{E}[\sum_{i=1}^{m}\lambda_k^i f^i(\boldsymbol{x}_{k+1}) - \sum_{i=1}^{m}\lambda_k^i f^i(\boldsymbol{x}^*)] \leq \left(\frac{1}{2\eta_n} + \frac{1}{2\eta_1}\right)D^2 + m^{3/2}DB\sigma\sum_{k=1}^{n}(1-\alpha_k)$$

$$+ 6m^{5/2}B^2 H\sigma\sum_{k=1}^{n}(1-\alpha_k)\eta_k + m^2 B^2(\sigma^2 + \sigma H + H^2)\sum_{k=1}^{n}\eta_k.$$

We thus prove the lemma. $\qquad\square$

By observing Lemma 10 and the final version we need to proof, the only different is the compared optimum is $\boldsymbol{x}^*$ for all $k$ not the adaptive one $\boldsymbol{x}_k^* = \arg\min_{\boldsymbol{x}\in\mathcal{K}}\sum_{i=1}^{m}\lambda_k^i f^i(\boldsymbol{x})$, the following lemma measures such a gap.

**Lemma 11.** *Under the same assumption as Theorem 2, for any $\boldsymbol{x}^*$ in Pareto set, we have the following inequality for CRMOGM algorithm*

$$\mathbb{E}\left[\min_{\boldsymbol{x}^*\in\mathcal{K}}\sum_{k=1}^{n}\sum_{i=1}^{m}\lambda_k^i f^i(\boldsymbol{x}^*) - \sum_{k=1}^{n}\min_{\boldsymbol{x}_k^*\in\mathcal{K}}\sum_{i=1}^{m}\lambda_k^i f^i(\boldsymbol{x}_k^*)\right] \leq 2F\sum_{l=1}^{n}l(1-\alpha_k)\sum_{i=1}^{m}|\lambda_l^i - \hat{\lambda}_{l+1}^i|$$

*Proof.* Denote $\bar{\lambda}^i = \sum_{k=1}^{n}\lambda_k^i/n$. Denote $\boldsymbol{x}^* = \arg\min_{\boldsymbol{x}^*\in\mathcal{K}}\sum_{k=1}^{n}\sum_{i=1}^{m}\lambda_k^i f^i(\boldsymbol{x}^*)$ and $\boldsymbol{x}_k^* = \arg\min_{\boldsymbol{x}_k^*\in\mathcal{K}}\sum_{i=1}^{m}\lambda_k^i f^i(\boldsymbol{x}_k^*), k = 1, \ldots, n$. We first rearrange into

$$\mathbb{E}\left[\sum_{k=1}^{n}\sum_{i=1}^{m}\lambda_k^i f^i(\boldsymbol{x}^*) - \sum_{k=1}^{n}\sum_{i=1}^{m}\lambda_k^i f^i(\boldsymbol{x}_k^*)\right] = \mathbb{E}\left[\sum_{k=1}^{n}\sum_{i=1}^{m}\bar{\lambda}^i f^i(\boldsymbol{x}^*) - \sum_{k=1}^{n}\sum_{i=1}^{m}\lambda_k^i f^i(\boldsymbol{x}_k^*)\right].$$

Since $\boldsymbol{x}^*$ is the minimum for $\bar{\lambda}_k^i f^i(\boldsymbol{x})$, we then have

$$\mathbb{E}\left[\sum_{k=1}^{n}\sum_{i=1}^{m}\lambda_k^i f^i(\boldsymbol{x}^*) - \sum_{k=1}^{n}\sum_{i=1}^{m}\lambda_k^i f^i(\boldsymbol{x}_k^*)\right] \leq \mathbb{E}\left[\sum_{k=1}^{n}\sum_{i=1}^{m}\bar{\lambda}^i f^i(\boldsymbol{x}_k^*) - \sum_{k=1}^{n}\sum_{i=1}^{m}\lambda_k^i f^i(\boldsymbol{x}_k^*)\right]$$

$$= \mathbb{E}\left[\sum_{k=1}^{n}\sum_{i=1}^{m}(\bar{\lambda}^i - \lambda_k^i)f^i(\boldsymbol{x}_k^*)\right]$$

$$\leq \mathbb{E}\left[\sum_{k=1}^{n}\sum_{i=1}^{m}|\bar{\lambda}^i - \lambda_k^i||f^i(\boldsymbol{x}_k^*)|\right].$$

By the assumption that objective functions $f^1(\boldsymbol{x}_k), \ldots, f^m(\boldsymbol{x}_k) \leq F$ are all bounded for $k = 1, \ldots, n$, we get

$$\mathbb{E}\left[\sum_{k=1}^{n}\sum_{i=1}^{m} |\bar{\lambda}^i - \lambda_k^i||f^i(\boldsymbol{x}_k^*)|\right] \leq F\mathbb{E}\left[\sum_{k=1}^{n}\sum_{i=1}^{m} |\bar{\lambda}^i - \lambda_k^i|\right].$$

As we know that

$$\sum_{k=1}^{n}\sum_{i=1}^{m} |\bar{\lambda}^i - \lambda_k^i| = \sum_{k=1}^{n}\sum_{i=1}^{m} |\frac{1}{n}\sum_{t=1}^{n}(\lambda_t^i - \lambda_k^i)| = \frac{1}{n}\sum_{k=1}^{n}\sum_{i=1}^{m} |\sum_{t=1}^{n}(\lambda_t^i - \lambda_k^i)|$$

$$\leq \frac{1}{n}\sum_{i=1}^{m}\sum_{k=1}^{n}\sum_{t=1}^{n} |\lambda_t^i - \lambda_k^i| = \frac{1}{n}\sum_{i=1}^{m}\sum_{k=1}^{n}\sum_{t=1}^{k-1} |\lambda_t^i - \lambda_k^i| + \frac{1}{n}\sum_{i=1}^{m}\sum_{k=1}^{n}\sum_{t=k+1}^{n} |\lambda_t^i - \lambda_k^i|$$

$$\leq \frac{2}{n}\sum_{i=1}^{m}\sum_{k=1}^{n}\sum_{t=k}^{n}\sum_{l=k}^{t} |\lambda_l^i - \lambda_{l+1}^i| = \frac{2}{n}\sum_{i=1}^{m}\sum_{k=1}^{n}\sum_{l=k}^{n}\sum_{t=l}^{n} |\lambda_l^i - \lambda_{l+1}^i|$$

$$= \frac{2}{n}\sum_{i=1}^{m}\sum_{k=1}^{n}\sum_{l=k}^{n}(n-l+1)|\lambda_l^i - \lambda_{l+1}^i| \leq 2\sum_{i=1}^{m}\sum_{k=1}^{n}\sum_{l=k}^{n} |\lambda_l^i - \lambda_{l+1}^i|$$

$$= 2\sum_{i=1}^{m}\sum_{l=1}^{n}\sum_{k=1}^{l} |\lambda_l^i - \lambda_{l+1}^i| = 2\sum_{i=1}^{m}\sum_{l=1}^{n} l|\lambda_l^i - \lambda_{l+1}^i|$$

$$= 2\sum_{i=1}^{m}\sum_{l=1}^{n} l(1-\alpha_k)|\lambda_l^i - \hat{\lambda}_{l+1}^i| = 2\sum_{l=1}^{n} l(1-\alpha_{l+1})\sum_{i=1}^{m} |\lambda_l^i - \hat{\lambda}_{l+1}^i|.$$

The last two equation is by the definition of Algorithm 1. $\qquad\square$

We are now ready to proof Theorem 2. Combine Lemma 10 and Lemma 11, we have

$$\sum_{k=1}^{n}\mathbb{E}[\sum_{i=1}^{m}\lambda_k^i f^i(\boldsymbol{x}_{k+1}) - \min_{\boldsymbol{x}_k^* \in \mathcal{K}}\sum_{i=1}^{m}\lambda_k^i f^i(\boldsymbol{x}_k^*)] \leq \left(\frac{1}{2\eta_n} + \frac{1}{2\eta_1}\right)D^2 + DB\sigma m^{3/2}\sum_{k=1}^{n}(1-\alpha_k)$$

$$+ 6m^{5/2}B^2 H\sigma\sum_{k=1}^{n}(1-\alpha_k)\eta_k + m^2 B^2(\sigma^2 + \sigma H + H^2)\sum_{k=1}^{n}\eta_k + 2F\sum_{l=1}^{n} l(1-\alpha_{l+1})\sum_{i=1}^{m} |\lambda_l^i - \hat{\lambda}_{l+1}^i|.$$

By averaging the above, and the fact that $\eta_n \leq \eta_1$, we prove the theorem.

### E.5  Proof for Theorem 3 (Non-convex Convergence)

**Theorem 3.** *Under Assumption 1, 2, 3. For the sequence $\boldsymbol{x}_0, \boldsymbol{x}_1, \ldots, \boldsymbol{x}_n$ generated by Algorithm 1, we assume objective functions $f^1(\boldsymbol{x}_k), \ldots, f^m(\boldsymbol{x}_k) \leq F$ are all bounded for $k = 1, \ldots, n$. Set $0 \leq \eta_n \leq \ldots \leq \eta_1 \leq 1/mLB$ and $\alpha_k \in (0, 1]$ in Algorithm 1, it achieves*

$$\frac{1}{n}\sum_{k=1}^{n}\mathbb{E}\left[\min_{\boldsymbol{\lambda}_k^* \in S_m} \|\nabla\boldsymbol{F}(\boldsymbol{x}_k)\boldsymbol{\lambda}_k^*\|^2\right] \leq \frac{F}{n}\sum_{k=2}^{n}\frac{2}{\eta_k}(1-\alpha_k)\sum_{i=1}^{m} |\hat{\lambda}_k^i - \lambda_{k-1}^i| + \frac{2mBF}{n\eta_n}$$

$$+ \frac{4m^{5/2}B^2 H\sigma}{n}\sum_{k=1}^{n}(1-\alpha_k) + \frac{Lm^3 B^3\sigma^2}{2n}\sum_{k=1}^{n}\eta_k.$$

Similar to the convex case, to prove Theorem 3, it is sufficient to prove the bound for

$$\frac{1}{n}\sum_{k=1}^{n}\mathbb{E}\left[\|\nabla\boldsymbol{F}(\boldsymbol{x}_k)\boldsymbol{\lambda}_k\|^2\right], \tag{7}$$

where $\boldsymbol{\lambda}_k$ is generated by MOGM or CRMOGM algorithms. The reason is also similar to the convex case, which is that 7 upper bounds the left side in Theorem 3 because

$$\frac{1}{n}\sum_{k=1}^{n}\mathbb{E}\left[\|\nabla \boldsymbol{F}(\boldsymbol{x}_k)\boldsymbol{\lambda}_k\|^2\right] \geq \frac{1}{n}\sum_{k=1}^{n}\mathbb{E}\left[\frac{\|\nabla \boldsymbol{F}(\boldsymbol{x}_k)\boldsymbol{\lambda}_k\|^2}{(\sum_{i=1}^{m}\lambda_k^i)^2}\right]$$

$$= \frac{1}{n}\sum_{k=1}^{n}\mathbb{E}\left[\|\nabla \boldsymbol{F}(\boldsymbol{x}_k)\bar{\boldsymbol{\lambda}}_k\|^2\right] \geq \frac{1}{n}\sum_{k=1}^{n}\mathbb{E}\left[\min_{\boldsymbol{\lambda}_k^* \in S_m}\|\nabla \boldsymbol{F}(\boldsymbol{x}_k)\boldsymbol{\lambda}_k^*\|^2\right],$$

where $\bar{\boldsymbol{\lambda}}_k = \boldsymbol{\lambda}_k/(\sum_{i=1}^{m}\lambda_k^i)^2 \in S_m$. The first inequality of the above is due to Assumption 1 that $\sum_{i=1}^{m}\lambda_k^i \geq 1$. We next present necessary lemmas.

**Lemma 12.** *Under the same assumption as Theorem 3, for any gradient descent algorithm updated with composite gradient $\boldsymbol{G}_k\boldsymbol{\lambda}_k$, we have the following inequality*

$$\frac{\eta_k}{2}\mathbb{E}_{\boldsymbol{\xi}_k}\left[\left\|\sum_{i=1}^{m}\lambda_k^i\nabla f^i(\boldsymbol{x}_k)\right\|_2^2\right] \leq \mathbb{E}_{\boldsymbol{\xi}_k}[\sum_{i=1}^{m}\lambda_k^i f^i(\boldsymbol{x}_k) - \sum_{i=1}^{m}\lambda_k^i f^i(\boldsymbol{x}_{k+1})]$$

$$+ 4mBH\eta_k\sqrt{\mathbb{V}_{\boldsymbol{\xi}_k}[\boldsymbol{\lambda}_k]\sum_{i=1}^{m}\mathbb{V}_{\boldsymbol{\xi}_k}[\boldsymbol{g}^i(\boldsymbol{x}_k)]} + \frac{Lm^2B^3\eta_k^2}{2}\sum_{i=1}^{m}\mathbb{V}_{\boldsymbol{\xi}_k}(\boldsymbol{g}_i).$$

*Proof.* From the $L$-smoothness of each objective function, we have

$$\lambda_k^i f^i(\boldsymbol{x}_{k+1}) \leq \lambda_k^i \left(f^i(\boldsymbol{x}_k) + \eta_k\nabla f^i(\boldsymbol{x}_k)^\top \boldsymbol{d}_k + \frac{L\eta_k^2}{2}\|\boldsymbol{d}_k\|_2^2\right).$$

Sum up both side for $i = 1, \ldots, m$, and take the expectation on random variable $\boldsymbol{\xi}_k$, we can get

$$\mathbb{E}_{\boldsymbol{\xi}_k}[\sum_{i=1}^{m}\lambda_k^i f^i(\boldsymbol{x}_{k+1}) - \sum_{i=1}^{m}\lambda_k^i f^i(\boldsymbol{x}_k)] \leq \eta_k\mathbb{E}_{\boldsymbol{\xi}_k}\left[\left(\sum_{i=1}^{m}\lambda_k^i\nabla f^i(\boldsymbol{x}_k)\right)^\top \boldsymbol{d}_k\right] + \frac{L\eta_k^2}{2}\mathbb{E}_{\boldsymbol{\xi}_k}\left[\sum_{i=1}^{m}\lambda_k^i\|\boldsymbol{d}_k\|_2^2\right]$$

$$\leq \eta_k\mathbb{E}_{\boldsymbol{\xi}_k}\left[\left(\sum_{i=1}^{m}\lambda_k^i\nabla f^i(\boldsymbol{x}_k)\right)^\top \boldsymbol{d}_k\right] + \frac{LmB\eta_k^2}{2}\mathbb{E}_{\boldsymbol{\xi}_k}\left[\|\boldsymbol{d}_k\|_2^2\right].$$

The last inequality is by the Assumption that $\lambda_k^i \leq B$, then $\mathbb{E}_{\boldsymbol{\xi}_k}\left[\sum_{i=1}^{m}\lambda_k^i\|\boldsymbol{d}_k\|_2^2\right] \leq \mathbb{E}_{\boldsymbol{\xi}_k}\left[\sum_{i=1}^{m}B\|\boldsymbol{d}_k\|_2^2\right] = mB\mathbb{E}_{\boldsymbol{\xi}_k}\left[\|\boldsymbol{d}_k\|_2^2\right]$. From the result of Lemma 7, we can bound the first term and obtain

$$\mathbb{E}_{\boldsymbol{\xi}_k}[\sum_{i=1}^{m}\lambda_k^i f^i(\boldsymbol{x}_{k+1}) - \sum_{i=1}^{m}\lambda_k^i f^i(\boldsymbol{x}_k)]$$

$$\leq 2\eta_k mBH\sqrt{\mathbb{V}_{\boldsymbol{\xi}_k}[\boldsymbol{\lambda}_k]\sum_{i=1}^{m}\mathbb{V}_{\boldsymbol{\xi}_k}[\boldsymbol{g}^i(\boldsymbol{x}_k)]} - \eta_k\mathbb{E}_{\boldsymbol{\xi}_k}\left[\left\|\sum_{i=1}^{m}\lambda_k^i\nabla f^i(\boldsymbol{x}_k)\right\|_2^2\right] + \frac{LmB\eta_k^2}{2}\mathbb{E}_{\boldsymbol{\xi}_k}\left[\|\boldsymbol{d}_k\|_2^2\right].$$

Then, adopting the result from Lemma 8, we know that

$$\mathbb{E}_{\boldsymbol{\xi}_k}[\sum_{i=1}^{m}\lambda_k^i f^i(\boldsymbol{x}_{k+1}) - \sum_{i=1}^{m}\lambda_k^i f^i(\boldsymbol{x}_k)] \leq (2\eta_k mBH + 2\eta_k^2 Lm^2B^2H)\sqrt{\mathbb{V}_{\boldsymbol{\xi}_k}[\boldsymbol{\lambda}_k]\sum_{i=1}^{m}\mathbb{V}_{\boldsymbol{\xi}_k}[\boldsymbol{g}^i(\boldsymbol{x}_k)]}$$

$$+ \left(\frac{LmB\eta_k^2}{2} - \eta_k\right)\mathbb{E}_{\boldsymbol{\xi}_k}\left[\left\|\sum_{i=1}^{m}\lambda_k^i\nabla f^i(\boldsymbol{x}_k)\right\|_2^2\right] + \frac{Lm^2B^3\eta_k^2}{2}\sum_{i=1}^{m}\mathbb{V}_{\boldsymbol{\xi}_k}(\boldsymbol{g}_i).$$

By the fact that $\eta_k \leq 1/LmB$, we further have

$$\mathbb{E}_{\boldsymbol{\xi}_k}[\sum_{i=1}^{m} \lambda_k^i f^i(\boldsymbol{x}_{k+1}) - \sum_{i=1}^{m} \lambda_k^i f^i(\boldsymbol{x}_k)] \leq 4mBH\eta_k \sqrt{\mathbb{V}_{\boldsymbol{\xi}_k}[\lambda_k] \sum_{i=1}^{m} \mathbb{V}_{\boldsymbol{\xi}_k}[\boldsymbol{g}^i(\boldsymbol{x}_k)]}$$

$$- \frac{\eta_k}{2} \mathbb{E}_{\boldsymbol{\xi}_k} \left[ \left\| \sum_{i=1}^{m} \lambda_k^i \nabla f^i(\boldsymbol{x}_k) \right\|_2^2 \right] + \frac{Lm^2 B^3 \eta_k^2}{2} \sum_{i=1}^{m} \mathbb{V}_{\boldsymbol{\xi}_k}(\boldsymbol{g}_i).$$

By rearrangement, we therefore have

$$\frac{\eta_k}{2} \mathbb{E}_{\boldsymbol{\xi}_k} \left[ \left\| \sum_{i=1}^{m} \lambda_k^i \nabla f^i(\boldsymbol{x}_k) \right\|_2^2 \right] \leq \mathbb{E}_{\boldsymbol{\xi}_k}[\sum_{i=1}^{m} \lambda_k^i f^i(\boldsymbol{x}_k) - \sum_{i=1}^{m} \lambda_k^i f^i(\boldsymbol{x}_{k+1})]$$

$$+ 4mBH\eta_k \sqrt{\mathbb{V}_{\boldsymbol{\xi}_k}[\lambda_k] \sum_{i=1}^{m} \mathbb{V}_{\boldsymbol{\xi}_k}[\boldsymbol{g}^i(\boldsymbol{x}_k)]} + \frac{Lm^2 B^3 \eta_k^2}{2} \sum_{i=1}^{m} \mathbb{V}_{\boldsymbol{\xi}_k}(\boldsymbol{g}_i).$$

Take expectation of $\xi_1, \xi_2, \ldots, \xi_n$ on the both sides, we finally get

$$\frac{\eta_k}{2} \mathbb{E} \left[ \left\| \sum_{i=1}^{m} \lambda_k^i \nabla f^i(\boldsymbol{x}_k) \right\|_2^2 \right] \leq \mathbb{E}[\sum_{i=1}^{m} \lambda_k^i f^i(\boldsymbol{x}_k) - \sum_{i=1}^{m} \lambda_k^i f^i(\boldsymbol{x}_{k+1})]$$

$$+ 4mBH\eta_k \sqrt{\mathbb{V}[\lambda_k] \sum_{i=1}^{m} \mathbb{V}[\boldsymbol{g}^i(\boldsymbol{x}_k)]} + \frac{Lm^2 B^3 \eta_k^2}{2} \sum_{i=1}^{m} \mathbb{V}(\boldsymbol{g}_i).$$

□

**Lemma 13.** *For CRMOGM algorithm, and under the same assumption with Theorem* 3, *we have*

$$\sum_{k=1}^{n} \frac{1}{\eta_k} \mathbb{E}[\sum_{i=1}^{m} \lambda_k^i f^i(\boldsymbol{x}_k) - \sum_{i=1}^{m} \lambda_k^i f^i(\boldsymbol{x}_{k+1})] \leq F \sum_{k=2}^{n} \frac{1}{\eta_k} \mathbb{E}[\sum_{i=1}^{m} (1-\alpha_k)|\hat{\lambda}_k^i - \lambda_{k-1}^i|] + \frac{2mBF}{\eta_n}.$$

*Proof.* First, we can decompose the left side as

$$\sum_{k=1}^{n} \frac{1}{\eta_k} \mathbb{E}[\sum_{i=1}^{m} \lambda_k^i f^i(\boldsymbol{x}_k) - \sum_{i=1}^{m} \lambda_k^i f^i(\boldsymbol{x}_{k+1})]$$

$$= \sum_{k=2}^{n} \mathbb{E}[\frac{1}{\eta_k} \sum_{i=1}^{m} \lambda_k^i f^i(\boldsymbol{x}_k) - \frac{1}{\eta_{k-1}} \sum_{i=1}^{m} \lambda_{k-1}^i f^i(\boldsymbol{x}_k)] + \mathbb{E}[\frac{1}{\eta_1} \sum_{i=1}^{m} \lambda_1^i f^i(\boldsymbol{x}_1) - \frac{1}{\eta_n} \sum_{i=1}^{m} \lambda_n^i f^i(\boldsymbol{x}_{n+1})]$$

By the definition of CRMOGM algorithm, we know that $\lambda_k^i = \alpha_k \lambda_{k-1}^i + (1-\alpha_k)\hat{\lambda}_k^i, i = 1, \ldots, m$, where $\{\hat{\lambda}_k^i\}_{i=1}^{m}$ are calculated by MOGM algorithms. Then we have the following decomposition for the first term

$$\sum_{k=2}^{n} \mathbb{E}[\frac{1}{\eta_k} \sum_{i=1}^{m} \lambda_k^i f^i(\boldsymbol{x}_k) - \frac{1}{\eta_{k-1}} \sum_{i=1}^{m} \lambda_{k-1}^i f^i(\boldsymbol{x}_k)]$$

$$= \sum_{k=2}^{n} \mathbb{E}[\frac{1}{\eta_k} \sum_{i=1}^{m} \lambda_k^i f^i(\boldsymbol{x}_k) - \frac{1}{\eta_k} \sum_{i=1}^{m} \lambda_{k-1}^i f^i(\boldsymbol{x}_k) + \frac{1}{\eta_k} \sum_{i=1}^{m} \lambda_{k-1}^i f^i(\boldsymbol{x}_k) - \frac{1}{\eta_{k-1}} \sum_{i=1}^{m} \lambda_{k-1}^i f^i(\boldsymbol{x}_k)]$$

$$= \sum_{k=2}^{n} \mathbb{E}[\sum_{i=1}^{m} \frac{1}{\eta_k} (\lambda_k^i - \lambda_{k-1}^i) f^i(\boldsymbol{x}_k)] + \sum_{k=2}^{n} \mathbb{E}[(\frac{1}{\eta_k} - \frac{1}{\eta_{k-1}}) \sum_{i=1}^{m} \lambda_{k-1}^i f^i(\boldsymbol{x}_k)]$$

$$= \sum_{k=2}^{n} \frac{1}{\eta_k} \mathbb{E}[\sum_{i=1}^{m} (1-\alpha_k)(\hat{\lambda}_k^i - \lambda_{k-1}^i) f^i(\boldsymbol{x}_k)] + \sum_{k=2}^{n} \mathbb{E}[(\frac{1}{\eta_k} - \frac{1}{\eta_{k-1}}) \sum_{i=1}^{m} \lambda_{k-1}^i f^i(\boldsymbol{x}_k)]$$

Combining the above, and by the assumption that $|f^i(\boldsymbol{x}_k)| \le F, \lambda_k^i, \hat{\lambda}_k^i \in [0, B]$ as well as the nonincreasing for $\eta_k$, we therefore have

$$
\sum_{k=1}^{n} \frac{1}{\eta_k} \mathbb{E}[\sum_{i=1}^{m} \lambda_k^i f^i(\boldsymbol{x}_k) - \sum_{i=1}^{m} \lambda_k^i f^i(\boldsymbol{x}_{k+1})]
$$

$$
\le \sum_{k=2}^{n} \frac{1}{\eta_k} \mathbb{E}[\sum_{i=1}^{m}(1-\alpha_k)|\hat{\lambda}_k^i - \lambda_{k-1}^i||f^i(\boldsymbol{x}_k)|] + \sum_{k=2}^{n} \mathbb{E}[(\frac{1}{\eta_k} - \frac{1}{\eta_{k-1}})\sum_{i=1}^{m} \lambda_{k-1}^i |f^i(\boldsymbol{x}_k)|]
$$

$$
+ \mathbb{E}[\frac{1}{\eta_1} \sum_{i=1}^{m} \lambda_1^i |f^i(\boldsymbol{x}_1)| + \frac{1}{\eta_n} \sum_{i=1}^{m} \lambda_n^i |f^i(\boldsymbol{x}_{n+1})|]
$$

$$
\le F \sum_{k=2}^{n} \frac{1}{\eta_k} \mathbb{E}[\sum_{i=1}^{m}(1-\alpha_k)|\hat{\lambda}_k^i - \lambda_{k-1}^i|] + mBF \sum_{k=2}^{n} \mathbb{E}[(\frac{1}{\eta_k} - \frac{1}{\eta_{k-1}}) + \frac{mBF}{\eta_1} + \frac{mBF}{\eta_n}
$$

$$
\le F \sum_{k=2}^{n} \frac{1}{\eta_k} \mathbb{E}[\sum_{i=1}^{m}(1-\alpha_k)|\hat{\lambda}_k^i - \lambda_{k-1}^i|] + \frac{2mBF}{\eta_n}.
$$

$\square$

Now, we can present the final proof. From Lemma 2, we know that $\mathbb{V}_{\boldsymbol{\xi}_k}[\lambda_k] \le m^2 B^2 (1-\alpha_k)^2$. From Assumption 2, we know that $\mathbb{V}_{\boldsymbol{\xi}_k}(\boldsymbol{g}^i(\boldsymbol{x}_k)) \le \sigma^2$. Then plug into Lemma 12, we get

$$
\mathbb{E}\left[\left\|\sum_{i=1}^{m} \lambda_k^i \nabla f^i(\boldsymbol{x}_k)\right\|_2^2\right]
$$

$$
\le \frac{2}{\eta_k} \mathbb{E}[\sum_{i=1}^{m} \lambda_k^i f^i(\boldsymbol{x}_k) - \sum_{i=1}^{m} \lambda_k^i f^i(\boldsymbol{x}_{k+1})] + 8m^{5/2} B^2 H\sigma(1-\alpha_k) + Lm^3 B^3 \sigma^2 \eta_k.
$$

The above is divided with $\eta_k/2$ for the both side. Then, sum up the above inequality from $k=1$ to $k=n$ and take the result from Lemma 13, we have

$$
\sum_{k=1}^{n} \mathbb{E}\left[\left\|\sum_{i=1}^{m} \lambda_k^i \nabla f^i(\boldsymbol{x}_k)\right\|_2^2\right]
$$

$$
\le \sum_{k=1}^{n} \frac{2}{\eta_k} \mathbb{E}[\sum_{i=1}^{m} \lambda_k^i f^i(\boldsymbol{x}_k) - \sum_{i=1}^{m} \lambda_k^i f^i(\boldsymbol{x}_{k+1})] + 4m^{5/2} B^2 H\sigma \sum_{k=1}^{n}(1-\alpha_k) + \frac{Lm^3 B^3 \sigma^2}{2} \sum_{k=1}^{n} \eta_k
$$

$$
\le F \sum_{k=2}^{n} \frac{2}{\eta_k} \mathbb{E}[\sum_{i=1}^{m}(1-\alpha_k)|\hat{\lambda}_k^i - \lambda_{k-1}^i|] + \frac{2mBF}{\eta_n} + 4m^{5/2} B^2 H\sigma \sum_{k=1}^{n}(1-\alpha_k) + \frac{Lm^3 B^3 \sigma^2}{2} \sum_{k=1}^{n} \eta_k.
$$

By averaging the above inequality, we prove the theorem.

### E.6 Proof of Corollary 1

**Corollary 1.** *(a) Set $1/mLB \ge \eta_k = O(1/\sqrt{k}), \alpha_k = \max\{1-\eta_k/\eta_1, 1-\eta_k/(\eta_1 k \sum_{i=1}^{m}|\lambda_{k-1}^i - \hat{\lambda}_k^i|)\}$ in Theorem 2, Algorithm 1 converges with $O(n^{1/2})$ rates in the convex case. (b) Set $1/mLB \ge \eta_k = O(1/\sqrt{k}), \alpha_k = \max\{1-\eta_k/\eta_1, 1-\eta_k/(\eta_1 \sqrt{k} \sum_{i=1}^{m}|\lambda_{k-1}^i - \hat{\lambda}_k^i|)\}$ in Theorem 3, Algorithm 1 $O(n^{1/4})$ rates in the non-convex case.*

*Proof.* For the convex case **(a)**, $\alpha_k = \max\{1 - \eta_k/\eta_1, 1 - \eta_k/(\eta_1 k \sum_{i \in [m]} |\lambda_{k-1}^i - \hat{\lambda}_k^i|)\}$ means $1 - \alpha_k = \min\{\eta_k/\eta_1, \eta_k/(\eta_1 k \sum_{i \in [m]} |\lambda_{k-1}^i - \hat{\lambda}_k^i|)\}$. Plug in Theorem 2, we have

$$\frac{1}{n} \sum_{k=1}^n \mathbb{E}\left[\max_{\boldsymbol{x}_k^* \in \mathcal{K}} \min_{\boldsymbol{\lambda}_k^* \in S_m} (\boldsymbol{\lambda}_k^{*\top} \boldsymbol{F}(\boldsymbol{x}_k) - \boldsymbol{\lambda}_k^{*\top} \boldsymbol{F}(\boldsymbol{x}_k^*))\right]$$

$$\leq \left(\frac{1}{2\eta_n} + \frac{1}{2\eta_1}\right) \frac{D^2}{n} + \frac{DB\sigma m^{3/2}}{n} \sum_{k=1}^n (1 - \alpha_k) + \frac{6m^{5/2}B^2H\sigma}{n} \sum_{k=1}^n (1 - \alpha_k)\eta_k$$

$$+ \frac{m^2 B^2(\sigma^2 + \sigma H + H^2)}{n} \sum_{k=1}^n \eta_k + \frac{2}{n} \sum_{l=1}^n l(1 - \alpha_{l+1}) \sum_{i=1}^m |\lambda_l^i - \hat{\lambda}_{l+1}^i|$$

$$\leq O(\frac{1}{n}) + \sum_{k=1}^n O(\frac{1}{\sqrt{k}}) + \sum_{k=1}^n O(\frac{1}{k}) + + \sum_{k=1}^n O(\frac{1}{\sqrt{k}}) + + \sum_{k=1}^n O(\frac{1}{\sqrt{k}}) \leq O(\sqrt{n}).$$

For the non-convex case **(b)**, $\alpha_k = \max\{1 - \eta_k/\eta_1, 1 - \eta_k/(\eta_1 \sqrt{k} \max_{i \in [m]} |\lambda_{k-1}^i - \hat{\lambda}_k^i|)\}$ means $1 - \alpha_k = \min\{\eta_k/\eta_1, \eta_k/(\eta_1 \sqrt{k} \max_{i \in [m]} |\lambda_{k-1}^i - \hat{\lambda}_k^i|)\}$. Plug in Theorem 2, we have

$$\frac{1}{n} \sum_{k=1}^n \mathbb{E}\left[\min_{\boldsymbol{\lambda}_k^* \in S_m} \|\nabla \boldsymbol{F}(\boldsymbol{x}_k)\boldsymbol{\lambda}_k^*\|^2\right] \leq \frac{F}{n} \sum_{k=2}^n \frac{2}{\eta_k} \mathbb{E}[\sum_{i=1}^m (1 - \alpha_k)|\hat{\lambda}_k^i - \lambda_{k-1}^i|] + \frac{2mBF}{n\eta_n}$$

$$+ \frac{4m^{5/2}B^2H\sigma}{n} \sum_{k=1}^n (1 - \alpha_k) + \frac{Lm^3 B^3 \sigma^2}{2n} \sum_{k=1}^n \eta_k$$

$$\leq \sum_{k=1}^n O(\frac{1}{\sqrt{k}}) + O(\frac{1}{n}) + \sum_{k=1}^n O(\frac{1}{\sqrt{k}}) + \sum_{k=1}^n O(\frac{1}{\sqrt{k}}) \leq O(\sqrt{n}).$$

□

***Remark 1.*** Also, it is not difficult to verify that setting $\alpha_k = \max\{1 - \eta_k/\eta_1, 1 - \eta_k/(\eta_1 k \sum_{i \in [m]} |\lambda_{k-1}^i - \hat{\lambda}_k^i|)\}$ also lead to the same convergence in the non-convex case. For simplicity, we present the unified version in the main text, and provide the complete version in the Appendix.

***Remark 2.*** Since multi-objective optimization could reduce to single-objective optimization if the multiple objectives are the same, the convergence rate of stochastic MOO will be no better than single-objective SGD. Note that our convergence rates align with those in single-objective SGD [5, 2], which suggests that our bounds are tight in order. In comparison with stochastic MOGM, we have demonstrated clearly in this paper that stochastic MOGM algorithms fail to converge to the Pareto optimal/critical points in Section 4, so it would be less meaningful to compare them. As for comparing with full-batch MOGM, the comparison is similar with GD and SGD.

### E.7    Discussion on Theorem 2 and Theorem 3

**Average scheme.** The averaging scheme for Theorem 2 and Theorem 3 can be transformed to traditional ones. It just needs to use a typical uniform sampling technique in stochastic optimization [4, 24, 5, 20] to modify the algorithmic output. Specifically, the output is changed from $\boldsymbol{x}_n$ to $\bar{\boldsymbol{x}}_n$ that is uniformly sampled in $\{\boldsymbol{x}_1, \boldsymbol{x}_2, \ldots, \boldsymbol{x}_n\}$. Then, by this randomness and the linearity of expectation, we can know that

$$\mathbb{E}\left[\max_{\boldsymbol{x}^* \in \mathcal{K}} \min_{\boldsymbol{\lambda}^* \in S_m} (\boldsymbol{\lambda}^{*\top} \boldsymbol{F}(\bar{\boldsymbol{x}}_n) - \boldsymbol{\lambda}^{*\top} \boldsymbol{F}(\boldsymbol{x}^*))\right] = \mathbb{E}\left[\frac{1}{n} \sum_{k=1}^n \max_{\boldsymbol{x}_k^* \in \mathcal{K}} \min_{\boldsymbol{\lambda}_k^* \in S_m} (\boldsymbol{\lambda}_k^{*\top} \boldsymbol{F}(\boldsymbol{x}_k) - \boldsymbol{\lambda}_k^{*\top} \boldsymbol{F}(\boldsymbol{x}_k^*))\right]$$

$$= \frac{1}{n} \sum_{k=1}^n \mathbb{E}\left[\max_{\boldsymbol{x}_k^* \in \mathcal{K}} \min_{\boldsymbol{\lambda}_k^* \in S_m} (\boldsymbol{\lambda}_k^{*\top} \boldsymbol{F}(\boldsymbol{x}_k) - \boldsymbol{\lambda}_k^{*\top} \boldsymbol{F}(\boldsymbol{x}_k^*))\right].$$

and

$$\mathbb{E}\left[\min_{\boldsymbol{\lambda}^* \in S_m} \|\nabla \boldsymbol{F}(\bar{\boldsymbol{x}}_n)\boldsymbol{\lambda}^*\|^2\right] = \mathbb{E}\left[\frac{1}{n}\sum_{k=1}^{n}\min_{\boldsymbol{\lambda}_k^* \in S_m}\|\nabla \boldsymbol{F}(\boldsymbol{x}_k)\boldsymbol{\lambda}_k^*\|^2\right] = \frac{1}{n}\sum_{k=1}^{n}\mathbb{E}\left[\min_{\boldsymbol{\lambda}_k^* \in S_m}\|\nabla \boldsymbol{F}(\boldsymbol{x}_k)\boldsymbol{\lambda}_k^*\|^2\right].$$

Therefore, we recover the traditional convergence bound for $\bar{\boldsymbol{x}}_n$.

**Assumptions of bounded gradient.** The assumption of bounded gradients appears very widely in related literature on stochastic optimization. Specifically, in the convex setting, the bounded gradient assumption is used to derive the convergence bound for various famous optimizers, such as Adam (Theorem 4.1 in [22]), Adagrad (Theorem 5 in [6]), AMSGrad (Theorem 4 in [36]), etc. In the non-convex setting, it helps to analyze the stochastic convergence of momentum (Theorem 1 & 2 in [44], Assumption 3 in [4], Assumption 2 in [45], and Assumption 1(ii) in [21]). Additionally, in practice with deep learning, there are many techniques that are designed to avoid gradient explosion, such as weight regularization [35] and gradient clipping [47]. These techniques ensure the boundness of the gradient in practice, which validates this assumption. Furthermore, our theoretical results only require that $\|\nabla f^i(x_k)\| \le H$ for $i = 1, \ldots, m$ and $k = 1, \ldots, n$, i.e., the gradients along the optimization path are bounded, which is also supported by our experiments.

**Assumptions of bounded function.** The assumption of bounded functions has been used in recent studies on non-convex stochastic optimization, such as analyzing SGD (Assumption 2 in [7]), and analyzing SGD+momentum (Assumption in Section 3 of [25]). Actually, this assumption can be further relaxed in our analysis. Specifically, in the convex setting, it can be relaxed to that the function values of Pareto optimal solutions are bounded, i.e., $|f^i(\boldsymbol{x}^*)| \le F, i = 1, \ldots, m$, if $\boldsymbol{x}^*$ is Pareto optimal. The reason is that we only require this condition in Lemma 11. The relaxed assumption is surely satisfied since the optimal solutions naturally have bounded function values. In the non-convex setting, note that in the related papers analyzing SGD (Remark 2 in the analysis of SGD [5]) as well as MOGM (Theorem 3.1 in the analysis of MGDA [9]) with non-convex functions, it is commonly assumed that the initial function values $f^i(\boldsymbol{x}_1), i = 1, \ldots, m$ are bounded. Essentially, our analysis only requires that $f^i(\boldsymbol{x}_k), i = 1, \ldots, m$ are bounded for $k = 1, \ldots, n$, since we only use it in Lemma 13. This assumption is mild in practice as long as the function values of iterates along the practical training path are bounded. In fact, all our experiments indicate that the function values are well located in a bounded regime.

# F  Convergence Analysis for Strongly Convex Functions

**Theorem 6.** *Under Assumption 1, 2, 3. For the sequence $\boldsymbol{x}_0, \boldsymbol{x}_1, \ldots, \boldsymbol{x}_n$ generated by Algorithm 1, we assume objective functions $f^1(\boldsymbol{x}), \ldots, f^m(\boldsymbol{x})$ are all $\mu$-strongly convex and bounded as $f^1(\boldsymbol{x}), \ldots, f^m(\boldsymbol{x}) \le F$, and the distance from sequence to Pareto set is bounded, i.e., $\|\boldsymbol{x}_k - \boldsymbol{x}^*\| \le D$ for $\boldsymbol{x}^*$ in Pareto set. Set $0 \le \eta_n \le \ldots \le \eta_1 \le 1/mLB, \alpha_k \in (0,1]$ for $k = 1, \ldots, n$ in Algorithm 1, it achieves*

$$\mathbb{E}\left[\sum_{i=1}^{m}\lambda_{k+1}^i f^i(\boldsymbol{x}_{k+1}) - \sum_{i=1}^{m}\lambda_{k+1}^i f^i(\boldsymbol{x}_{k+1}^*)\right] \le (1 - \eta_k\mu)\mathbb{E}\left[\sum_{i=1}^{m}\lambda_k^i f^i(\boldsymbol{x}_k) - \sum_{i=1}^{m}\lambda_k^i f^i(\boldsymbol{x}_k^*)\right]$$
$$+ 4m^{5/2}B^2 H\eta_k(1 - \alpha_k) + \frac{Lm^3B^3\sigma^2\eta_k^2}{2} + 2F(1 - \alpha_{k+1})\sum_{i=1}^{m}|\hat{\lambda}_{k+1}^i - \lambda_k^i|.$$

*Proof.* By the property of $\mu$-strongly convex ((4.12) from [2]), we have

$$2\mu\left(\sum_{i=1}^{m}\lambda_k^i f^i(\boldsymbol{x}_k) - \sum_{i=1}^{m}\lambda_k^i f^i(\boldsymbol{x}_k^*)\right) \le \left\|\sum_{i=1}^{m}\lambda_k^i \nabla f^i(\boldsymbol{x}_k)\right\|_2^2,$$

where $\boldsymbol{x}_k^* = \arg\min_{\boldsymbol{x}} \sum_{i=1}^m \lambda_k^i f^i(\boldsymbol{x})$. Recall Lemma 12, we know that

$$\frac{\eta_k}{2}\mathbb{E}_{\boldsymbol{\xi}_k}\left[\left\|\sum_{i=1}^m \lambda_k^i \nabla f^i(\boldsymbol{x}_k)\right\|_2^2\right] \leq \mathbb{E}_{\boldsymbol{\xi}_k}\left[\sum_{i=1}^m \lambda_k^i f^i(\boldsymbol{x}_k) - \sum_{i=1}^m \lambda_k^i f^i(\boldsymbol{x}_{k+1})\right]$$
$$+ 4mBH\eta_k\sqrt{\mathbb{V}_{\boldsymbol{\xi}_k}[\lambda_k]\sum_{i=1}^m \mathbb{V}_{\boldsymbol{\xi}_k}[\boldsymbol{g}^i(\boldsymbol{x}_k)]} + \frac{Lm^2B^3\eta_k^2}{2}\sum_{i=1}^m \mathbb{V}_{\boldsymbol{\xi}_k}(\boldsymbol{g}_i).$$

By combining the above, we get

$$\eta_k\mu\mathbb{E}_{\boldsymbol{\xi}_k}\left[\sum_{i=1}^m \lambda_k^i f^i(\boldsymbol{x}_k) - \sum_{i=1}^m \lambda_k^i f^i(\boldsymbol{x}_k^*)\right] \leq \mathbb{E}_{\boldsymbol{\xi}_k}\left[\sum_{i=1}^m \lambda_k^i f^i(\boldsymbol{x}_k) - \sum_{i=1}^m \lambda_k^i f^i(\boldsymbol{x}_{k+1})\right]$$
$$+ 4mBH\eta_k\sqrt{\mathbb{V}_{\boldsymbol{\xi}_k}[\lambda_k]\sum_{i=1}^m \mathbb{V}_{\boldsymbol{\xi}_k}[\boldsymbol{g}^i(\boldsymbol{x}_k)]} + \frac{Lm^2B^3\eta_k^2}{2}\sum_{i=1}^m \mathbb{V}_{\boldsymbol{\xi}_k}(\boldsymbol{g}_i).$$

By rearranging the inequality, we further have

$$\mathbb{E}_{\boldsymbol{\xi}_k}\left[\sum_{i=1}^m \lambda_k^i f^i(\boldsymbol{x}_{k+1}) - \sum_{i=1}^m \lambda_k^i f^i(\boldsymbol{x}_k^*)\right] \leq (1-\eta_k\mu)\mathbb{E}_{\boldsymbol{\xi}_k}\left[\sum_{i=1}^m \lambda_k^i f^i(\boldsymbol{x}_k) - \sum_{i=1}^m \lambda_k^i f^i(\boldsymbol{x}_k^*)\right]$$
$$+ 4mBH\eta_k\sqrt{\mathbb{V}_{\boldsymbol{\xi}_k}[\lambda_k]\sum_{i=1}^m \mathbb{V}_{\boldsymbol{\xi}_k}[\boldsymbol{g}^i(\boldsymbol{x}_k)]} + \frac{Lm^2B^3\eta_k^2}{2}\sum_{i=1}^m \mathbb{V}_{\boldsymbol{\xi}_k}(\boldsymbol{g}_i).$$

By the optimality of $\boldsymbol{x}_k^*$, we know that $\sum_{i=1}^m \lambda_k^i f^i(\boldsymbol{x}_k^*) \leq \sum_{i=1}^m \lambda_k^i f^i(\boldsymbol{x}_{k+1}^*)$. Therefore, we can get

$$\mathbb{E}_{\boldsymbol{\xi}_k}\left[\sum_{i=1}^m \lambda_k^i f^i(\boldsymbol{x}_{k+1}) - \sum_{i=1}^m \lambda_k^i f^i(\boldsymbol{x}_{k+1}^*)\right] \leq (1-\eta_k\mu)\mathbb{E}_{\boldsymbol{\xi}_k}\left[\sum_{i=1}^m \lambda_k^i f^i(\boldsymbol{x}_k) - \sum_{i=1}^m \lambda_k^i f^i(\boldsymbol{x}_k^*)\right]$$
$$+ 4mBH\eta_k\sqrt{\mathbb{V}_{\boldsymbol{\xi}_k}[\lambda_k]\sum_{i=1}^m \mathbb{V}_{\boldsymbol{\xi}_k}[\boldsymbol{g}^i(\boldsymbol{x}_k)]} + \frac{Lm^2B^3\eta_k^2}{2}\sum_{i=1}^m \mathbb{V}_{\boldsymbol{\xi}_k}(\boldsymbol{g}_i).$$

Further by decomposing, we can obtain

$$\sum_{i=1}^m \lambda_{k+1}^i f^i(\boldsymbol{x}_{k+1}) - \sum_{i=1}^m \lambda_{k+1}^i f^i(\boldsymbol{x}_{k+1}^*)$$
$$= \sum_{i=1}^m \lambda_k^i f^i(\boldsymbol{x}_{k+1}) - \sum_{i=1}^m \lambda_k^i f^i(\boldsymbol{x}_{k+1}^*) - \sum_{i=1}^m (\lambda_k^i - \lambda_{k+1}^i)f^i(\boldsymbol{x}_{k+1}) + \sum_{i=1}^m (\lambda_k^i - \lambda_{k+1}^i)f^i(\boldsymbol{x}_{k+1}^*).$$

By the definition of Algorithm 1 and the boundness of gradient function. We know that $\lambda_{k+1}^i - \lambda_k^i = (1-\alpha_{k+1})(\hat{\lambda}_{k+1}^i - \lambda_k^i)$ and $|f^i(\boldsymbol{x}_k)| \leq F, |f^i(\boldsymbol{x}_k^*)| \leq F$. Then, we can easily get that $-(\lambda_k^i - \lambda_{k+1}^i)f^i(\boldsymbol{x}_{k+1}) \leq F(1-\alpha_{k+1})|\hat{\lambda}_{k+1}^i - \lambda_k^i|$ and $(\lambda_k^i - \lambda_{k+1}^i)f^i(\boldsymbol{x}_{k+1}^*) \leq F(1-\alpha_{k+1})|\hat{\lambda}_{k+1}^i - \lambda_k^i|$. Thus, we have

$$\sum_{i=1}^m \lambda_{k+1}^i f^i(\boldsymbol{x}_{k+1}) - \sum_{i=1}^m \lambda_{k+1}^i f^i(\boldsymbol{x}_{k+1}^*)$$
$$\leq \sum_{i=1}^m \lambda_k^i f^i(\boldsymbol{x}_{k+1}) - \sum_{i=1}^m \lambda_k^i f^i(\boldsymbol{x}_{k+1}^*) + 2F\sum_{i=1}^m (1-\alpha_{k+1})|\hat{\lambda}_{k+1}^i - \lambda_k^i|.$$

Then combining the above, we now obtain

$$\mathbb{E}_{\boldsymbol{\xi}_k}\left[\sum_{i=1}^m \lambda_{k+1}^i f^i(\boldsymbol{x}_{k+1}) - \sum_{i=1}^m \lambda_{k+1}^i f^i(\boldsymbol{x}_{k+1}^*)\right] \leq (1-\eta_k\mu)\mathbb{E}_{\boldsymbol{\xi}_k}\left[\sum_{i=1}^m \lambda_k^i f^i(\boldsymbol{x}_k) - \sum_{i=1}^m \lambda_k^i f^i(\boldsymbol{x}_k^*)\right]$$
$$+ 4mBH\eta_k\sqrt{\mathbb{V}_{\boldsymbol{\xi}_k}[\lambda_k]\sum_{i=1}^m \mathbb{V}_{\boldsymbol{\xi}_k}[\boldsymbol{g}^i(\boldsymbol{x}_k)]} + \frac{Lm^2B^3\eta_k^2}{2}\sum_{i=1}^m \mathbb{V}_{\boldsymbol{\xi}_k}(\boldsymbol{g}_i) + 2F\sum_{i=1}^m (1-\alpha_{k+1})|\hat{\lambda}_{k+1}^i - \lambda_k^i|.$$

From Lemma 2, we know that $\mathbb{V}_{\boldsymbol{\xi}_k}[\lambda_k] \leq m^2 B^2 (1-\alpha_k)^2$. From Assumption 2, we know that $\mathbb{V}_{\boldsymbol{\xi}_k}(\boldsymbol{g}^i(\boldsymbol{x}_k)) \leq m\sigma^2$. Then plug into the above, we get

$$\mathbb{E}_{\boldsymbol{\xi}_k}\left[\sum_{i=1}^m \lambda_{k+1}^i f^i(\boldsymbol{x}_{k+1}) - \sum_{i=1}^m \lambda_{k+1}^i f^i(\boldsymbol{x}_{k+1}^*)\right] \leq (1-\eta_k\mu)\mathbb{E}_{\boldsymbol{\xi}_k}\left[\sum_{i=1}^m \lambda_k^i f^i(\boldsymbol{x}_k) - \sum_{i=1}^m \lambda_k^i f^i(\boldsymbol{x}_k^*)\right]$$

$$+ 4m^{5/2}B^2 H\eta_k(1-\alpha_k) + \frac{Lm^3 B^3 \sigma^2 \eta_k^2}{2} + 2F(1-\alpha_{k+1})\sum_{i=1}^m |\hat{\lambda}_{k+1}^i - \lambda_k^i|.$$

Taking total expectations, this yields

$$\mathbb{E}\left[\sum_{i=1}^m \lambda_{k+1}^i f^i(\boldsymbol{x}_{k+1}) - \sum_{i=1}^m \lambda_{k+1}^i f^i(\boldsymbol{x}_{k+1}^*)\right] \leq (1-\eta_k\mu)\mathbb{E}\left[\sum_{i=1}^m \lambda_k^i f^i(\boldsymbol{x}_k) - \sum_{i=1}^m \lambda_k^i f^i(\boldsymbol{x}_k^*)\right]$$

$$+ 4m^{5/2}B^2 H\eta_k(1-\alpha_k) + \frac{Lm^3 B^3 \sigma^2 \eta_k^2}{2} + 2F(1-\alpha_{k+1})\sum_{i=1}^m |\hat{\lambda}_{k+1}^i - \lambda_k^i|.$$

$\square$

We now are able to provide the convergence analysis for Algorithm 1 with fixed stepsizes.

**Corollary 2.** *Set $0 < \eta_k = \eta \leq \min\{\frac{1}{mLB}, \frac{1}{L}\}$ and $\alpha_k = \max\{1-\eta, 1-\eta^2/(\sum_{i=1}^m |\lambda_{k-1}^i - \hat{\lambda}_k^i|)\}$ in Theorem 6, Algorithm 1 linearly converges to a solution near the Pareto optimal, and the expected gap limits to $\eta M/\mu$, where constant $M = 4m^{5/2}B^2 H + \frac{Lm^3 B^2 \sigma^2}{2} + 2F$.*

*Proof.* From the parameter setting, we know that $1-\alpha_k = \min\{\eta, \eta^2/(\sum_{i=1}^m |\lambda_{k-1}^i - \hat{\lambda}_k^i|)\}$, which suggests that $1-\alpha_k \leq \eta$ and $(1-\alpha_{k+1})\sum_{i=1}^m |\hat{\lambda}_{k+1}^i - \lambda_k^i| \leq \eta^2$. By Theorem 6, we have

$$\mathbb{E}\left[\sum_{i=1}^m \lambda_{k+1}^i f^i(\boldsymbol{x}_{k+1}) - \sum_{i=1}^m \lambda_{k+1}^i f^i(\boldsymbol{x}_{k+1}^*)\right] \leq (1-\eta\mu)\mathbb{E}\left[\sum_{i=1}^m \lambda_k^i f^i(\boldsymbol{x}_k) - \sum_{i=1}^m \lambda_k^i f^i(\boldsymbol{x}_k^*)\right]$$

$$+ 4m^{5/2}B^2 H\eta(1-\alpha_k) + \frac{Lm^3 B^3 \sigma^2 \eta^2}{2} + 2F(1-\alpha_{k+1})\sum_{i=1}^m |\hat{\lambda}_{k+1}^i - \lambda_k^i|$$

$$\leq (1-\eta\mu)\mathbb{E}\left[\sum_{i=1}^m \lambda_k^i f^i(\boldsymbol{x}_k) - \sum_{i=1}^m \lambda_k^i f^i(\boldsymbol{x}_k^*)\right] + \eta^2\left(4m^{5/2}B^2 H + \frac{Lm^3 B^2 \sigma^2}{2} + 2F\right)$$

$$= (1-\eta\mu)\mathbb{E}\left[\sum_{i=1}^m \lambda_k^i f^i(\boldsymbol{x}_k) - \sum_{i=1}^m \lambda_k^i f^i(\boldsymbol{x}_k^*)\right] + \eta^2 M,$$

where $M = 4m^{5/2}B^2 H + \frac{Lm^3 B^2 \sigma^2}{2} + 2F$. Then, subtracting the constant $\eta M/\mu$ from both sides, we obtain

$$\mathbb{E}\left[\sum_{i=1}^m \lambda_{k+1}^i f^i(\boldsymbol{x}_{k+1}) - \sum_{i=1}^m \lambda_{k+1}^i f^i(\boldsymbol{x}_{k+1}^*)\right] - \frac{\eta M}{\mu}$$

$$\leq (1-\eta\mu)\mathbb{E}\left[\sum_{i=1}^m \lambda_k^i f^i(\boldsymbol{x}_k) - \sum_{i=1}^m \lambda_k^i f^i(\boldsymbol{x}_k^*)\right] + \eta^2 M - \frac{\eta M}{\mu}$$

$$= (1-\eta\mu)\mathbb{E}\left[\sum_{i=1}^m \lambda_k^i f^i(\boldsymbol{x}_k) - \sum_{i=1}^m \lambda_k^i f^i(\boldsymbol{x}_k^*)\right] - \frac{\eta M}{\mu}(1-\eta\mu)$$

$$= (1-\eta\mu)\left(\mathbb{E}\left[\sum_{i=1}^m \lambda_k^i f^i(\boldsymbol{x}_k) - \sum_{i=1}^m \lambda_k^i f^i(\boldsymbol{x}_k^*)\right] - \frac{\eta M}{\mu}\right).$$

Observe that the above is a contraction inequality, by the fact that

$$0 < \eta\mu \leq \frac{\mu}{L} \leq 1.$$

The second inequality is by the range of $\eta$. The result thus follows by applying the above contraction inequality repeatedly through iteration $k = 1, \ldots, n$.

$$\mathbb{E}\left[\sum_{i=1}^m \lambda_n^i f^i(\boldsymbol{x}_n) - \sum_{i=1}^m \lambda_n^i f^i(\boldsymbol{x}_n^*)\right]$$
$$\leq (1 - \eta\mu)^{k-1}\left(\mathbb{E}\left[\sum_{i=1}^m \lambda_1^i f^i(\boldsymbol{x}_1) - \sum_{i=1}^m \lambda_1^i f^i(\boldsymbol{x}_1^*)\right] - \frac{\eta M}{\mu}\right) + \frac{\eta M}{\mu}.$$

If $n \to \infty$, then the suboptimal gap for the solution will limit to $\frac{\eta M}{\mu}$. □

***Remark.*** Here, the algorithm does not converge to Pareto optimal but is a suboptimal solution near an optimal solution. This result aligns with the one in single-objective stochastic optimization, and thus is tight in order. Note that this gap is unavoidable, as is the case in single-objective SGD, because the stepsize needs to tradeoff the stochastic variance and the convergence rate (the discussion of Theorem 4.6 in [2]). Additionally, the suboptimal gap is related to the scale of the stepsize $\eta$, and hence we can set sufficiently small $\eta$ to get better convergence.

To guarantee the convergence to Pareto optimal solutions, we can use a diminishing stepsize as the case in single-objective stochastic optimization [2]. The result is as follows.

**Corollary 3.** *Set* $\eta_k = \frac{\beta}{k+\gamma}, \beta > \frac{1}{\mu}$ *where* $\gamma > 0$ *such that* $\eta_k \leq \frac{1}{mLB}$, *and* $\alpha_k = \max\{1 - \eta_k/\eta_1, 1 - \eta_k^2/(\eta_1 \sum_{i=1}^m |\lambda_{k-1}^i - \hat{\lambda}_k^i|)\}$ *in Theorem 6. Algorithm 1 linearly converges to Pareto optimal with $O(1/n)$ rates.*

*Proof.* Specifically, we prove the following result for $k = 1, \ldots, n$

$$\mathbb{E}\left[\sum_{i=1}^m \lambda_k^i f^i(\boldsymbol{x}_k) - \sum_{i=1}^m \lambda_k^i f^i(\boldsymbol{x}_k^*)\right] \leq \frac{v}{k+\gamma},$$

where $v$ satisfies that $v = \max\{\frac{\beta^2 M}{\beta\mu-1}, (1+\gamma)(\sum_{i=1}^m \lambda_1^i f^i(\boldsymbol{x}_1) - \sum_{i=1}^m \lambda_1^i f^i(\boldsymbol{x}_1^*))\}$. From Theorem 6, we know that

$$\mathbb{E}\left[\sum_{i=1}^m \lambda_{k+1}^i f^i(\boldsymbol{x}_{k+1}) - \sum_{i=1}^m \lambda_{k+1}^i f^i(\boldsymbol{x}_{k+1}^*)\right] \leq (1 - \eta_k\mu)\mathbb{E}\left[\sum_{i=1}^m \lambda_k^i f^i(\boldsymbol{x}_k) - \sum_{i=1}^m \lambda_k^i f^i(\boldsymbol{x}_k^*)\right]$$
$$+ 4m^{5/2}B^2 H\eta_k(1-\alpha_k) + \frac{Lm^3 B^3 \sigma^2 \eta_k^2}{2} + 2F(1-\alpha_{k+1})\sum_{i=1}^m |\hat{\lambda}_{k+1}^i - \lambda_k^i|$$

$$\leq (1 - \eta_k\mu)\mathbb{E}\left[\sum_{i=1}^m \lambda_k^i f^i(\boldsymbol{x}_k) - \sum_{i=1}^m \lambda_k^i f^i(\boldsymbol{x}_k^*)\right] + \eta_k^2\left(\frac{4m^{5/2}B^2 H}{\eta_1} + \frac{Lm^3 B^2 \sigma^2}{2} + \frac{2F}{\eta_1}\right) + \frac{2F\eta_{k+1}^2}{\eta_1}$$

$$\leq (1 - \eta_k\mu)\mathbb{E}\left[\sum_{i=1}^m \lambda_k^i f^i(\boldsymbol{x}_k) - \sum_{i=1}^m \lambda_k^i f^i(\boldsymbol{x}_k^*)\right] + \eta_k^2\left(\frac{4m^{5/2}B^2 H}{\eta_1} + \frac{Lm^3 B^2 \sigma^2}{2} + \frac{4F}{\eta_1}\right)$$

$$= (1 - \eta_k\mu)\mathbb{E}\left[\sum_{i=1}^m \lambda_k^i f^i(\boldsymbol{x}_k) - \sum_{i=1}^m \lambda_k^i f^i(\boldsymbol{x}_k^*)\right] + \eta_k^2 M,$$

where the constant in the above is defined as $M = \frac{4m^{5/2}B^2 H}{\eta_1} + \frac{Lm^3 B^2 \sigma^2}{2} + \frac{4F}{\eta_1}$ for simplicity. The last inequality is by the fact that $\eta_k \geq \eta_{k+1}$. We then prove the corollary by mathematical induction. First, the parameter setting here ensures the result holds for $k = 1$. Next, we assume the inequality of the theorem holds for $k$, which holds

$$\mathbb{E}\left[\sum_{i=1}^m \lambda_k^i f^i(\boldsymbol{x}_k) - \sum_{i=1}^m \lambda_k^i f^i(\boldsymbol{x}_k^*)\right] \leq \frac{v}{k+\gamma}.$$

Then by the parameter setting, for the case of $k + 1$, we have

$$\mathbb{E}\left[\sum_{i=1}^{m} \lambda_{k+1}^{i} f^{i}(\boldsymbol{x}_{k+1}) - \sum_{i=1}^{m} \lambda_{k+1}^{i} f^{i}(\boldsymbol{x}_{k+\gamma}^{*})\right]$$

$$= (1 - \eta_k \mu)\mathbb{E}\left[\sum_{i=1}^{m} \lambda_k^{i} f^{i}(\boldsymbol{x}_k) - \sum_{i=1}^{m} \lambda_k^{i} f^{i}(\boldsymbol{x}_k^{*})\right] + \eta_k^2 M$$

$$\leq (1 - \frac{\beta\mu}{k+\gamma})\frac{v}{k+\gamma} + \frac{\beta^2 M}{(k+\gamma)^2}$$

$$= \frac{k+\gamma - \beta\mu}{(k+\gamma)^2} v + \frac{\beta^2 M}{(k+\gamma)^2}$$

$$= \frac{k+\gamma - 1}{(k+\gamma)^2} v + \frac{1 - \beta\mu}{(k+\gamma)^2} v + \frac{\beta^2 M}{(k+\gamma)^2}$$

$$= \frac{k+\gamma - 1}{(k+\gamma)^2} v - \frac{1}{(k+\gamma)^2}((\beta\mu - 1)v - \beta^2 M)$$

$$\leq \frac{k+\gamma - 1}{(k+\gamma)^2} v,$$

where the last inequality is by the condition of $v$. By the fact that $(a-1)(a+1) \leq a^2$ for $a > 0$, we have $\frac{k+\gamma-1}{(k+\gamma)^2} \leq \frac{k+\gamma-1}{(k+\gamma-1)(k+1+\gamma)} = \frac{1}{k+1+\gamma}$. Plug in the above, we can prove the result for $k+1$. Thus, we theorem holds for $k = 1, \ldots, n$, which suggests an $O(1/n)$ convergence. $\square$

***Remark 1.*** The result from Corollary 3 shows the convergence rate is aligned with single-objective SGD with adaptive stepsize, suggesting that Algorithm 1 is tight in order with strongly convex functions.

***Remark 2.*** The bounded function assumption in Theorem 6 can be relaxed to the combination of the assumptions in the convex and non-convex settings. Specifically, the function values of Pareto optimal solutions are bounded, i.e., $|f^i(\boldsymbol{x}^*)| \leq F, i = 1, \ldots, m$, and the function values of iterates along the practical training path are bounded, i.e., $f^1(\boldsymbol{x}_k), \ldots, f^m(\boldsymbol{x}_k) \leq F$ for $k = 1, \ldots, n$.

## G  Additional Experiments and More Details

### G.1  More Insights of CR-MOGM

**How the non-convergence reflects in practice.** In fact, the similar situation reflected by the counter-example is not rare in practice, especially when the stochastic noises are large. For example, we have added an additional experiment of MultiMNIST by setting batch size from 8 to 2, representing increasing levels of stochastic noises (a smaller batch size introduces a larger stochastic noise in the mini-batch gradient). We report the training losses for the final outputs of MGDA and CR-MGDA as follows, representing the comparison of MOGM and CR-MOGM. From results of Table 2, we observe that the loss of vanilla MGDA becomes significantly larger when the batch size is smaller, while CR-MGDA appears not sensitive to the batch size. This possible reason is that the convergence issue of MOGM becomes more severe with larger stochastic noise, which severely degrades the performance of MOGM. Since CR-MOGM is proven to converge to the Pareto optimal/critical, it is much more robust to stochastic noises. Thus, in this regard, our method is more practical for real-world applications.

**Comparison with the full-batch MOGM.** Although there are no guarantees for the closeness between $\boldsymbol{\lambda}$ calculated by CR-MOGM and the actual $\boldsymbol{\lambda}$ calculated by MOGM with full-batch gradients. As shown in Figure 4, we empirically observe that the $\boldsymbol{\lambda}_k$ of CR-MOGM is very close to the actual $\boldsymbol{\lambda}_k^*$ yielded by the MOGM with full batch gradients. Specifically, the gaps between $\boldsymbol{\lambda}_k$ of CR-MOGM and actual $\boldsymbol{\lambda}_k^*$ are below 0.01 with low error bars in the toy example, while the gaps of vanilla MOGM algorithms are all averagely larger than 0.10 with large error bars.

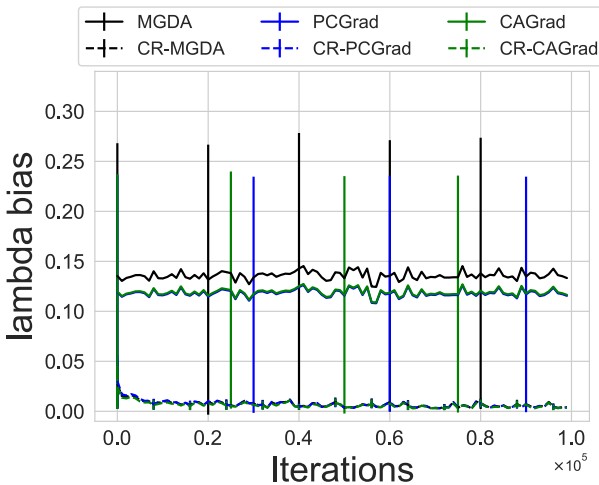

Figure 4: Experiments on the illustrative example in Section 4. We measure the bias between $\boldsymbol{\lambda}$ calculated by CR-MOGM and the actual $\boldsymbol{\lambda}$ calculated by MOGM with full-batch gradients. We report the mean and std.

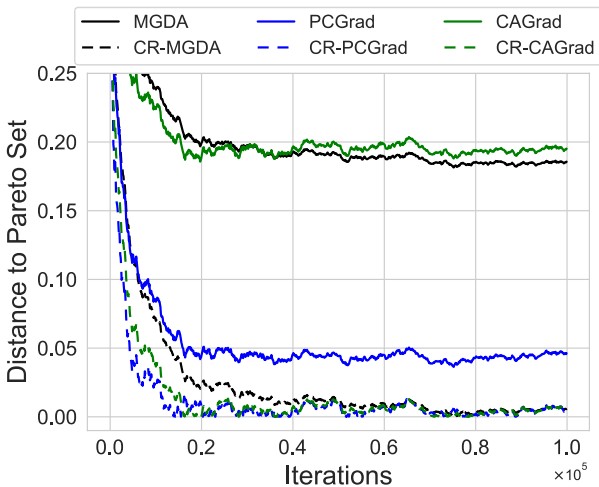

Figure 5: Simulation with Gaussian noise.

### G.2 Additional Stochastic Convex Simulation

In this simulation, we establish a similar setting with Section 4, where we aim to optimize $\boldsymbol{x} \in \mathbb{R}^2$ to attain smaller distances to two fixed points $\boldsymbol{a} = [-3, 0], \boldsymbol{b} = [3, 0]$. The first objective is $f^1(\boldsymbol{x}) = \|\boldsymbol{x} - \boldsymbol{a}\|_2^2$, and the second one is $f^2(\boldsymbol{x}) = \|\boldsymbol{x} - \boldsymbol{b}\|_2^2$. The stochastic gradient for each objective can be represented as $g^1(\boldsymbol{x}) = \boldsymbol{x} - \boldsymbol{a} + \boldsymbol{\xi}^1$ and $g^2(\boldsymbol{x}) = \boldsymbol{x} - \boldsymbol{b} + \boldsymbol{\xi}^2$. The stochastic noises $\boldsymbol{\xi}^1, \boldsymbol{\xi}^2$ are two dimensional random noise with zero-mean Gaussian distribution, where the covariance matrix for $\boldsymbol{\xi}^1$ is $[[3, 2], [2, 3]]$ and for $\boldsymbol{\xi}^2$ is $[[3, -2], [-2, 3]]$.

Figure 5 demonstrates the distance to Pareto set. We observe that all three MOGM algorithms can not converge to the Pareto optimal points, while all CRMOGM algorithms can approach the Pareto set. This result is aligned with the result in Section 4, and the more general Gaussian noise shows that the non-convergence is not special in stochastic multiobjective optimization.

### G.3 Implementation Details of Experiments

The implementation for CRMOGM is very simple, just by adding one additional line that exponentially smoothes the composite weights to MOGM algorithms. Note that this mechanism does not require additional computational cost and is flexible to be implemented in MOGM algorithms.