# OpenReview forum: "On the Convergence of Stochastic Multi-Objective Gradient Manipulation and Beyond"
_NeurIPS.cc/2022/Conference — NeurIPS 2022 Accept_

### Official Review · Reviewer_DjJR · 2022-07-05

**Rating:** 7
**Confidence:** 4
**Soundness:** 4 excellent
**Presentation:** 4 excellent
**Contribution:** 4 excellent

**Summary:**

This paper proposed a new algorithm for stochastic multi-objective learning that solves some non-convergence issues in existing multi-objective learning algorithms. At each iteration, the update direction is computed as the weighted combination of gradient directions of every objective, and the weight is an exponential average of weights in past iterations. Such a weight scheme aims at reducing the correlation between the weight and stochastic gradient at each iteration. The convergence analysis is provided for the proposed algorithm and experiments show its advantages over existing algorithms.

This paper is clearly written. The proposed algorithm is simple and shows great improvement over existing algorithms. Therefore I recommend "Accept"

**Questions:**

All of them are minor:

1. line 105, the author may want to clarify here what "$F$ is convex" means, it only gets clear when I read Theorem 2.
2. line 315/316, probably add citations for the convergence rate in the single-objective setting?


**Limitations:**

The author claimed that there are discussions on limitations in the appendix but it is hard to find. Maybe it is better to put the discussions in a separate section.

**Strengths And Weaknesses:**

This paper gives an excellent introduction to multi-objective learning, identifies convergence issues in existing approaches, and proposes a new algorithm as the solution. The paper is clearly written and easy to follow even for a non-expert in multi-objective optimization. Theoretical results on convergence are also provided together with experiments showing improvement over existing approaches. I can not assess the novelty of the paper as I'm less familiar with the literature on multi-objective learning.

---

> ### Author Response · Authors · 2022-08-02
> **Author Response**
>
> Thanks for your positive review!
>
> **Q1: line 105, the author may want to clarify here what "F is convex" means, it only gets clear when I read Theorem 2.**
>
> **A:** We have rewritten it to "For convex objectives $f^i,i=1,\ldots,m$."
>
> **Q2: line 315/316, probably add citations for the convergence rate in the single-objective setting?**
>
> **A:** We have added related citations in the revision ([1] for the non-convex setting, [2] for the convex and strongly convex settings).
>
> **Q3: The author claimed that there are discussions on limitations in the appendix but it is hard to find. Maybe it is better to put the discussions in a separate section.**
>
> **A:** We apologize for missing this part! Now, we have added the discussion of limitations in Appendix H in the revision. Specifically, due to the space limitation, this paper only discusses the typical MOGA algorithms with strict convergence guarantees, i.e., MGDA, PCGrad, and CAGrad. Other empirical-driven methods without rigorous theoretical guarantees, such as RotoGrad [3], are too expensive to be included in our analysis. We believe that the convergence issue in that line of literature will be a very good future work to explore in depth.
>
> **Q4: The novelty of the paper.**
>
> **A:**  We would like to highlight that this paper is the first to point out the non-convergence issue of existing MOGA algorithms under the stochastic setting, by constructing an intuitive showcase. Our theory incorporated with the general framework provides a unified convergence analysis for typical MOGA algorithms.
>
> ### Reference
>
> [1] Drori, et al. “The complexity of finding stationary points with stochastic gradient descent.” ICML 2020.
>
> [2] Léon, et al. "Optimization methods for large-scale machine learning."  Siam Review 2018.
>
> [3] Javaloy, et al. "RotoGrad: Gradient Homogenization in Multitask Learning." ICLR 2021.

---

### Official Review · Reviewer_eaLg · 2022-07-06

**Rating:** 4
**Confidence:** 4
**Soundness:** 3 good
**Presentation:** 3 good
**Contribution:** 2 fair

**Summary:**

Multi-objective optimization (MOO) has been an area of research interest lately, where the goal is to optimize multiple objectives simultaneously, using some optimality condition (e.g. Pareto optimality). One popular MOO algorithm is the multi-objective gradient alteration (MOGA) algorithm, where a common descent direction is chosen for all the objectives by manipulating objective gradients. In most MOGA literature, analysis is only provided using deterministic gradients, although in practice only stochastic gradients are used for calculating the common descent direction. This work addresses this issue. The authors propose an optimizer-agnostic framework to reduce the correlation between the aforementioned weights and the stochastic gradients, which can be applied to many MOGA algorithms. The authors specify the convergence criteria and provide theoretical guarantees for the convergence of the proposed algorithm for convex and non-convex cases. The authors also provide some empirical results to support their claims.


**Questions:**

* Could you justify the use of bounded objective assumption in the convergence analysis of the non-convex case, since it may not be true in general?

* Could you provide justification/proof for the claim in line 542? Specifically, the concern is that a Pareto stationary point need not be a weakly Pareto optimal point in general even in a convex setting.

* Could you provide any insight into how close $\lambda_k$ in CR-MOGA is to the actual $\lambda_k$ that is calculated by the corresponding MOGA? The work does not seem to provide adequate insight in this regard. If this closeness is not significant, could you explain the reason?

* Could you provide an intuition for the expected value of $\lambda_k$? By the variance reduction, $\lambda_k$ becomes closer to the expected value, yet it is unclear what this value represents. This might be related to the previous question.

**Limitations:**

- Authors have identified the limitations of the examples they have used to introduce the problem, but provide additional experiments in a more practical setting that support their claims.

- Authors may need to address the limitations of the work due to the strong assumptions such as the boundedness of the objectives.

- There is no potential negative societal impact from this work.


**Strengths And Weaknesses:**

***Strengths***
- Investigate a key and important issue in the current MOO line of research;
- Provide rigorous evidence of the non-convergence problem, and provide key insights into the source of the problem;
- Propose a simple solution to the problem which has insignificant computational overhead compared to existing MOGA algorithms;
- Provide theoretical and empirical justification for the performance improvement using the proposed algorithm.

***Weaknesses***
- The main theoretical results provided in the main text are inaccurate or different from those in the supplementary;
- Some strong assumptions are not standard without providing a clear justifications. Specifically, the authors use a bounded objectives assumption for Theorems 2 and 3 (line 739,  798), which is not standard in optimization literature and prior work on MOGA algorithms.
- It is not clear how the unified framework still guarantees the benefits of the original MOGA algorithms. Specifically, the assumptions on $\lambda_k$ do not reflect the special properties of different MOGA algorithms, thus it is unclear how the modification will allow behavior similar to the original algorithm.
- Seemingly incorrect/unclear proof steps possibly resulting in weaker dependence on the number of objectives $m$ in final results (e.g. lines 671, 687). Specifically, a factor of $m$ seems missing when upper bounding the squared norm.
- Some claims on the relation between Pareto criticality and weak Pareto optimally in the convex setting need to be justified (line 542. Refer to the “Questions” for further elaboration of the concern).
- Missing definition of notations (e.g. $[m]$ in text, possibly meaning $\{1, 2, , …, m}$; p in toy example, probably meaning probability of occurrence)
- Inconsistent notations used ( e.g. line 688, $\leq$ used for comparing scalers and vectors without clarification)

---

> ### Author Response · Authors · 2022-08-02
> **Author Response Part I**
>
> Thanks for your insightful and detailed review!
>
> **Q1: The main theoretical results provided in the main text are inaccurate or different from those in the supplementary.**
>
> **A:** We apologize for incurring the confusion. During preparing the supplementary material, we found some minor issues that do not affect the main results in the theorems. Therefore, we have fixed these issues in the appendix of the supplementary material, and highlighted this correction in the introduction at the beginning of the appendix. In the revision, we have revised the theorems in the main text.
>
> **Q2: Some strong assumptions are not standard without providing clear justifications. Specifically, the authors use a bounded objectives assumption which is not standard in optimization literature and prior work on MOGA algorithms.**
>
> **A:** In fact, the assumption of bounded functions has been used in recent studies on non-convex stochastic optimization, such as analyzing SGD (Assumption 2 in [1]), and analyzing SGD+momentum (Assumption in Section 3 of [2]).
>
> Actually, this assumption can be further relaxed. In the convex setting, it can be relaxed to that the function values of Pareto optimal solutions are bounded, i.e., $|f^i(\boldsymbol{x}^*)|\leq F, i=1,\ldots,m$, if $\boldsymbol{x}^*$ is Pareto optimal. The reason is that we only require this condition in Lemma 11 (line 791). The relaxed assumption is surely satisfied since the optimal solutions naturally have bounded function values. In the non-convex setting, note that in the related papers analyzing SGD (Remark 2 in the analysis of SGD [3]) as well as MOGA (Theorem 3.1 in the analysis of MGDA [4]) with non-convex functions, it is commonly assumed that the initial function values $f^i(\boldsymbol{x}_1),i=1,\ldots,m$ are bounded. Essentially, our analysis only requires that $f^i(\boldsymbol{x}_k),i=1,\ldots,m$ are bounded for $k=1,\ldots,n$, since we only use it in Lemma 13 (line 821). This assumption is mild in practice as long as the function values of iterates along the practical training path are bounded. In fact, all our experiments indicate that the function values are well located in a bounded regime.
>
> In Appendix E.7, we have added further discussions to clarify the use of the assumptions.
>
> **Q3: It is not clear how the unified framework still guarantees the benefits of the original MOGA algorithms. Specifically, the assumptions on  $\lambda_k$  do not reflect the special properties of different MOGA algorithms, thus it is unclear how the modification will allow behavior similar to the original algorithm.**
>
> **A:** Note that the unified framework is a general template, aiming at abstracting the unified property of different MOGA algorithms rather than their specific properties. Therefore, the general framework also includes other recent algorithms such as GradVac [5], IMTL-G [6]. In fact, the special properties of MOGA algorithms do not lie in the unified framework, but in the specific derivation of composite weights $\boldsymbol{\lambda}_k$, which is determined by specific algorithms. Assumption 1 is only a general proposition for studying the convergence, and does not need to reflect the property of a specific algorithm.
>
> In addition, the modification of CR-MOGA is only exponentially smoothing the composite weights $\boldsymbol{\lambda}_k$ calculated by the corresponding MOGA algorithm. Intuitively, the smoothing parameter $\alpha_k$ is a tunable parameter. Specifically, if $\alpha_k$ is small, then the CR-MOGA performs similarly to the original MOGA, while the convergence is hard to guarantee; if $\alpha_k$ is closed to $1$, then the non-convergence issue disappears, but it may discard most of the property of the original MOGA. By setting an increasing $\alpha_k$, CR-MOGA can remain the property of the original algorithm as much as possible under the condition for convergence. Specifically, in the early training stage, the main goal is not to converge to Pareto optimal/critical, so $\alpha_k$ is set to be small to maintain the algorithmic property. In the late training stage, the primary goal should be the access of Pareto optimality/criticality, so $\alpha_k$ is set to be closed to $1$ to guarantee the convergence.
>
> We will add the above discussions in the final revision.
>
> ### Reference
> [1] Fehrman, et al. "Convergence rates for the stochastic gradient descent method for non-convex objective functions." JMLR 2020.
>
> [2] Levy, et al. "STORM+: Fully adaptive SGD with recursive momentum for nonconvex optimization." NeurIPS 2021.
>
> [3] Drori, et al. “The complexity of finding stationary points with stochastic gradient descent.” ICML 2020.
>
> [4] Fliege, et al. "Complexity of gradient descent for multiobjective optimization." Optimization Methods and Software 2019.
>
> [5] Wang, et al. "Gradient Vaccine: Investigating and Improving Multi-task Optimization in Massively Multilingual Models." ICLR 2020.
>
> [6] Liu, et al. "Towards impartial multi-task learning." ICLR 2021.

---

> > ### Author Response · Authors · 2022-08-02
> > **Author Response Part II**
> >
> > **Q4: Weaker dependence on the number of objectives  $m$  in final results (e.g. lines 671, 687).**
> >
> > **A:** It is a mistake. We have fixed the dependence on $m$ in the revision.
> >
> > **Q5: Could you provide any insight into how close  $\boldsymbol{\lambda}_k$ in CR-MOGA is to the actual  $\boldsymbol{\lambda}_k^\*$  that is calculated by the corresponding MOGA? The work does not seem to provide adequate insight in this regard. If this closeness is not significant, could you explain the reason?**
> >
> > **A:** Insightful question! In fact, in this paper, our focus is to fix the non-convergence issue of the vanilla MOGA algorithms under the stochastic setting. Our proposed momentum-like method successfully ensures convergence. However, in our theoretical derivation, we found that the convergence to Pareto optimality/criticality does not depend on how close the $\boldsymbol{\lambda}_k$ with the actual $\boldsymbol{\lambda}_k^\*$ (calculated by full-batch gradients). The root reason is that in the multi-objective setting, there are usually many Pareto optimal points, and approaching any of them can be reviewed as convergence. Therefore, in our paper, we do not discuss the closeness between $\boldsymbol{\lambda}_k$ and $\boldsymbol{\lambda}_k^\*$. In fact, most multi-objective algorithms (offline) do not guarantee to converge to any specific point in the Pareto set, which corresponds to a certain fixed $\boldsymbol{\lambda}$. In particular, it is another line of works of multi-objective learning, such as [1] and [2], that guarantees to converge to a solution with a fixed $\boldsymbol{\lambda}$.
> >
> > Although there are no closeness guarantees, in the added experiments in the revision, we empirically observe that the $\boldsymbol{\lambda}_k$ of CR-MOGA is very close to the actual $\boldsymbol{\lambda}_k^\*$ yielded by the MOGA with full batch gradients. Specifically, the gaps between $\boldsymbol{\lambda}_k$ CR-MOGA and actual $\boldsymbol{\lambda}_k^\*$ are below 0.01 in the toy example, while the gaps of vanilla MOGA algorithms are all averagely larger than 0.10.
> >
> > **Q6: Could you provide an intuition for the expected value of  $\boldsymbol{\lambda}_k$? By the variance reduction,  $\boldsymbol{\lambda}_k$  becomes closer to the expected value, yet it is unclear what this value represents. This might be related to the previous question.**
> >
> > **A:** The expected value of $\boldsymbol{\lambda}\_k$ is defined as $\mathbb{E}_{\xi_k}[\boldsymbol{\lambda}_k] = \lim\_{n\rightarrow \infty} \frac{1}{n} \sum\_{l=1}\^n \boldsymbol{\lambda}_k(\boldsymbol{G}\_k^l)$, where $\boldsymbol{G}\_k^l,l=1,\ldots,n$ are i.i.d. sampled multiple gradients. It can be viewed as the average composite weights for the infinite sampled gradient batches by vanilla MOGA algorithms. As illustrated in Section 4, the non-convergence is caused by the inherited randomness of $\boldsymbol{\lambda}\_k$ from the underlying stochastic gradients, i.e., $\boldsymbol{\lambda}\_k$ is correlated with the stochastic gradients. Intuitively, compared to one single $\boldsymbol{\lambda}\_k$ yielded by vanilla MOGA, $\mathbb{E}\_{\xi_k}[\boldsymbol{\lambda}\_k]$ represents a stable version of the calculated composite weights, which largely removes the inherent randomness.
> >
> > You may expect similar behavior of momentum-based variance reduction of SGD [3], where the smoothed gradient asymptotically converges to the full (actual) gradient. However, since the derivation for $\boldsymbol{\lambda}_k$ is very complex, recovering the actual $\boldsymbol{\lambda}_k^\*$ yielded by the full-batch MOGA algorithm is very difficult. Specifically, since $\boldsymbol{\lambda}_k$ is not Lipschitz (Proposition 2), $\boldsymbol{\lambda}_k$ will have a large bias even if the stochastic noise is very small. Therefore, it is unlikely to have an unbiased estimator of the actual $\boldsymbol{\lambda}_k^\*$ if just using the stochastic gradients.
> >
> > Note that our focus is "_fixing the non-convergence issue of vanilla MOGA algorithms that can not converge to Pareto optimal/critical points_." From **Q5**, we know that the closeness with the actual $\boldsymbol{\lambda}_k^\*$ derived from the full-batch algorithms is not necessary for convergence. CR-MOGA is a general technique to fix the non-convergence issue under the condition of maintaining the main properties of the original MOGA algorithm (see **Q3**), which adequately solves the main problem.
> >
> > ### Reference
> > [1] Mahapatra, et al. "Multi-task learning with user preferences: Gradient descent with controlled ascent in pareto optimization." ICML 2020.
> >
> > [2] Momma, et al. "A multi-objective/multi-task learning framework induced by Pareto stationarity." ICML 2022.
> >
> > [3] Cutkosky, et al. "Momentum-based variance reduction in non-convex sgd." NeurIPS 2019.

---

> > > ### Author Response · Authors · 2022-08-03
> > > **Author Response Part III**
> > >
> > > **Q7: Some claims on the relation between Pareto criticality and weak Pareto optimally in the convex setting need to be justified.  Could you provide justification/proof for the claim in line 542? Specifically, the concern is that a Pareto stationary point need not be a weakly Pareto optimal point in general even in a convex setting.**
> > >
> > > **A:** In fact, it is a well-acknowledged result of multi-objective optimization that a Pareto stationary point should be a weakly Pareto optimal point if the functions are convex. Please refer to Lemma 2.2 in [1] and also Lemma 2.1 in [2]. It is also not difficult to verify that if $\boldsymbol{x}^*$ is a Pareto stationary point, then it satisfies $\nabla \boldsymbol{F}(\boldsymbol{x}\^\*)\boldsymbol{\lambda}\^\*=0$ for one $\boldsymbol{\lambda}\^\*\in S\_m$. If the functions are convex, this meets the optimal condition of the composite function ${\boldsymbol{\lambda}^*}\^\top \boldsymbol{F}(\boldsymbol{x}^*)$. And by Proposition 1 (b): "For convex objectives $f\^i(\cdot),i=1,\ldots,m$. If there exists $\boldsymbol{\lambda}\in S\_m$ such that $\boldsymbol{x}\^* = \arg\min\_{\boldsymbol{x}\in \mathcal{K}} \boldsymbol{\lambda}\^T \boldsymbol{F}(\boldsymbol{x})$, then $\boldsymbol{x}^*$ is weak Pareto optimal", we then know that $\boldsymbol{x}$ is weak Pareto optimal. For the ease of better understanding, we have added these references in these claims in the revision.
> > >
> > > **Q8: Missing definition of notations (e.g.  $[m]$  in text, possibly meaning  $1,2,,…,m$; $p$ in toy example, probably meaning probability of occurrence). Inconsistent notations used ( e.g. line 688,  ≤  used for comparing scalers and vectors without clarification)**
> > >
> > > **A:** We have added the missing definitions and revised all the typos in the revision.
> > >
> > > ### References
> > >
> > > [1] Tanabe, et al. "Convergence rates analysis of a multiobjective proximal gradient method." Optimization Letters 2022.
> > >
> > > [2] Tanabe, et al. "Proximal gradient methods for multiobjective optimization and their applications" Computational Optimization and Applications 2019.

---

### Official Review · Reviewer_XWtt · 2022-07-08

**Rating:** 7
**Confidence:** 3
**Soundness:** 3 good
**Presentation:** 4 excellent
**Contribution:** 3 good

**Summary:**

This work studies the conflicting gradient problem in multi-objective optimization in the stochastic (minibatch) setting. The authors survey some recent gradient alterations algorithms, put them under one general framework (or update rule), and then construct a counter-example that shows that such algorithms can fail to converge to a Pareto optimal solution in the stochastic setting. They propose to deal with this using a momentum-based update for the gradient composite weights.

**Questions:**

How well does the counter-example translate to real-world data? It seems to be specifically constructed to make convergence fail. It is mentioned that a similar situation can happen with “more general Gaussian noise”, but wouldn’t the noise from SGD(+momentum) help mitigate getting into such rigid suboptimal situations?

Perhaps proposing a reasonable assumption on the data distribution can avoid such a counter-example altogether?

**Limitations:**

This work has no apparent limitations and potential negative societal impact.

**Strengths And Weaknesses:**

The paper is well-written and has a clear structure that makes it easy to read. It starts by stating the problem, then summarizes known approaches and pointing the common problem in them, and finally proposing a solution and validating it experimentally. The problem is well-posed and is clearly illustrated in the counter-example and in Figure 1. The improvements of the proposed fix in this specific example are significant.

The fix employs an exponential average of gradient composite weights. It is simple, practical, and easy to implement. Works just like momentum in SGD, so it can be added to any gradient alteration algorithm. The additional compute and memory footprint are not significant.

The results overall are better, but not significantly so. Perhaps this is because the situation in the counter-example is not as serious in real-world data as is thought to be. In any case, the adjustment can be made to any gradient alteration algorithm without affecting performance, so this is not a big problem. Still, it would be nice to see real-world situations where this adjustment can yield clear and significant improvements just to account for the additional complexity,

Minor issues:
* There is a typo in Assumption 2 (line 193): “… which are unbiased estimates of $f^1$ …”. There should be a $\nabla$ before $f^1$, and so on.
* Figure 1 is a bit too small, especially the font. I had to zoom in to see clearly.
* Are the results in Figure 2 for MultiMNIST done for multiple seeds? If so, then it should be mentioned somewhere and the std bands should be plotted.
* As for Table 1, reporting the std would be better. Also, some acronyms are not explained, such as mIoU and Pix Acc. And finally, the relative error under Depth->Testing->Rel Err for CR-PCGrad is larger than PCGrad but is in bold. Looking at the absolute errors, I assume there is a mistake in the calculation.
* I think the bounds in Theorem 2 and Theorem 3 can be written a bit more clearly. The third term in the right hand side of Theorem 1 has an $\eta_k$ in the denominator, but the summation of the index $k$ is in the numerator.
* The conclusion is also a bit lacking. It would be nice for the readers to discuss future directions and room for improvements.

---

> ### Author Response · Authors · 2022-08-02
> **Author Response Part I**
>
> Thanks for your inspirational review!
>
> **Q1: The results overall are better, but not significantly so. Perhaps this is because the situation in the counter-example is not as serious in real-world data as is thought to be. It would be nice to see real-world situations where this adjustment can yield clear and significant improvements just to account for the additional complexity.**
>
> **A:** Thought-provoking question! We would like to highlight that the real-world datasets used in our paper have been studied for several years, so further improving the performance, especially for all the objectives simultaneously, is very difficult. Note that in the experiment for MultiMNIST (Table 2), PCGrad improves MGDA with only 0.17\% and 0.08\% increase in terms of accuracy of task-L and task-R, while CR-PCGrad further improves PCGrad with 0.27\% and 0.38\% for the corresponding tasks respectively. Additionally, if we look at the improvement in terms of the loss values in Figure 2,  the relative improvement of CR-MOGA is about 5\% to 30\%, which is much more significant.
>
> In fact, the similar situation reflected by the counter-example is not rare in practice, especially when the stochastic noises are large. For example, we have added an additional experiment of MultiMNIST by setting batch size from 8 to 2 (Table 3 in Appendix G.1), representing increasing levels of stochastic noises (a smaller batch size introduces a larger stochastic noise in the batch gradient). We report the training losses for the final outputs of MGDA and CR-MGDA as follows, representing the comparison of MOGA and CR-MOGA.
>
> | Batch Size |  MGDA loss-L  |  MGDA loss-R  | CR-MGDA loss-L | CR-MGDA loss-R |
> |:----------:|:-------------:|:-------------:|:--------------:|:--------------:|
> |      8     | 0.44$\pm$0.02 | 0.54$\pm$0.03 |  0.24$\pm$0.01 |  0.35$\pm$0.01 |
> |      4     | 0.57$\pm$0.03 | 0.65$\pm$0.03 |  0.26$\pm$0.02 |  0.36$\pm$0.02 |
> |      2     | 0.71$\pm$0.05 | 0.81$\pm$0.05 |  0.28$\pm$0.03 |  0.40$\pm$0.04 |
>
> From the above results, we observe that the loss of vanilla MGDA becomes significantly larger when the batch size is smaller, while CR-MGDA appears not sensitive to the batch size. This possible reason is that the convergence issue of MOGA becomes more severe with larger stochastic noise, which severely degrades the performance of MOGA. Since CR-MOGA is proven to converge to the Pareto optimal/critical, it is much more robust to stochastic noises. Thus, in this regard, our method is more practical for real-world applications.
>
> As for the additional complexity, we note that the only extra operation in CR-MOGA is to smooth the composite weights $\lambda_k$. The complexity of such an operation only depends on the dimension of $\lambda_k$, which is the same as the number of objectives that is usually not very large (often 2 or 3). Thus, it introduces nearly negligible extra computational overhead.
>
> **Q2: Figure 1 is a bit too small, especially the font. I had to zoom in to see clearly.**
>
> **A:** Good suggestion! In the revision, we have enlarged the font in Figure 1 to make it clearer.
>
> **Q3: Are the results in Figure 2 for MultiMNIST done for multiple seeds? If so, then it should be mentioned somewhere and the std bands should be plotted. As for Table 1, reporting the std would be better.**
>
> **A:** Yes. We report the average results over three runs with different random seeds. We have added this claim in the Figure 2 caption. Additionally, for the std bands, we have added error bars in all Tables and Figures in the revision (Appendix G.2). Note that for the Figures, since we have six lines in the same figure, the error bars make the figures too complex to be understood, as shown in Appendix F in the revision. With this kind of consideration, we maintain the original Figures in the main text, given that the error bars in Table 2 can reveal similar information.
>
> **Q4: Also, some acronyms are not explained, such as mIoU and Pix Acc.**
>
> **A:** Good suggestion! We have added explanations of all acronyms in the revision.
>
> **Q5: The relative error under Depth->Testing->Rel Err for CR-PCGrad is larger than PCGrad but is in bold. Looking at the absolute errors, I assume there is a mistake in the calculation.**
>
> **A:** Actually, we had reported the wrong number. We have corrected it in the revision.
>
> **Q6: I think the bounds in Theorem 2 and Theorem 3 can be written a bit more clearly. The third term in the right hand side of Theorem 1 has an  $\eta_k$  in the denominator, but the summation of the index  $k$  is in the numerator.**
>
> **A:** We apologize for the confusion. The index $k$ in the numerator should be $n$, which has been fixed in our revision.

---

> > ### Author Response · Authors · 2022-08-02
> > **Author Response Part II**
> >
> > **Q7: The conclusion is also a bit lacking. It would be nice for the readers to discuss future directions and room for improvements.**
> >
> > **A:** Good suggestion! We have added the discussion of limitations and future directions in Appendix H in the revision. Specifically, due to the space limitation, this paper only discusses the typical MOGA algorithms with strict convergence guarantees, i.e., MGDA, PCGrad, and CAGrad. Other empirical-driven methods without rigorous theoretical guarantees, such as RotoGrad [1], are too expensive to be included in our analysis. We believe that the convergence issue in that line of literature will be a very good future work to explore in depth.
> >
> > **Q8: How well does the counter-example translate to real-world data? It seems to be specifically constructed to make convergence fail. Perhaps proposing a reasonable assumption on the data distribution can avoid such a counter-example altogether?**
> >
> > **A:** Insightful question! To illustrate the non-convergence issue more clearly, we need an intuitive example, which can demonstrate the key insight of how the non-convergence occurs. To this end, we construct a counter-example, where the resulting composite gradients of MOGA can inversely optimize all the objectives simultaneously. The key insight behind the counter-example is to demonstrate that the correlation between the composite weights and the underlying noisy stochastic gradients can result in undesired or even totally wrong 'descent' directions. The example is specifically constructed to make the insight more easily accessible. In fact, illustrating the non-convergence issue via specifically constructed examples is commonly used in the existing literature on stochastic optimization (see [2] for a famous example illustrating Adam's non-convergence). You can imagine that if we directly provide a complex real-world example, it would be much harder to understand how the non-convergence happens. Actually, the result is indeed very general in real-world datasets, where the stochastic gradients can be even more noisy. As discussed in Q1, we have demonstrated that the convergence of MOGA algorithms will become much worse in real-world datasets when the stochastic noises are more significant.
> >
> > As for your desired assumption on the data distribution, currently, we can only intuitively summarize it as follows: when the data is noisy, the corresponding noisy stochastic gradients will possibly incur certain composite weights that yield undesired descent directions. We apologize for not abling to give you an ideal answer on the data distribution as the real-world data is too complex to analyze. We believe that finding a more reasonable assumption on data distribution can be left as a good direction for future work.
> >
> > **Q9: It is mentioned that a similar situation can happen with “more general Gaussian noise”, but wouldn’t the noise from SGD(+momentum) help mitigate getting into such rigid suboptimal situations?**
> >
> > **A:** We clarify that in CR-MOGA, the momentum is only applied to composite weight $\lambda_k$ rather than the gradients. In fact, the momentum only decreases the variance of $\lambda_k$ (which eschews the "inverse direction" issue that causes non-convergence of MOGA; see the illustration in Figure 1. (a)) and does not really affect the stochastic noises of gradients. Therefore, CR-MOGA can ensure convergence while maintaining the benefit of gradient noise.
> >
> > ### Reference
> > [1] Javaloy, et al. "RotoGrad: Gradient Homogenization in Multitask Learning." ICLR 2021.
> >
> > [2] Reddi, et al. "On the Convergence of Adam and Beyond." ICLR 2018.

---

> > > ### Comment · Reviewer_XWtt · 2022-08-07
> > > **Response to Author**
> > >
> > > Thank you for covering up all the points I raised in such a detailed manner. In particular, your response to Q1 is insightful. I think that the experiment (Table 3 in Appendix G.1) is good because it clearly shows the improvement of CR-MOGA on real data with respect to the stochastic noise. I also apologize for missing the fact that the cost of adding momentum for the $\lambda_k$ is indeed negligible.
> > >
> > > As for the counter-example, it is true that CR-MOGA is theoretically better, but as you know Adam is still widely used, for example, which is why I mentioned that real-data might be "nice" in the sense that we would not encounter the non-converge scenarios in practice that often. Still, the extensive experiments in the appendix show that your algorithm is better in many different cases. I can't comment on the analysis as I only skimmed the proof.
> > >
> > > Overall, from my side, I think that this is good research, so I will increase my rating.

---

### Official Review · Reviewer_nhC2 · 2022-07-11

**Rating:** 6
**Confidence:** 3
**Soundness:** 3 good
**Presentation:** 3 good
**Contribution:** 2 fair

**Summary:**

The main claim of the paper is that the stochastic gradient alteration algorithms do not always converge to Pareto optimal solutions.
The authors focus on multi-objective settings and present a unified algorithmic framework consisting of various gradients with respect to multiple objectives. The authors further propose a novel scheme that averages the previously calculated weights exponentially.
The authors also show that gradient alteration algorithms can converge to an optimal value with the same rate as single objective stochastic optimization by using a momentum-like exponentially averaging mechanism.

**Questions:**

- Could the authors include the error bars for Table 1 and 2 and also in the Figures?
- Could the authors also maybe adjust the Figures to have time in x axis?

**Limitations:**

The limitations are not discussed. There is no potential negative societal impact.

**Strengths And Weaknesses:**

## STRENGTHS
- Multi objective optimization is an important topic, since there are many real life scenarios, applications and this is explained well in the paper.
- The conjecture "altering the stochastic gradients yields a direction that even degrades all the objectives in expectation" is novel and presents a different way of looking at the problem
- The paper is well-written.
## WEAKNESSES
- The presented results seem promising but I kindly request minor extensions for a slightly more convincing experimental results.
- The error bar is missing for the experimental results, since the numerical values are very close it is important to do the analysis for different stochastic settings multiple times.  It is mentioned that the reported value is the mean of 3 runs but it would be still nice to see the error bar and maybe more number of iterations.
- Similarly time analysis along with the accuracy and error values would be great.
- Minor: In the checklist the limitations are said to discussed in Appendix but it is not discussed.

---

> ### Author Response · Authors · 2022-08-01
> **Author Response**
>
> Thanks for your constructive review!
>
> **Q1: The error bar for Tables and Figures.**
>
> **A:** Good suggestion! We have added error bars in all Tables and Figures in Appendix G.2. For the Figures, since we have six lines in the same figure, the error bars further complicate the visualization, as shown in Appendix F in the revision. With this kind of consideration, we maintain the original Figures in the main text, given that the error bars in Table 2 can reveal similar information. For the tables, we will add the version with error bars in the final revision due to the space limitations of the main text.
>
>
> **Q2: Similarly time analysis along with the accuracy and error values would be great. Could the authors also maybe adjust the Figures to have time in x axis?**
>
> **A:** In fact, our modification is only smoothing the composite weight $\lambda_k$, whose dimension $m$ is the same as the number of objectives, which is usually not very large (often 2 or 3). Thus, it introduces nearly negligible extra computational overhead. We have added your mentioned figures with the running time as the x-axis in the revision (Figure 6 in Appendix G.2). They look nearly identical to the original Figures, respectively.
>
> **Q3: In the checklist the limitations are said to discussed in Appendix but it is not discussed.**
>
> **A:** Thanks for pointing this out! Now, we have added the discussion of limitations in Appendix H. Specifically, due to the space limitation, this paper only discusses the typical MOGA algorithms with strict convergence guarantees, i.e., MGDA, PCGrad, and CAGrad. Other empirical-driven methods without rigorous theoretical guarantees, such as RotoGrad [1], are too expensive to be included in our analysis. We believe that the convergence issue in that line of literature will be a very good future work to explore in depth.
>
>
> ### Reference
> [1] Javaloy, et al. "RotoGrad: Gradient Homogenization in Multitask Learning." ICLR 2021.

---

> > ### Comment · Reviewer_nhC2 · 2022-08-08
> > **Reply to authors**
> >
> > Thank you for addressing my questions and providing the missing parts.

---

### Official Review · Reviewer_sBF7 · 2022-07-11

**Rating:** 4
**Confidence:** 3
**Soundness:** 2 fair
**Presentation:** 2 fair
**Contribution:** 2 fair

**Summary:**

This work explores three gradient alteration methods: MGDA, PCGrad and CAGrad  used for multi-objective optimization and  shows that these methods are not guaranteed to converge to Pareto optimal/critical point in the stochastic cases, and illustrates that in a simple 2 stochastic quadratic functions case.
In fact, the included example showed that all three methods generate a direction that would degrade all objective functions. To remedy this issue, the authors proposed a momentum inspired algorithm that would reduce correlation of the stochastic gradients and the composite weights the stochastic gradients calculated to generate a descent direction.
The main claim is that this correlation issue does not become apparent until several iterations of the algorithms, therefore while MGDA, PCGrad and CAGrad will generate a good descent direction in early iterations, they will fail to do so as the algorithms proceeds. and therefore using momentum coefficient that starts close to 0 and convergences to 1 will solve this issue. The authors provide analysis for both convex and nonconvex cases with no assumptions of the lipschitzness of the composite weights .

**Questions:**

302: added sum in the theorem ? $\sum_{i=1}^m$.
760 (second line): why $||E_{\zeta_k}[\| \sum_{i=1}^m (\lambda_i^k \nabla f^i(x_k) )+  d_k\|] || \leq \sqrt{\|||E_{\zeta_k}[\| \sum_{i=1}^m (\lambda_i^k \nabla f^i(x_k) )+  d_k\|] ||\|}$?

**Limitations:**

The property of the momentum coefficient $\alpha_k$ i.e. it starts close $0$ and converges to $1$ and decreasing step size $\eta_k$ propose slower convergence speed than methods that does not use momentum and fixed step-size.
 No comparison with the convergence rate of vanilla MOGA alogorithms is provided.

**Strengths And Weaknesses:**

Strenghts:
This paper serves as a general framework that includes many methods used MOO.
Very straight forward example of a stochastic case where MGDA, PCGrad and CAGrad do not generate a descent direction.
This paper avoids unrealistic assumptions such as gradient-wise lipschitzness of composite weights.
Analysis for two important classes of functions: convex and nonconvex.
Illustrative experiments showing the effectiveness of the proposed approach.
Weaknesses:
No analysis for strongly convex functions.
Assumption of bounded gradients.
Hard to understand analysis.
Critical typos:
225: $E[d_k] = (0,-\epsilon)$  should be $E[d_k] = (0,\epsilon)$ i think.
240: the direction is $- \nabla F(x_k) \lambda_k$.
691: missing $\Sigma$ and added $-$?
716: the bound for the first term is not correct.
Additional typos:
128: $\mu$ should be in subscript.
47: coefficients plural
168: R is in calligraphic style
197: asymptotically converge
581: missing brackets
672: $=$ instead of $\leq$
294: MODA ?
495: Pareto optimal is always weakly Pareto optimal.
702: Double sum $\sum_{i=1}^m$
725: missing sum (second line)
727: $\lambda_k^i \in [0,B]$
747: Before the final proof
766; double $E_{\zeta_k}[\| d_k\|]$
806: missing $n_k^2$

---

> ### Author Response · Authors · 2022-08-01
> **Author Response Part I**
>
> Thanks for your professional and detailed review!
>
> **Q1: Analysis for strongly convex functions.**
>
> **A:** Due to the space limit, we have omitted the analysis for strongly convex functions. In fact, it is easy to derive the convergence rate for strongly convex functions using Lemma 12. Now we have added the corresponding analysis in Appendix F in the revision. Specifically, with a diminishing stepsize and an increasing $\alpha_k$, it achieves $O(1/n)$ convergence rate. And with fixed stepsize and $\alpha_k$, it obtains linear convergence to suboptimal solutions with a predictable optimality gap. Note that this gap is unavoidable, as is the case in single-objective SGD, because the stepsize needs to tradeoff the stochastic variance and the convergence rate (the discussion of Theorem 4.6 in [1]). For more details, please refer to Appendix F in the revision.
>
> Furthermore, since multi-objective optimization could reduce to single-objective optimization if the multiple objectives are the same, the convergence rate of stochastic MOO will be no better than single-objective SGD. Note that our convergence rates align with those in single-objective SGD (Theorem 4.6 for fixed stepsize and Theorem 4.7 for diminishing stepsize in [1]), which suggests that our bounds are tight in order.
>
> **Q2: Assumption of bounded gradients.**
>
> **A:** In fact, the assumption of bounded gradients appears very widely in related literature on stochastic optimization. Specifically, in the convex setting, the bounded gradient assumption is used to derive the convergence bound for various famous optimizers, such as Adam (Theorem 4.1 in [2]), Adagrad (Theorem 5 in [3]), AMSGrad (Theorem 4 in [4]), etc. In the non-convex setting,  it helps to analyze the stochastic convergence of momentum (Theorem 1 \& 2 in [5], Assumption 3 in [6], Assumption 2 in [7], and Assumption 1(ii) in [8]). Additionally, in practice with deep learning, there are many techniques that are designed to avoid gradient explosion, such as weight regularization [9] and gradient clipping [10]. These techniques ensure the boundness of the gradient in practice, which validates this assumption. Furthermore, our theoretical results only require that $\|\nabla f^i (x_k)\|\leq H$ for $i=1,\ldots,m$ and $k=1,\ldots,n$, i.e., the gradients along the optimization path are bounded. In fact, many works of related literature adopt this version. Therefore, we have added further discussion that justifies the relaxed assumption in Appendix E.7 in the revision.
>
> **Q3: Hard to understand analysis**
>
> **A:** Due to the page limit, we have not included the proof sketch in the main paper. In fact, our analysis has a clear structure. Specifically, Appendix E.4 and E.5 decompose the convex and non-convex convergence bounds into sub-terms bounded by lemmas from Appendix E.3. In order to help better understand our analysis, we have added a proof introduction at the beginning of Appendix E, and fixed the typos. Also, we will refine the presentation of the proof in the final version.
>
> **Q4: 716: the bound for the first term**
>
> **A:** We apologize for the confusion. We aimed to demonstrate that the local decrease is bounded by the sum of the first term (line 716) and the second term (linear 718) in Lemma 7. We feel that you might have a misunderstanding that the decrease is only bounded by the first term, presumably because this sentence is too long in the original version. We have rewritten this sentence in the revision to eschew possible misunderstandings. Specifically, the revised sentence is:
>
> "_The result from Lemma 7 indicates that the local decrease for expected composite loss $\mathbb{E}[\boldsymbol{\lambda}\_k]\^T \boldsymbol{F}(\boldsymbol{x})$ is bounded by $2mBH\sqrt{ \mathbb{V}\_{\boldsymbol{\xi}\_k}[\boldsymbol{\lambda}\_k] \sum\_{i=1}\^m\mathbb{V}\_{\boldsymbol{\xi}\_k} [\boldsymbol{g}\^i(\boldsymbol{x}\_k)]}-\mathbb{E}\_{\boldsymbol{\xi}\_k}\left[\left||\sum\_{i=1}\^m  \lambda\_k^i\nabla f\^i (\boldsymbol{x}\_{k})\right||\_2\^2\right]$. The first term consists of the variances for composite weights and multiple gradients, and the second term is the negative norm of the effective direction._"
>
>
> **Q5: Added sum in the theorem 2?**
>
> **A:** The added sum represents the average scheme version of convergence that is typically used in the analysis of stochastic optimization [5,11] and multiobjective optimization [12], as described in the remark below Theorem 3. We have also discussed how to transform the average scheme into the traditional convergence bound in Appendix E.7. It just needs to use a typical uniform sampling technique in stochastic optimization to modify the algorithmic output [5,13]. We have highlighted the discussion in the main text of the revision for better understanding.

---

> > ### Author Response · Authors · 2022-08-01
> > **Author Response Part II**
> >
> > **Q6: 760 (second line): why $\left||\mathbb{E}\_{\boldsymbol{\xi}\_k} \left[\sum\_{i=1}\^m \lambda\_k\^i \nabla f\^i(\boldsymbol{x}\_k) + \boldsymbol{d}\_k\right]\right||\_2 \leq  \sqrt{\left||\mathbb{E}\_{\boldsymbol{\xi}\_k} \left[\sum\_{i=1}\^m \lambda\_k\^i \nabla f\^i(\boldsymbol{x}\_k) + \boldsymbol{d}\_k\right]\right||\_2}$?**
> >
> > **A:** We apologize for the confusion. This is a typo that we have missed a square symbol, and we have revised it as $\left||\mathbb{E}\_{\boldsymbol{\xi}\_k} \left[\sum\_{i=1}\^m \lambda\_k\^i \nabla f\^i(\boldsymbol{x}\_k) + \boldsymbol{d}\_k\right]\right||\_2 =  \sqrt{\left||\mathbb{E}\_{\boldsymbol{\xi}\_k} \left[\sum\_{i=1}\^m \lambda\_k\^i \nabla f\^i(\boldsymbol{x}\_k) + \boldsymbol{d}\_k\right]\right||\_2\^2}$. In fact, this is an obvious equation.
> >
> > **Q7: The property of the momentum coefficient $\alpha_k$ i.e. it starts close 0 and converges to 1 and decreasing step size $\eta_k$ propose slower convergence speed than methods that do not use momentum and fixed step-size.**
> >
> > **A:** In fact, the momentum in CR-MOGA is applied to the composite weights $\lambda_k$ rather than to the gradients. Hence, in our algorithm, the gradients retain the same. Thus, our modification does not impact the convergence rate, unlike the gradient momentum slows down the rate of SGD. In fact, we have discussed the tightness of the convergence order for CR-MOGA under non-convex, convex settings (see Corollary 1). Additionally, we also discuss its tightness under the strongly convex setting in Corollary 2 \& 3 (Appendix F) in the revision. Note that Corollary 2 specifically discusses the rate of CR-MOGA with fixed stepsizes.
> >
> > **Q8: No comparison with the convergence rate of vanilla MOGA algorithms is provided.**
> >
> > **A:** In comparison with stochastic MOGA, we have demonstrated clearly in this paper that stochastic MOGA algorithms fail to converge to the Pareto optimal/critical points in Section 4, so comparing them is not very meaningful. Also, we have proved that CR-MOGA matches the bound for single-objective SGD, which is the best rate that can be achieved for stochastic MOO.
> >
> > As for comparing with full-batch MOGA, the difference between them is just like GD with SGD.
> >
> > The above discussions have been added in the revision.
> >
> > **Q9: Other typos.**
> >
> > **A:** Thanks for pointing them out! We have revised all of them correspondingly in the revision.
> >
> > ### Reference
> > [1] Léon, et al. "Optimization methods for large-scale machine learning."  Siam Review 2018.
> >
> > [2] Kingma, et al. "Adam: A Method for Stochastic Optimization." ICLR 2015. (Theorem 4.1)
> >
> > [3] Duchi, et al. "Adaptive subgradient methods for online learning and stochastic optimization." JMLR 2011. (Convergence depends on bounded gradient in Theorem 5)
> >
> > [4] Reddi, et al. "On the Convergence of Adam and Beyond." ICLR 2018. (Theorem 4)
> >
> > [5] Yan, et al. "A unified analysis of stochastic momentum methods for deep learning." IJCAI 2018. (Theorem 1 & 2)
> >
> > [6] Cutkosky, et al. "Momentum-based variance reduction in non-convex sgd." NeurIPS 2019. (Assumption 3 in Section 3)
> >
> > [7] Yang, et al. "Provably faster algorithms for bilevel optimization." NeurIPS 2021. (Assumption 2)
> >
> > [8] Khanduri, et al. "A near-optimal algorithm for stochastic bilevel optimization via double-momentum." NeurIPS 2021. (Assumption 1 (ii))
> >
> > [9] Pascanu, et al. "On the difficulty of training recurrent neural networks." ICML 2013.
> >
> > [10] Zhang, et al. "Why Gradient Clipping Accelerates Training: A Theoretical Justification for Adaptivity." ICML 2019.
> >
> > [11] Ohad Shamir and Tong Zhang.  Stochastic gradient descent for non-smooth optimization Convergence results and optimal averaging schemes. ICML 2013.
> >
> > [12] Liu, et al. "Conflict-averse gradient descent for multi-task learning." NeurIPS 2021. (Theorem 3.2)
> >
> > [13] Drori, et al. "The complexity of finding stationary points with stochastic gradient descent." ICML 2020.

---

> > > ### Comment · Reviewer_sBF7 · 2022-08-09
> > > **Thank you for answering my questions and revising the typos.**
> > >
> > > Thank you for answering my questions and revising the typos.

---

### Author Response · Authors · 2022-08-02
**General Author Response**

Dear all reviewers,

Thanks for your valuable comments. We hope our response addresses the concerns. We are happy to address any additional questions during the discussion phase. We apologize for uploading the response and the revision late, mainly due to the limited time and the enormous additional workload on analysis and experiments. We will complete the revision in the coming day.

---

### Author Response · Authors · 2022-08-09
**Update**

Dear Reviewers：

Thank you very much for reading our rebuttal and giving feedbacks!

As for Reviewer sBF7 and Reviewer eaLg, both of you have pressed the "author rebuttal acknowledgment" button, but did not provide more specific comments. So we're very confused about whether our response sufficiently addresses your concerns or you still have additional questions. Since the rolling discussion window is still open, we respectfully remind you that, if possible, you can post more additional comments on our response. We're readily prepared to answer them in time, enabling more sufficient discussion. We greatly appreciate your time and efforts!

---

### Meta-Review · Area_Chair_qrKb · 2022-08-27

**Recommendation:** Accept
**Confidence:** Less certain

**Metareview:**

This paper analyses a multi-objective minimization problem. Author(s) show that several methods, such as PCGrad, MGDA, and CAGrad, can fail to even converge to Pareto optimal solutions. On top of that, they are mostly analyzed in a batch setting.
Therefore, the author(s) design a carefully crafted problem where these phenomena can be studied.
By averaging past weights with a carefully designed scheme, they proposed a new algorithm that can provably converge to a Pareto optimal solution. I believe that the NeurIPS community will benefit from this paper, and therefore I recommend acceptance.

Please, for the camera-ready version, please incorporate the suggestions from the authors and explain better parts that were not clear.

Thanks


**Award:**

No

---

### Decision · Program_Chairs · 2022-09-14

Accept